# PRISM: Progressive Robust Learning for Open-World Continual Category Discovery

**Wei Feng**[1,2]   **Sijin Zhou**[1,2]   **Yiwen Jiang**[1]   **Zongyuan Ge**[1,2*]

[1]AIM for Health Lab, Monash University

[2]Airdoc–Monash Research, Monash University

{wf02429, sjzhou1995}@gmail.com,{yiwen.jiang, zongyuan.ge}@monash.edu

## Abstract

Continual Category Discovery (CCD) aims to leverage models trained on known categories to automatically discover novel category concepts from continuously arriving streams of unlabeled data, while retaining the ability to recognize previously known classes. Despite recent progress, existing methods often assume that data across all stages are drawn from a single, stationary distribution—a condition rarely satisfied in open-world scenarios. In this paper, we challenge this stationary-distribution assumption by introducing the Open-World Continual Category Discovery (OW-CCD) setting. We address this challenge with PRISM (Progressive Robust dIscovery under StreaMing data), an adaptive continual discovery framework consisting of three key components. First, inspired by spectral properties, we develop a high-frequency-driven category separation technique that exploits high-frequency components—preserving more global information—to distinguish known from unknown categories. Second, for known categories, we design a sparse assignment matching strategy, which performs proximal sparse sample-to-label matching to assign reliable cluster labels to known-class samples. Finally, to better recognize novel categories, we propose an invariant knowledge transfer module that enforces domain-invariant category relation consistency, thereby facilitating robust knowledge transfer from known to unknown classes under domain shifts. Extensive experiments on the SSB-C and DomainNet benchmarks demonstrate that our method significantly outperforms state-of-the-art CCD approaches, highlighting its effectiveness and superiority.

## 1 Introduction

Visual concepts in the real world are open-ended and continually evolving, far exceeding any predefined category set (Feng et al., 2023a; 2022; Ju et al., 2024; Jiang et al., 2025; Tang et al., 2025b; Zhang et al., 2025; Ma et al., 2025a). Although deep learning has achieved remarkable progress in visual recognition (Hang et al., 2020; Jiang et al., 2025; Feng et al., 2025a; Hakeem et al., 2022; Ma et al., 2025a; Zhang et al., 2025), most advances rely on closed-world assumptions—models are trained on fixed label spaces and therefore struggle when encountering previously unseen categories. Humans, by contrast, naturally generalize from prior knowledge to organize and recognize new concepts. This discrepancy has led to growing interest in category discovery (Vaze et al., 2022; Han et al., 2021; Wen et al., 2023).

Early studies formulated this task as Novel Class Discovery (NCD) (Han et al., 2021; Feng et al., 2023a), where all unlabeled samples belong to novel categories. To better reflect realistic conditions, Generalized Category Discovery (GCD) (Vaze et al., 2022; Wen et al., 2023; Feng et al., 2025b) extended this setting by allowing unlabeled data to contain a mixture of known and unknown classes, requiring models to both identify known classes and cluster new ones. However, both NCD and GCD are built upon static datasets and assume simultaneous access to labeled and unlabeled data. These assumptions diverge from real-world conditions, where data typically arrive as continuously evolving, unlabelled streams. As a result, NCD and GCD overlook the dynamic nature of open

---

[*]Corresponding author.

environments and fall short of modeling realistic data-stream scenarios. To close this gap, the community has recently moved toward Continual Category Discovery (CCD) (Park et al., 2024; Cendra et al., 2024), which integrates continual learning with category discovery. CCD aims to progressively identify emerging categories while preventing catastrophic forgetting of previous knowledge. Despite this progress, existing CCD settings commonly assume that data at each stage comes from a single, fixed domain—an assumption rarely met in open environments. In practical scenarios, samples may originate from diverse sources or shift across domains while new categories appear. For example, an online platform may continuously receive animal images from different cameras or users; as the domain (e.g., lighting, style, device) changes, rare species can emerge concurrently with existing categories.

Motivated by these limitations, we propose a more realistic setting called **Open-World Continual Category Discovery (OW-CCD)**. In OW-CCD, models must automatically discover known and unknown categories from unlabeled streams without assuming domain consistency. This introduces several challenges. First, it is difficult to preserve recognition ability for known categories under distribution shifts, as existing CCD methods are not designed to handle domain variations. Second, the model must continually discover emerging categories in dynamic, non-stationary streams. Traditional domain adaptation techniques are unsuitable, as they often assume overlapping label spaces; naïve alignment may even cause negative transfer and suppress novel-category discovery. Moreover, most adaptation methods focus on aligning known classes, offering little guidance for discovering unseen ones.

To address these challenges, we introduce **PRISM** (Progressive Robust dIscovery under StreaMing data), a new adaptive divide-and-conquer framework for OW-CCD. Our design is inspired by spectral analysis (Fig. 1(a–c)): high-frequency components tend to capture domain-invariant global semantics (e.g., structures), whereas low-frequency components encode domain-dependent details (e.g., style). Leveraging this insight, we develop a high-frequency-driven category separation module to distinguish known from unknown samples under domain shift. To ensure reliable recognition of known categories, we further propose a sparse assignment matching module based on proximal optimal transport, producing stable and sparse pseudo-labels. Finally, following the core principle of category discovery—transferring knowledge from known to unknown classes through semantic relations—we introduce an invariant knowledge transfer (IKT) module. Instead of relying on domain-specific cues that may distort associations, IKT represents the relations between known and unknown classes as ranking permutations. These permutations are converted into ranking probability distributions and enforced

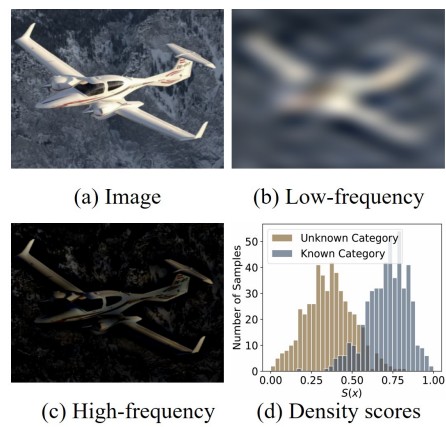

(a) Image  (b) Low-frequency

(c) High-frequency  (d) Density scores

Figure 1: (a-c) Visualization of low-frequency and high-frequency components of the images. (d) Visualization of the density scores $S(x)$, where the density distribution exhibits a clear bimodal pattern corresponding to known and unknown samples.

to remain consistent across domains. This rank-based formulation ensures that semantically closer classes contribute stronger knowledge transfer; for instance, in CUB, closely related bird species such as Indigo Bunting and Lazuli Bunting share meaningful high-level semantics despite subtle visual differences. Maintaining these relationships across domains enables stable transfer and facilitates robust discovery of novel categories.

In summary, our contributions are as follows: (1) we introduce the Open-World Continual Category Discovery (OW-CCD) setting and present PRISM, an adaptive divide-and-conquer framework; (2) we propose a high-frequency-driven category separation strategy to distinguish known and unknown samples under domain shifts; (3) we design a sparse assignment matching module based on proximal optimal transport for reliable pseudo-labeling of known categories; (4) we develop an invariant knowledge transfer module that preserves semantic relations between known and unknown categories across domains for stable discovery; and (5) through extensive evaluation on the SSB-C and

DomainNet benchmarks, we demonstrate that PRISM achieves strong effectiveness and robustness, consistently outperforming state-of-the-art CCD approaches.

## 2 RELATED WORK

**Category Discovery.** Category discovery aims to leverage knowledge from labeled classes to identify novel concepts in unlabeled data. Novel Class Discovery (NCD) was introduced to address scenarios where all unlabeled samples belong to unseen categories. Early methods adopted two-stage pipelines, such as AutoNovel (Han et al., 2021), which transfers knowledge through self-supervised learning with ranking statistics. In contrast, unified end-to-end approaches (Fini et al., 2021) directly integrate representation learning and clustering into a single stage. Later extensions address sample imbalance by designing self-cooperation mechanisms that leverage both known and novel representations for mutual learning (Wang et al., 2024c), or enhance class-level knowledge transfer through symmetric relationship modeling and pairwise consistency constraints (Zhou & Chen, 2025). Generalized Category Discovery (GCD) relaxes this setting by mixing known and unknown categories. Early frameworks combined supervised and unsupervised contrastive learning with clustering (Vaze et al., 2022; Feng & Ge, 2025), while SimGCD (Wen et al., 2023) introduced a parametric classifier for efficiency. More recent work explores hierarchical modeling (Liu et al., 2025b), prototype-based learning (Ma et al., 2025b), and reciprocal learning with distribution regularization (Liu et al., 2025a). Beyond these directions, some studies investigate domain-level extensions, addressing category discovery under domain shifts (Wang et al., 2024a; Rongali et al., 2024). Continuous Category Discovery (CCD) further considers streaming settings. Methods such as grow and merge (Zhang et al., 2022), energy-guided discovery (Park et al., 2024), Gaussian mixture prompting (Cendra et al., 2024), and Bayesian inference (Dai & Chauhan, 2025) have been proposed to tackle class discovery in streaming data, though they often operate under the simplifying assumption of single-domain streams.

**Domain Adaptation.** Domain adaptation tackles distribution gaps between labeled source and target domains. Unsupervised domain adaptation (UDA) leverages labeled source and unlabeled target data, typically by learning domain-invariant representations. Discrepancy-based methods minimize moment mismatches (Sun & Saenko, 2016; Long et al., 2015; Tzeng et al., 2014; Feng et al., 2022; 2023b), while adversarial approaches employ domain discriminators (Saito et al., 2018a; Sankaranarayanan et al., 2018). Transformer-based backbones (Dosovitskiy et al., 2020) have also been explored with attention-driven alignment (Sun et al., 2022; Xu et al., 2021). Source-Free Domain Adaptation (SFDA) removes source data access. Representative works include prototype transfer (Chidlovskii et al., 2016), iterative pseudo-labeling (Liang et al., 2019), SHOT (Krause et al., 2010; Shi & Sha, 2012), and neighborhood regularization (Yang et al., 2021). Beyond this, Open-Set Domain Adaptation (OSDA) addresses unknown target categories. Strategies include confidence thresholding (Saito et al., 2018b), progressive separation (Liu et al., 2019), and causal adjustment (Li et al., 2023b). While OSDA extends domain adaptation to more realistic scenarios, most existing methods remain centered on classifying known categories and pay limited attention to systematically exploring the unknown label space.

## 3 METHODOLOGY

### 3.1 PROBLEM STATEMENT

Open-World Continual Category Discovery (OW-CCD) involves one base learning session followed by $T$ online continual discovery sessions. In the base session, we are provided with a labeled dataset $\mathcal{D}^l = \{(x_i, y_i)\}_{i=1}^{N^l}$ consisting of $N^l$ labeled samples drawn from a known category space $\mathcal{C}^l$. In each subsequent online discovery session, an unlabeled data stream $\mathcal{D}_t^u = \{x_i\}_{i=1}^{N_t^u}$ is introduced incrementally. This stream not only contains samples from the previously seen known categories, but also includes samples from novel categories $\mathcal{C}_t^u$ in session $t$. Moreover, these samples may originate from domains that differ from the domain distributions observed in the base session, thus introducing additional domain shift. The goal of OW-CCD is to enable the model to robustly discover novel concepts from the dynamic unlabeled data stream in an online manner, while simultaneously main-

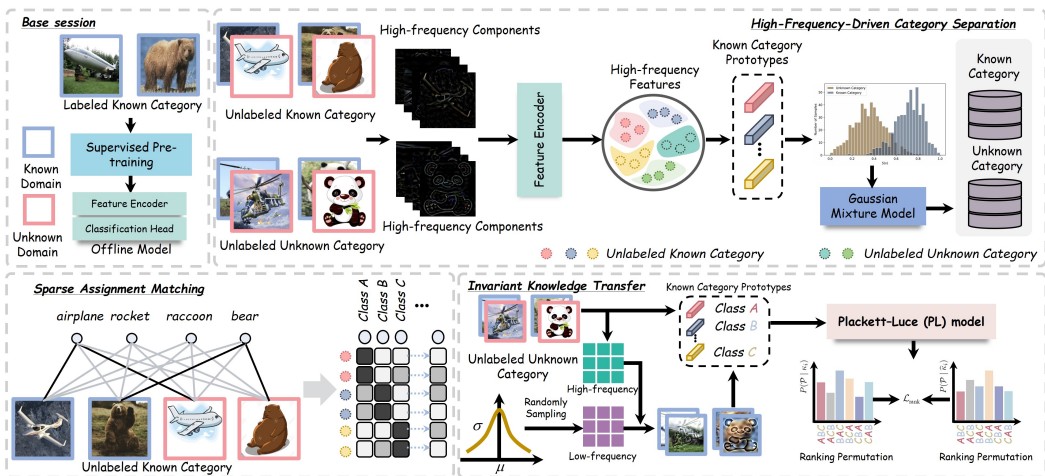

Figure 2: The overall framework of our proposed method.

taining recognition capability for known categories and alleviating the impact of distribution shifts as much as possible.

Fig. 2 illustrates the overall framework of our proposed PRISM method. In the base session, we first pre-train a model $\theta = \{f, g\}$ on the labeled dataset using cross-entropy loss, where $f(\cdot)$ denotes the feature extractor and $g(\cdot)$ is the classifier head. This provides discriminative representations as a foundation for subsequent discovery. In the online discovery stage, we then introduce three key innovations. First, the High-Frequency-Driven Category Separation (HCS) is employed to automatically separate known and unknown categories by exploiting high-frequency information in images. Second, the Sparse Assignment Matching (SAM) module assigns reliable cluster labels to samples from known categories. Finally, the Invariant Knowledge Transfer (IKT) module captures robust category associations across different domains, thereby enabling stable and effective novel category discovery.

## 3.2 HIGH-FREQUENCY-DRIVEN CATEGORY SEPARATION (HCS)

Since direct distribution alignment may lead to negative transfer, we adopt a divide-and-conquer strategy. Below, we first describe how to extract the high- and low-frequency components of images, and then employ the HCS module for category separation. Given an input image $x_i \in \mathbb{R}^{H \times W \times C}$, where $H$, $W$, and $C$ denote the height, width, and number of channels, respectively, we first transform it into the frequency domain using the Discrete Fourier Transform (DFT):

$$\mathcal{F}(x_i)(u, v, c) = \sum_{h=0}^{H-1} \sum_{w=0}^{W-1} x_i(h, w, c) \, e^{-j2\pi \left( \frac{hu}{H} + \frac{wv}{W} \right)}, \tag{1}$$

where $j^2 = -1$, $u$ and $v$ denote the spatial frequency coordinates, and $c$ indexes the RGB channels. Following common practice, the low-frequency components are shifted to the center of the spectrum for convenience. To separate low- and high-frequency information, we construct a binary mask $M \in \mathbb{R}^{r \times r}$:

$$M_{u,v} = \begin{cases} 1, & \text{if } \max(|u - \frac{H}{2}|, |v - \frac{W}{2}|) \leq r \cdot \frac{\min(H,W)}{2}, \\ 0, & \text{otherwise}, \end{cases} \tag{2}$$

where $r$ controls the relative size of the mask. The low- and high-pass frequency components are then obtained as:

$$\mathcal{F}^l(x_i) = M \odot \mathcal{F}(x_i), \quad \mathcal{F}^h(x_i) = (I - M) \odot \mathcal{F}(x_i), \tag{3}$$

where $\odot$ denotes element-wise multiplication, and $\mathcal{F}^l$ and $\mathcal{F}^h$ are the masked low- and high-frequency spectra obtained from the DFT $\mathcal{F}(x_i)$. Finally, inverse DFT is applied to recover spatial representations of low- and high-frequency images: $x_i^l = \mathcal{F}^{-1}(\mathcal{F}^l(x_i)), x_i^h = \mathcal{F}^{-1}(\mathcal{F}^h(x_i))$.

We focus on the high-frequency component $x_i^h$ of the unlabeled data, as high-frequency cues often contain discriminative structural information that helps distinguish known from unknown categories (see Fig. 1(a-c)). These components are fed into the pre-trained feature extractor $f$ from the previous stage to obtain high-frequency representations. Based on these representations, we define a density scoring function: $S(x) = \nu\left(\max_c \frac{f(x^h)\cdot e_c}{\|f(x^h)\|\cdot\|e_c\|}\right)$, where $e_c$ denotes the prototype of the known class $c$ from the previous stage, and $\nu(\cdot)$ is a min–max normalization mapping scores to $[0, 1]$. Intuitively, $S(x)$ measures the maximum similarity between the high-frequency representation of an unlabeled sample and known category prototypes: larger values indicate that the sample is closer to known classes, while smaller values imply it may belong to an unknown class. As shown in Fig. 1(d), we empirically observe that the distribution of $S(x)$ often exhibits a bimodal shape, corresponding to known and unknown samples, respectively. To automatically separate them, we model the distribution of $S(x)$ using a two-component Gaussian Mixture Model (GMM): $P(x) = \pi(x)\mathcal{N}(x|\mu_{\text{kno}}, \sigma_{\text{kno}}^2) + (1-\pi(x))\mathcal{N}(x|\mu_{\text{unk}}, \sigma_{\text{unk}}^2)$, where $\pi(x)$ is the posterior probability of being assigned to the known component, estimated using the EM algorithm. $\mathcal{N}(x \mid \mu, \sigma^2)$ denotes a Gaussian distribution with mean $\mu$ and variance $\sigma^2$, and $(\mu_{\text{kno}}, \sigma_{\text{kno}}^2)$ and $(\mu_{\text{unk}}, \sigma_{\text{unk}}^2)$ correspond to the parameters of the known and unknown components, respectively. Finally, at each online session $t$, we split the incoming data stream $\mathcal{D}_t^u$ into known and unknown subsets:

$$\mathcal{D}_{t,\text{kno}}^u = \{x \mid x \in \mathcal{D}_t^u \ \wedge \ \pi(x) \geq 0.5\}, \quad \mathcal{D}_{t,\text{unk}}^u = \{x \mid x \in \mathcal{D}_t^u \ \wedge \ \pi(x) < 0.5\}. \tag{4}$$

This separation provides a reliable mechanism for dynamically identifying known-like and unknown-like samples, which lays the foundation for subsequent discriminative learning and novel category discovery.

### 3.3 SPARSE ASSIGNMENT MATCHING (SAM)

Since we have already obtained the known-category samples via the proposed HCS module, we next explore the possible labels for these known samples $x_{\text{kno}} \in \mathcal{D}_{t,\text{kno}}^u$. As $x_{\text{kno}}$ shares the same semantic space with the known prototypes from the previous stage, domain adaptation techniques can be employed to uncover the latent alignment. In this process, Optimal Transport (OT) provides a powerful tool to automatically discover proper sample–prototype correspondences across domains, thereby alleviating domain discrepancies (Courty et al., 2014; Flamary et al., 2016). However, directly solving the OT problem with linear programming incurs a prohibitive computational cost (Xu & Dan, 2025); even though the entropy regularization can

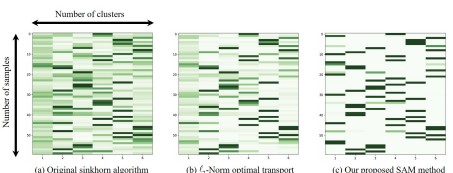

(a) Original sinkhorn algorithm  (b) $\ell_2$-Norm optimal transport  (c) Our proposed SAM method

Figure 3: Toy illustration of our proposed SAM method. Conventional Sinkhorn and $\ell_2$-regularized solvers tend to produce smooth yet dense couplings, whereas the proposed SAM yields a sparser transport plan with better performance.

improve efficiency, it usually yields overly dense transport plans, leading to inaccurate assignments as illustrated in Fig. 3(a). To overcome this limitation, we propose a Sparse Assignment Matching (SAM) scheme by incorporating an $\ell_2$-norm proximal term. The optimization objective is formulated as:

$$\min_{\gamma \in \Delta} \ell(\gamma) = \sum_{i=1}^{N_{t,\text{kno}}} \sum_{j=1}^{\mathcal{C}^{t-1}} \left[\gamma_{ij} C_{ij} + \frac{\varepsilon}{2}(\gamma_{ij} - \gamma_{ij}^{(l)})^2\right], \tag{5}$$

where $N_{t,\text{kno}}$ denotes the number of samples in $\mathcal{D}_{t,\text{kno}}^u$. $\mathcal{C}^{t-1}$ denotes the number of known classes from the previous stage. The cost matrix is defined as $C_{ij} = -\log\left(g\left(f(x_{i,\text{kno}})\right)_j\right)$. A smaller value of $C_{ij}$ indicates that sample $x_{i,\text{kno}}$ has a higher probability of belonging to the $j$-th class, while a larger cost suggests a less plausible assignment. The second term $\frac{\varepsilon}{2}\sum_{i,j}(\gamma_{ij} - \gamma_{ij}^{(l)})^2$ acts as an $\ell_2$-based proximal regularizer, which suppresses oscillations across iterations and enforces sparse and stable solutions. Meanwhile, $\Delta = \left\{\sum_{j=1}^{\mathcal{C}^{t-1}} \gamma_{ij} = \hat{a}_i = 1, \sum_{i=1}^{N^{t,\text{kno}}} \gamma_{ij} = \hat{b}_j = \frac{N_{t,\text{kno}}}{\mathcal{C}^{t-1}}, \gamma_{ij} \geq 0\right\}$ is the feasible set. We initialize $\gamma_{ij}^{(0)} = \frac{1}{\mathcal{C}^{t-1}}$ for the following iterations. To avoid directly handling the

constrained problem, we first obtain the Fenchel–Legendre dual formulation of the original problem:

$$\max_{\psi,\varphi} \sum_{i=1}^{N_{t,\text{kno}}} \psi_i \hat{a}_i + \sum_{j=1}^{\mathcal{C}^{t-1}} \varphi_j \hat{b}_j - \frac{\varepsilon}{2} \sum_{i=1}^{N_{t,\text{kno}}} \sum_{j=1}^{\mathcal{C}^{t-1}} \left[ \frac{\psi_i + \varphi_j - \tilde{C}_{ij}}{\varepsilon} \right]_+^2, \tag{6}$$

where $\varphi$ and $\psi$ denote the Lagrange multipliers, and $[z]_+^2 = (\max\{0, z\})^2$ denotes the truncated quadratic operator ensuring non-negativity. $\tilde{C}_{ij} = C_{ij} - \varepsilon \gamma_{ij}^{(l)}$. Note that the original constrained OT problem in Eq. (5) is transformed into an unconstrained optimization over $(\psi, \varphi)$, which is computationally more efficient. The detailed optimization procedure for solving $\psi$ and $\varphi$ is provided in the Appendix. Then the transport plan $\gamma^{(l+1)}$ can be updated in closed form as: $\gamma_{ij}^{(l+1)} = \max\left\{0, (\psi_i + \varphi_j + \varepsilon \gamma_{ij}^{(l)} - C_{ij})/\varepsilon\right\}$.

With the optimal transport plan $\gamma^*$ obtained from sparse sample-prototype alignment, we assign reliable pseudo labels to the known samples $x_{\text{kno}} \in \mathcal{D}_{t,\text{kno}}^u$. Moreover, compared with the standard $\ell_2$-regularized OT, the proposed SAM produces significantly sparser and clearer assignment patterns, as depicted in Fig. 3(b)–(c), which contributes to more trustworthy pseudo-label generation.

## 3.4 INVARIANT KNOWLEDGE TRANSFER (IKT)

For unknown category discovery, we build on the core idea of category discovery—transferring category knowledge from known to unknown classes by exploiting their semantic associations. Under domain shift, however, such associations may be distorted by domain-specific style factors, leading the model to capture spurious rather than genuine relations. We argue that effective discovery should instead depend on domain-invariant category associations that reflect stable semantic structures. To this end, we propose an *Invariant Knowledge Transfer* module, which explicitly models the relationships between unknown samples and known prototypes across domains and enforces their invariance, thereby facilitating the transfer of authentic semantic knowledge. Specifically, in each previous stage we pre-compute the frequency-domain statistics of the known domain. For this purpose, we apply the discrete Fourier transform as in Eq. (1) and decompose each spectrum into low- and high-frequency components $\{\mathcal{F}^l, \mathcal{F}^h\}$ according to Eq. (3). Inspired by recent works (Li et al., 2022; Wang et al., 2022) that employ spatial feature statistics to characterize style, we characterize the low-frequency spectrum by the channel-wise mean and standard deviation:

$$\mu(\mathcal{F}_i^l) = \frac{1}{HW} \sum_{u,v} \mathcal{F}_i^l(u, v, c), \quad \sigma(\mathcal{F}_i^l) = \frac{1}{HW} \sum_{u,v} [\mathcal{F}_i^l(u, v, c) - \mu(\mathcal{F}_i^l)]^2. \tag{7}$$

We assume that the distribution of these statistics follows a Gaussian distribution, and estimate their variances across the data from the previous stage:

$$\Sigma_\mu^2(\mathcal{F}_i^l) = \frac{1}{N_{t-1}} \sum_{i=1}^{N_{t-1}} \left[ \mu(\mathcal{F}_i^l) - \mathbb{E}[\mu(\mathcal{F}_i^l)] \right]^2, \quad \Sigma_\sigma^2(\mathcal{F}_i^l) = \frac{1}{N_{t-1}} \sum_{i=1}^{N_{t-1}} \left[ \sigma(\mathcal{F}_i^l) - \mathbb{E}[\sigma(\mathcal{F}_i^l)] \right]^2, \tag{8}$$

where $N_{t-1}$ denotes the number of samples from the previous stage. We then sample perturbed low-frequency statistics from these Gaussian distributions:

$$\hat{\mu}(\mathcal{F}_i^l) = \mu(\mathcal{F}_i^l) + \epsilon_\mu \Sigma_\mu(\mathcal{F}_i^l), \quad \epsilon_\mu \sim \mathcal{N}(0, 1), \quad \hat{\sigma}(\mathcal{F}_i^l) = \sigma(\mathcal{F}_i^l) + \epsilon_\sigma \Sigma_\sigma(\mathcal{F}_i^l), \quad \epsilon_\sigma \sim \mathcal{N}(0, 1). \tag{9}$$

For an unknown sample $x_{i,\text{unk}}^t$ in the current data stream, we extract its low-/high-frequency components $(\mathcal{F}_{i,\text{unk}}^l, \mathcal{F}_{i,\text{unk}}^h)$ via Eq. (3) and reconstruct the low-frequency spectrum with the sampled statistics:

$$\widehat{\mathcal{F}}_{i,\text{unk}}^l = \hat{\sigma}(\mathcal{F}_i^l) \cdot \frac{\mathcal{F}_{i,\text{unk}}^l - \mu(\mathcal{F}_{i,\text{unk}}^l)}{\sigma(\mathcal{F}_{i,\text{unk}}^l)} + \hat{\mu}(\mathcal{F}_i^l). \tag{10}$$

Finally, we combine $\widehat{\mathcal{F}}_{i,\text{unk}}^l$ with the original high-frequency part $\mathcal{F}_{i,\text{unk}}^h$ to form an augmented spectrum $\widehat{\mathcal{F}}(x_{i,\text{unk}}^t)$ and apply inverse DFT to obtain the augmented sample $\hat{x}_{i,\text{unk}}^t$. Then, for each unknown-category sample $x_{i,\text{unk}}^t$ and its style-transferred counterpart $\hat{x}_{i,\text{unk}}^t$, we extract their feature representations: $z_{i,\text{unk}}^t = f(x_{i,\text{unk}}^t), \hat{z}_{i,\text{unk}}^t = f(\hat{x}_{i,\text{unk}}^t)$. Let $\{e_c^{t-1}\}_{c=1}^{\mathcal{C}^{t-1}}$ denote the set of known class

prototypes from the previous stage. For each unknown sample, we convert its cosine similarities to these prototypes under both the original and style-transferred views into the *strength parameters* of the Plackett–Luce (PL) model: $\kappa_{i,c} = \exp\big(\cos(z_{i,\text{unk}}^t, e_c^{t-1})\big), \widehat{\kappa}_{i,c} = \exp\big(\cos(\widehat{z}_{i,\text{unk}}^t, e_c^{t-1})\big)$. As the number of known classes grows, enumerating or even implicitly handling all $\mathcal{C}^{t-1}!$ permutations becomes infeasible. Therefore, following standard practice in listwise ranking (e.g., ListMLE (Xia et al., 2008)), we adopt the *factorized* PL likelihood, whose sequential decomposition provides an analytically exact and computationally tractable form of the model. Given $\kappa_i = \big\{\kappa_{i,1}, \ldots, \kappa_{i,\mathcal{C}^{t-1}}\big\}$, the likelihood of a permutation $\xi$ is:

$$P(\xi \mid \kappa_i) = \prod_{k=1}^{\mathcal{C}^{t-1}} \frac{\kappa_{i,\xi(k)}}{\sum_{k'=k}^{\mathcal{C}^{t-1}} \kappa_{i,\xi(k')}}, \tag{11}$$

where $\xi(k)$ denotes the prototype placed at position $k$. For illustration, when $\mathcal{C}^{t-1} = 3$ and $\xi = (a, b, c)$, $P(\xi \mid \kappa_i) = \frac{\kappa_{i,a}}{\kappa_{i,a} + \kappa_{i,b} + \kappa_{i,c}} \cdot \frac{\kappa_{i,b}}{\kappa_{i,b} + \kappa_{i,c}} \cdot \frac{\kappa_{i,c}}{\kappa_{i,c}}$. This example reflects the inherent sequential normalization of the PL model and does not require constructing or summing over all permutations. To ensure that these associations remain consistent across domains, we enforce view-invariant ranking by aligning the PL likelihoods from the original and style-transferred views through divergence minimization:

$$\mathcal{L}_{\text{rank}} = \frac{1}{N_{t,\text{unk}}} \sum_{i=1}^{N_{t,\text{unk}}} \ell_{\text{KL}}\big(P(\cdot \mid \kappa_i),\, P(\cdot \mid \widehat{\kappa}_i)\big), \tag{12}$$

where $N_{t,\text{unk}}$ denotes the number of unknown-category samples, and $\ell_{\text{KL}}$ is the Kullback–Leibler divergence. Crucially, the KL divergence between two PL distributions also decomposes into a sum over the corresponding local log-probability terms, meaning that our implementation optimizes the exact computable KL induced by the factorized PL model rather than an approximation over the full permutation space. By enforcing agreement between the two factorized PL likelihoods, the model preserves the *global relative ranking* between unknown samples and known prototypes, providing a strong structure-aware regularization signal for category discovery under domain shift.

### 3.5 Online Adaptation

Following the setup of (Park et al., 2024), we assign labels to known and novel samples with different strategies. For data from known categories, we employ the SAM module to generate reliable pseudo-labels. For previously unseen categories, i.e., samples not belonging to any known class, we adopt Affinity Propagation (Frey & Dueck, 2007) to automatically infer cluster memberships. As a non-parametric clustering algorithm, Affinity Propagation iteratively exchanges messages between samples based on pairwise similarities, thereby estimating the optimal number of clusters without requiring it as a prior, which is particularly suitable for open-world scenarios where the number of novel classes is unknown. The inferred clusters are then used to dynamically expand the online classifier, enabling the integration of emerging categories. During online learning, we combine pseudo-labeled known samples with clustered novel samples and incrementally update the model using a standard cross-entropy loss, allowing the system to acquire new semantic knowledge without revisiting past data. The overall optimization objective of our method can be formulated as: $\mathcal{L}_{\text{total}} = \mathcal{L}_{\text{ce}} + \lambda_1 \mathcal{L}_{\text{rank}}$, where $\mathcal{L}_{\text{ce}}$ denotes the cross-entropy loss computed on pseudo-labeled data and $\lambda_1$ is a balancing hyperparameter.

## 4 Experiments

### 4.1 Experimental Setup

**Dataset** We evaluate our method on two representative benchmarks: the Corrupted Semantic Shift Benchmark (SSB-C) (Wang et al., 2024a) and DomainNet (Peng et al., 2019). SSB-C extends the Semantic Shift Benchmark (SSB) with nine corruption types at five severity levels, covering three fine-grained datasets. This benchmark provides a challenging platform to assess robustness under both semantic and visual perturbations. DomainNet is a large-scale dataset with six diverse domains, featuring hundreds of categories and substantial domain gaps. Following (Wang et al., 2024a), we use the original datasets in SSB-C as known domains and their corrupted versions as

unknown domains. For DomainNet, the *Real* domain serves as the known domain, while each of the remaining domains is treated as an unknown domain in turn; we also evaluate a mixed setting where all non-*Real* domains are merged into one unknown domain (details in the Appendix). The category split follows (Cendra et al., 2024): a subset of labeled known classes from the known domain is used in the base session, and subsequent sessions sequentially introduce new streams containing both known and novel categories. Importantly, each session includes samples from both known and unknown domains, simulating realistic scenarios with simultaneous category expansion and domain shift.

**Implementation details**   We adopt ViT-B/16 (Dosovitskiy et al., 2020) as the backbone, pretrained with DINO (Caron et al., 2021; Oquab et al., 2023). Following prior work (Wen et al., 2023; Park et al., 2024), only the final transformer block is fine-tuned at each stage using SGD for 30 epochs with a batch size of 128. The initial learning rate is 0.1 and decayed to $1 \times 10^{-4}$ via cosine annealing, and weight decay is fixed at $5 \times 10^{-5}$. We set the trade-off parameter $\lambda_1 = 1$, the number of stages $T = 3$, and the binary mask ratio $r = 0.3$. The proximal strength parameter $\varepsilon$ in the SAM module is fixed to 0.5. All experiments are repeated with three random seeds, and averaged results are reported. Models are implemented in PyTorch and trained on eight NVIDIA RTX 4090 GPUs.

**Evaluation protocol.**   We adopt continual clustering accuracy (cACC) (Cendra et al., 2024) as our primary evaluation metric. cACC measures the average clustering performance over all sessions up to stage $t$, defined as: $\text{cACC}_t = \frac{1}{t}\sum_{k=1}^{t} \text{ACC}_k$. Here, $\text{ACC}_k$ denotes the clustering accuracy on the test dataset of session $k$. Following (Wang et al., 2024a; Vaze et al., 2022), clustering accuracy (ACC) is defined by comparing the ground-truth labels $y_i$ with the predicted cluster assignments $\hat{y}_i$: $\text{ACC} = \frac{1}{|\mathcal{D}_t^u|}\sum_{i=1}^{|\mathcal{D}_t^u|} \mathbb{I}\{y_i = g^*(\hat{y}_i)\}$, where $g^*$ denotes the optimal permutation mapping predicted clusters to their ground-truth counterparts. We report cACC results on both known and unknown domains, and further break them down into *All*, *Old*, and *New* categories for a comprehensive evaluation.

Table 1: Clustering performance on DomainNet benchmark. We use Real as the known domain and each of the remaining domains as the unknown domain. We report the average All / Old / New accuracy across all stages for both domains.

| Methods | Real → Painting | | | | | | Real → Sketch | | | | | | Real → Quickdraw | | | | | | Real → Clipart | | | | | | Real → Infograph | | | | | |
|---|---|---|---|---|---|---|---|---|---|---|---|---|---|---|---|---|---|---|---|---|---|---|---|---|---|---|---|---|---|---|---|
| | Real | | | Painting | | | Real | | | Sketch | | | Real | | | Quickdraw | | | Real | | | Clipart | | | Real | | | Infograph | | |
| | All | Old | New | All | Old | New | All | Old | New | All | Old | New | All | Old | New | All | Old | New | All | Old | New | All | Old | New | All | Old | New | All | Old | New |
| GCD | 51.3 | 67.2 | 45.4 | 27.4 | 26.7 | 28.1 | 52.3 | 65.7 | 41.7 | 9.2 | 14.5 | 10.1 | 38.7 | 56.2 | 29.6 | 5.0 | 4.7 | 5.8 | 46.7 | 65.7 | 40.1 | 14.5 | 21.2 | 10.1 | 39.8 | 55.3 | 32.4 | 8.1 | 9.8 | 6.4 |
| SimGCD | 48.4 | 63.9 | 41.3 | 22.6 | 22.4 | 23.5 | 48.5 | 60.2 | 36.5 | 7.2 | 11.3 | 9.2 | 32.4 | 50.3 | 23.5 | 4.2 | 4.0 | 5.1 | 40.2 | 58.8 | 33.5 | 10.3 | 18.8 | 8.2 | 33.6 | 49.2 | 27.8 | 6.7 | 7.8 | 5.2 |
| SPTNet | 49.8 | 64.5 | 42.5 | 24.1 | 23.5 | 24.3 | 49.9 | 62.3 | 37.8 | 7.9 | 11.7 | 9.6 | 34.8 | 52.6 | 24.8 | 4.9 | 4.6 | 5.5 | 43.1 | 60.3 | 35.9 | 11.6 | 19.3 | 8.9 | 35.9 | 51.4 | 29.8 | 7.2 | 8.0 | 5.9 |
| RLCD | 50.8 | 66.2 | 44.1 | 25.5 | 24.6 | 25.8 | 51.2 | 64.8 | 40.1 | 8.4 | 12.1 | 10.0 | 36.1 | 54.0 | 25.7 | 4.8 | 4.7 | 5.3 | 45.2 | 62.1 | 36.9 | 13.5 | 20.9 | 9.8 | 37.1 | 53.2 | 32.5 | 8.4 | 8.9 | 6.8 |
| G&M | 47.1 | 62.3 | 41.2 | 26.3 | 25.5 | 26.2 | 50.9 | 63.4 | 42.3 | 10.9 | 15.1 | 10.5 | 34.1 | 50.2 | 27.3 | 4.3 | 4.1 | 5.2 | 40.3 | 61.1 | 34.2 | 11.4 | 19.2 | 8.8 | 32.4 | 50.1 | 27.6 | 7.5 | 9.2 | 5.5 |
| Happy | 50.6 | 66.5 | 44.7 | 28.0 | 27.1 | 28.9 | 52.0 | 65.0 | 41.2 | 11.2 | 15.6 | 10.7 | 35.6 | 51.4 | 28.9 | 4.6 | 4.5 | 5.2 | 45.6 | 62.4 | 37.1 | 12.0 | 19.6 | 9.0 | 34.2 | 50.5 | 28.0 | 7.9 | 9.4 | 5.6 |
| PA-CGCD | 55.4 | 70.3 | 48.1 | 30.1 | 30.8 | 30.2 | 55.1 | 70.7 | 46.6 | 12.3 | 16.1 | 11.2 | 43.6 | 60.4 | 34.2 | 5.1 | 5.0 | 6.0 | 52.2 | 70.3 | 44.6 | 17.8 | 24.5 | 12.3 | 45.2 | 61.3 | 38.1 | 9.0 | 11.8 | 7.1 |
| DEAN | 56.0 | 71.7 | 47.9 | 32.8 | 34.4 | 31.5 | 56.7 | 71.5 | 47.6 | 12.9 | 16.8 | 11.2 | 44.0 | 61.0 | 35.1 | 5.3 | 5.1 | 6.2 | 55.1 | 72.7 | 47.5 | 20.3 | 26.7 | 15.0 | 46.7 | 62.3 | 40.8 | 9.5 | 12.5 | 7.9 |
| PromptCCD | 56.5 | 71.2 | 50.3 | 31.5 | 32.1 | 31.2 | 57.4 | 73.6 | 48.6 | 13.4 | 17.7 | 12.1 | 45.2 | 62.3 | 36.7 | 5.8 | 5.1 | 6.5 | 54.1 | 71.2 | 46.7 | 19.8 | 26.1 | 14.4 | 47.1 | 63.1 | 40.2 | 9.2 | 12.2 | 7.8 |
| VB-CGCD | 57.3 | 71.0 | 52.4 | 32.4 | 33.6 | 32.5 | 56.9 | 73.1 | 48.8 | 13.9 | 18.1 | 12.9 | 47.1 | 62.1 | 38.1 | 6.0 | 4.9 | 6.8 | 55.4 | 72.0 | 47.5 | 19.6 | 25.8 | 14.2 | 48.3 | 63.9 | 41.9 | 9.4 | 12.4 | 8.0 |
| **PRISM** | **60.9** | **74.1** | **55.1** | **39.2** | **39.0** | **38.2** | **60.1** | **73.4** | **51.0** | **16.9** | **20.1** | **15.9** | **54.0** | **74.0** | **49.2** | **7.1** | **6.5** | **7.4** | **58.0** | **72.3** | **51.2** | **24.0** | **30.4** | **19.1** | **60.1** | **73.8** | **53.1** | **10.9** | **14.1** | **9.8** |

Table 2: Clustering performance on SSB-C benchmarks. Each dataset contains both Original and Corrupted settings, and we report the average All / Old / New accuracy across all stages for both domains.

| Methods | CUB-C | | | | | | Stanford Cars-C | | | | | | FGVC-Aircraft-C | | | | | |
|---|---|---|---|---|---|---|---|---|---|---|---|---|---|---|---|---|---|---|
| | Original | | | Corrupted | | | Original | | | Corrupted | | | Original | | | Corrupted | | |
| | All | Old | New | All | Old | New | All | Old | New | All | Old | New | All | Old | New | All | Old | New |
| GCD | 29.4 | 47.7 | 23.4 | 26.8 | 45.9 | 20.1 | 26.4 | 56.1 | 21.5 | 22.3 | 43.1 | 11.2 | 27.7 | 33.6 | 24.9 | 28.8 | 41.4 | 28.8 |
| SimGCD | 26.6 | 44.5 | 21.0 | 23.4 | 42.4 | 17.7 | 23.1 | 52.5 | 18.9 | 19.3 | 39.7 | 9.8 | 25.4 | 30.1 | 22.1 | 25.2 | 38.1 | 25.8 |
| SPTNet | 27.8 | 45.2 | 22.0 | 25.1 | 44.2 | 18.1 | 24.9 | 55.0 | 20.3 | 21.1 | 41.6 | 9.9 | 26.1 | 31.2 | 23.3 | 26.9 | 39.5 | 26.7 |
| RLCD | 29.1 | 46.8 | 23.8 | 26.2 | 45.3 | 19.4 | 26.8 | 56.9 | 22.1 | 22.9 | 43.2 | 9.7 | 27.8 | 32.3 | 24.2 | 27.3 | 40.7 | 28.1 |
| G&M | 16.4 | 34.1 | 10.5 | 13.7 | 32.1 | 7.7 | 15.7 | 43.8 | 12.3 | 11.4 | 30.5 | 6.7 | 20.5 | 24.8 | 17.9 | 21.6 | 32.7 | 22.3 |
| Happy | 22.0 | 39.4 | 16.9 | 19.8 | 38.4 | 14.2 | 21.9 | 48.7 | 18.9 | 18.1 | 37.0 | 13.2 | 24.3 | 27.9 | 21.3 | 24.8 | 35.6 | 25.7 |
| PA-CGCD | 28.3 | 46.5 | 22.7 | 25.4 | 44.7 | 18.4 | 25.2 | 55.1 | 20.9 | 21.2 | 41.5 | 10.2 | 27.8 | 40.1 | 23.7 | 27.8 | 40.1 | 27.2 |
| DEAN | 28.9 | 47.1 | 23.0 | 26.3 | 46.2 | 18.2 | 26.1 | 58.1 | 19.4 | 22.1 | 41.2 | 12.9 | 28.1 | 32.8 | 28.9 | 29.1 | 40.1 | 30.3 |
| PromptCCD | 30.1 | 48.1 | 24.5 | 27.4 | 46.1 | 20.3 | 27.4 | 57.4 | 22.1 | 23.1 | 44.4 | 11.4 | 29.9 | 34.5 | 26.4 | 30.3 | 42.9 | 29.9 |
| VB-CGCD | 34.2 | 51.8 | 26.3 | 31.7 | 49.2 | 23.4 | 31.6 | 59.9 | 26.1 | 26.3 | 47.9 | 15.1 | 33.2 | 37.3 | 29.7 | 32.3 | 44.5 | 31.6 |
| **PRISM** | **49.3** | **64.9** | **44.2** | **44.0** | **60.9** | **37.0** | **36.9** | **60.0** | **29.1** | **33.3** | **56.5** | **23.5** | **40.1** | **48.9** | **40.1** | **36.4** | **46.1** | **34.1** |

Table 3: Component-wise ablation on **Real →
Painting**.

| Components | | | Real | | | Painting | | |
|---|---|---|---|---|---|---|---|---|
| HCS | SAM | IKT | All | Old | New | All | Old | New |
| ✗ | ✗ | ✗ | 54.6 | 68.7 | 46.5 | 28.7 | 28.1 | 27.9 |
| ✓ | ✓ | ✗ | 58.1 | 72.9 | 49.9 | 35.0 | 35.9 | 32.5 |
| ✓ | ✗ | ✓ | 56.9 | 70.2 | 52.7 | 33.2 | 31.8 | 35.2 |
| ✓ | ✓ | ✓ | **60.9** | **74.1** | **55.1** | **39.2** | **39.0** | **38.2** |

Table 4: Comparison of separation strategies on
**Real → Painting**.

| Methods | Real | | | Painting | | |
|---|---|---|---|---|---|---|
| | All | Old | New | All | Old | New |
| origin image | 55.0 | 68.7 | 47.2 | 29.6 | 28.9 | 28.3 |
| entropy-based | 54.4 | 69.0 | 46.7 | 29.9 | 29.1 | 28.6 |
| energy-based | 55.8 | 69.9 | 48.1 | 30.6 | 29.5 | 29.9 |
| **PRISM** | **60.9** | **74.1** | **55.1** | **39.2** | **39.0** | **38.2** |

## 4.2 Main results

We compare our method with representative continual discovery baselines, including Grow & Merge (G&M) (Zhang et al., 2022), Happy (Ma et al., 2024), PA-CGCD (Kim et al., 2023), DEAN (Park et al., 2024), PromptCCD (Cendra et al., 2024), and VB-CGCD (Dai & Chauhan, 2025), as well as re-implemented GCD methods (GCD (Vaze et al., 2022), SimGCD (Wen et al., 2023), SPT-Net (Wang et al., 2024b) and RLCD (Liu et al., 2025a)). We also note that some recent works, such as HiLo (Wang et al., 2024a) and CDAD-Net (Rongali et al., 2024), have explored handling distribution shifts in GCD. However, since these methods require access to the entire dataset rather than session-based streams, they cannot be directly applied to CCD, and are therefore not included in our comparison. Table 1 and Table 2 present the results on the SSB-C and DomainNet benchmarks, respectively. It can be observed that in the challenging OW-CCD setting, existing GCD and CCD approaches struggle to cope with domain shifts, leading to unreliable recognition of known classes and poor discovery of new ones. In contrast, our approach consistently achieves more robust clustering performance, outperforming both prior CCD and GCD methods by a clear margin. For instance, on CUB-C, our method surpasses the strongest CCD competitor, VB-CGCD, by 15.1% in the clean domain and 12.3% in the corrupted domain, highlighting its robustness against both semantic and visual perturbations. On the more demanding DomainNet benchmark, similar gains are observed. For instance, in the Real→Painting task, PRISM outperforms VB-CGCD by 3.6% on the source domain (Real) and 6.8% on the target domain (Painting). These results highlight that our approach generalizes effectively to new domains while reliably discovering novel categories in continuous streams.

## 4.3 Analysis

**Effectiveness of different components.** We conduct a comprehensive ablation study to examine the contribution of each component in our framework. As shown in Table 3, the baseline performs poorly, highlighting the severe impact of domain shifts on both known and novel category recognition. Incorporating the HCS module to separate known from unknown samples, followed by the SAM module, substantially improves clustering accuracy on known categories, confirming the effectiveness of sparse assignment matching. Introducing the IKT module further enhances the discovery of novel categories, underscoring the importance of preserving robust category associations under distribution shifts. When all components are integrated, the model achieves the best overall performance, demonstrating the benefit of combining these modules for reliable open-world continual category discovery.

**Comparison with alternative separation modules.** To further validate the contribution of the HCS module, we carried out a focused ablation study. We compared with three baselines: (1) an entropy-driven separation scheme (Safaei et al., 2024), (2) an energy-based approach (Park et al., 2024), and (3) a simplified variant of HCS that relies on raw image features without applying frequency decomposition. As reported in Table 4, the proposed module consistently outperforms these alternatives. Its strength lies in exploiting high-frequency information, which preserves more detailed structural and semantic patterns, allowing the model to more effectively separate unlabeled data. This leads to a more reliable basis for recognizing both previously seen and emerging categories in continual discovery. In addition, Figure 5 in Appendix provides a qualitative illustration of this effect. The HCS module provides a clearer separation between known and unknown groups, demonstrating its ability to filter out style-related noise while retaining meaningful semantic relations. These observations collectively indicate that HCS is not only beneficial for sample separation

but also crucial for enhancing overall performance in open-world category discovery under distribution shifts.

## 5 CONCLUSION

In this work, we take the first step toward tackling the challenging problem of open-world continual category discovery and introduce three key innovations to address it. First, a high-frequency-driven category separation module leverages spectral details to reliably distinguish between known and novel categories. Second, a sparse assignment matching module employs proximal optimal transport to assign trustworthy clustering labels to known classes. Third, an invariant knowledge transfer module enforces semantic association consistency across domains, enabling robust knowledge transfer under distributional shifts. Extensive experiments on multiple benchmarks validate the effectiveness of our framework, demonstrating its ability to consistently recognize known categories and uncover new ones in dynamic, non-stationary data streams.

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

# A APPENDIX

## A.1 MORE RELATED WORK

### A.1.1 CATEGORY DISCOVERY

Category discovery aims to transfer knowledge from known classes to identify novel concepts, where unlabeled data may contain unseen categories. Novel Class Discovery (NCD) was first introduced to explore how knowledge from labeled classes can be leveraged to discover entirely new ones. Early solutions followed a two-stage strategy. For example, AutoNovel (Han et al., 2021) employs self-supervised learning with ranking statistics to transfer knowledge for clustering. Subsequently, Fini et al. (2021) proposed a unified end-to-end framework optimizing multiple objectives simultaneously. IIC (Li et al., 2023a) further model inter-class separability and intra-class consistency to improve robustness. Feng et al. (2023a) leverage uncertainty-aware multi-view pseudo-supervision and neighborhood aggregation to refine pseudo-labels for novel skin disease discovery. While NCD assumes that all unlabeled data belong to novel categories, this assumption limits its practicality. To address more realistic scenarios, Generalized Category Discovery (GCD) was introduced, where the unlabeled pool contains both previously seen and unseen categories. Early GCD methods combined supervised contrastive objectives with self-supervised representation learning followed by semi-supervised clustering (Vaze et al., 2022). Later, SimGCD (Wen et al., 2023) introduced a parametric classifier to improve efficiency and inference speed, establishing a strong baseline. Building on these foundations, researchers have proposed a series of more advanced approaches to tackle different challenges in GCD. For instance, Feng et al. (2025b) introduced NGUF, a neighbor-guided unbiased framework that leverages nearest-neighbor information to mitigate seen-class prediction bias in medical GCD. Liu et al. (2025b) explored hierarchical space modeling,

arguing that Euclidean or spherical spaces are suboptimal for encoding data with hierarchical structures, and instead proposed a hyperbolic embedding space to better capture both seen and unseen categories. To unify the treatment of old and new classes, Ma et al. (2025b) developed ProtoGCD, which leverages joint prototypes and dual-level pseudo-labeling to balance the recognition of known and novel categories while also estimating the number of unseen classes. Beyond the single-domain setting, Wang et al. (2024a) and Feng & Ge (2025) extended GCD into cross-domain scenarios, addressing domain shift through representation alignment and frequency-domain separation with specialized augmentation and spectral perturbation strategies. In parallel, multimodal extensions have also been explored: Zheng et al. (2024) proposed TextGCD, a two-phase framework that generates descriptive texts via retrieval and employs cross-modality co-teaching, while $M^3$GCD (Feng et al., 2026) introduces a medical multimodal GCD framework that leverages image–text pairs, dynamic expert fusion, and category diffusion for known and novel disease category discovery. Concurrently, recent works have begun to rethink the methodological foundations of GCD: Tang et al. (2025a) proposes a novel multiplex consensus framework via self-deconstruction, while Tang et al. pioneers a manifold-geometric formulation through Bures-isotropy alignment. From a broader open-world perspective, Tang et al. (2025b) explores object-concept-relation reasoning to enhance foundation models. Together, these works significantly improve the balance between known and novel classes, and continually push the performance boundaries of GCD across both generic and fine-grained datasets. However, these approaches remain limited to static GCD datasets, whereas our work focuses on tackling domain shift under continuous streaming data, a setting that more faithfully reflects real-world dynamics.

Going further, Continuous Category Discovery (CCD) extends GCD to an incremental setting, where models continually receive new streams of unlabeled data. The key challenge lies in discovering new categories while retaining knowledge of past ones. Recent progress in CCD has introduced diverse strategies to alleviate forgetting and improve discovery quality. Zhang et al. (2022) presented the Grow-and-Merge framework, which alternates between a growth phase for enriching feature diversity via self-supervised learning and a merging phase that stabilizes recognition of previously learned classes. Wu et al. (2023) proposed a meta-learning optimization approach that balances class-discriminative representations for known categories with diverse features for novel discovery. Park et al. (2024) designed DEAN, an online method that performs discovery through energy-based guidance and enhances reliability using variance-driven feature augmentation. Cendra et al. (2024) introduced PromptCCD, where Gaussian Mixture Prompting acts as a dynamic pool that prevents forgetting and enables adaptive estimation of category numbers. Dai & Chauhan (2025) developed VB-CGCD, which explains forgetting as covariance misalignment and employs variational Bayesian inference with covariance-aware classification to improve robustness under noisy pseudo-labels.

While these advances move CCD closer to practical continual learning, most methods still rely on the simplifying assumption of a fixed domain within each stage. In reality, streaming data often involve domain variations or shifts, making such assumptions unrealistic and motivating new frameworks that explicitly address multi-domain continual discovery.

### A.1.2 DOMAIN ADAPTATION

Domain adaptation seeks to mitigate distribution shifts between a labeled source and a target domain. A key setting is unsupervised domain adaptation (UDA), which leverages labeled source data and unlabeled target data for model adaptation. UDA methods typically learn domain-invariant representations to reduce distribution shifts. Discrepancy-based approaches minimize statistical differences between domains via moment-matching techniques (e.g., correlation alignment (Sun & Saenko, 2016) or Maximum Mean Discrepancy (Long et al., 2015; Tzeng et al., 2014)), while adversarial approaches (Saito et al., 2018a; Sankaranarayanan et al., 2018) employ domain discriminators to encourage indistinguishable cross-domain features. Recently, Transformer-based backbones (Dosovitskiy et al., 2020) have been explored to enhance feature alignment through attention mechanisms (Sun et al., 2022; Xu et al., 2021). However, most UDA methods assume joint access to source and target data, which is impractical under privacy constraints. Source-Free Domain Adaptation (SFDA) addresses this by adapting only a source-trained model with unlabeled target data. Chidlovskii et al. (2016) suggested using a small set of prototypes instead of the complete source data to facilitate adaptation, while Liang et al. (2019) enhanced target learning by iteratively refining pseudo-labels through self-training. SHOT (Krause et al., 2010; Shi & Sha, 2012) transfers the source-trained encoder to the target domain by combining information maximization with clustering,

keeping the classifier unchanged. To further improve pseudo-label reliability, Yang et al. (2021) introduced neighborhood consistency regularization across target samples. Beyond these transductive settings, researchers have also examined Open-Set Domain Adaptation (OSDA), where target data may involve categories unseen in the source. OSBP (Saito et al., 2018b) introduced a thresholding strategy to separate unknown samples from the known target subset, while STA (Liu et al., 2019) proposed a progressive weighting scheme to gradually disentangle them. More recently, ANNA (Li et al., 2023b) incorporated causal front-door adjustment and decoupled alignment to mitigate semantic bias and enable more reliable transfer under open-set conditions. Although OSDA broadens the applicability of domain adaptation, it still mainly focuses on recognizing known categories, while overlooking further exploration of the unknown category space. Orthogonal to such input- and feature-level alignment, a recent line of work analyzes the spectral structure of network weights themselves to diagnose layer-wise training quality (Hu et al., 2025) and to balance fine-tuning when supervised data is scarce (Liu et al., 2024)—signals that become particularly relevant when newly emerging categories in a continual stream initially carry only a handful of labeled samples.

## A.2 THEORETICAL PROOF

### A.2.1 OPTIMIZATION OF SAM

In this section, we elaborate on the optimization procedure for solving the Sparse Assignment Matching (SAM) objective. Let $\psi$ and $\varphi$ denote the dual variables. The SAM problem can then be formulated as:

$$\max_{\psi,\varphi} \mathcal{L}_S = \sum_{i=1}^{N_{t,\mathrm{kno}}} \psi_i \hat{a}_i + \sum_{j=1}^{\mathcal{C}^{t-1}} \varphi_j \hat{b}_j - \frac{\varepsilon}{2} \sum_{i=1}^{N_{t,\mathrm{kno}}} \sum_{j=1}^{\mathcal{C}^{t-1}} \left[ \frac{\psi_i + \varphi_j - \tilde{C}_{ij}}{\varepsilon} \right]_+^2, \tag{13}$$

where $N_{t,\mathrm{kno}}$ denotes the number of known samples, $\mathcal{C}^{t-1}$ the number of known category prototypes, $\tilde{C}_{ij} = C_{ij} - \varepsilon \gamma_{ij}^{(l)}$ is the transport cost, $\hat{a}_i$ and $\hat{b}_j$ are the corresponding marginals. To efficiently optimize Eq. equation 13, we adopt the Block Coordinate Descent (BCD) strategy. The updates of the dual variables are derived by alternatingly fixing one variable and optimizing the other.

**Update of $\psi$.** Taking the derivative of $\mathcal{L}_S$ with respect to $\psi_i$ and setting it to zero yields:

$$\Psi(\psi_i) = \sum_{j=1}^{\mathcal{C}^{t-1}} \left[ \psi_i - \left( \tilde{C}_{ij} - \varphi_j \right) \right]_+ = \varepsilon \hat{a}_i. \tag{14}$$

**Update of $\varphi$.** Similarly, for $\varphi_j$, we have:

$$\Phi(\varphi_j) = \sum_{i=1}^{N_{t,\mathrm{kno}}} \left[ \varphi_j - \left( \tilde{C}_{ij} - \psi_i \right) \right]_+ = \varepsilon \hat{b}_j. \tag{15}$$

**Update of $\gamma$.** With the updated dual variables, the primal transport plan $\gamma$ can be updated. At the $l$-th iteration, the optimal $\gamma^{(l+1)}$ is obtained as:

$$\begin{aligned} \gamma_{ij}^{(l+1)} &= \max\left( 0, \frac{\psi_i^{(l)} + \varphi_j^{(l)} + \varepsilon \gamma_{ij}^{(l)} - C_{ij}}{\varepsilon} \right), \\ \tilde{C}_{ij}^{(l+1)} &= C_{ij} - \varepsilon \gamma_{ij}^{(l)}. \end{aligned} \tag{16}$$

After several iterations, the optimal solutions of $\psi$ and $\varphi$ are obtained, based on which the corresponding optimal transport plan $\gamma$ can be subsequently derived.

### A.2.2 THEORETICAL INTUITION ON DOMAIN INVARIANCE OF HIGH-FREQUENCY CUES

Given an input image $x^{(d)} \in \mathbb{R}^{H \times W \times C}$ from domain $d \in \{s, t\}$, we apply the discrete Fourier transform (DFT) $\mathcal{F}(\cdot)$ and its inverse $\mathcal{F}^{-1}(\cdot)$. A binary mask $M \in \mathbb{R}^{r \times r}$ is constructed to separate

Table 5: Class counts at each incremental stage for the Corrupted SSB and DomainNet benchmarks. We present the cumulative number of categories in both **Original** and **Corrupted** settings over four stages.

| Stage | CUB-C | | Stanford Cars-C | | FGVC-Aircraft-C | | DomainNet | |
|---|---|---|---|---|---|---|---|---|
| | Original | Corrupted | Original | Corrupted | Original | Corrupted | Real | Other Domains |
| 0 | 140 | N/A | 130 | N/A | 70 | N/A | 225 | N/A |
| 1 | 160 | 160 | 152 | 152 | 80 | 80 | 265 | 265 |
| 2 | 180 | 180 | 174 | 174 | 90 | 90 | 305 | 305 |
| 3 | 200 | 200 | 196 | 196 | 100 | 100 | 345 | 345 |

Table 6: Overview of class partitions in the labeled dataset ($\mathcal{D}^l$) and unlabeled streams ($\mathcal{D}_1^u$, $\mathcal{D}_2^u$, $\mathcal{D}_3^u$), covering both the known and unknown domains.

| Class Range | Known Domain | | | | Unknown Domain | | | |
|---|---|---|---|---|---|---|---|---|
| | $\mathcal{D}^l$ | $\mathcal{D}_1^u$ | $\mathcal{D}_2^u$ | $\mathcal{D}_3^u$ | $\mathcal{D}^l$ | $\mathcal{D}_1^u$ | $\mathcal{D}_2^u$ | $\mathcal{D}_3^u$ |
| $y_i \in [1, 0.7|\mathcal{C}|]$ | 87% | 7% | 3% | 3% | 0% | 7% | 3% | 3% |
| $y_i \in (0.7|\mathcal{C}|, 0.8|\mathcal{C}|]$ | 0% | 70% | 20% | 10% | 0% | 70% | 20% | 10% |
| $y_i \in (0.8|\mathcal{C}|, 0.9|\mathcal{C}|]$ | 0% | 0% | 90% | 10% | 0% | 0% | 90% | 10% |
| $y_i \in (0.9|\mathcal{C}|, |\mathcal{C}|]$ | 0% | 0% | 0% | 100% | 0% | 0% | 0% | 100% |

low- and high-frequency components:

$$M_{u,v} = \begin{cases} 1, & \text{if } \max(|u - \frac{H}{2}|, |v - \frac{W}{2}|) \leq r \cdot \frac{\min(H,W)}{2}, \\ 0, & \text{otherwise}, \end{cases} \tag{17}$$

and we define

$$\mathcal{F}^l(x) = M \odot \mathcal{F}(x), \quad \mathcal{F}^h(x) = (I - M) \odot \mathcal{F}(x), \tag{18}$$

where $\odot$ denotes element-wise multiplication. The corresponding spatial components are obtained by

$$x^l = \mathcal{F}^{-1}(\mathcal{F}^l(x)), \quad x^h = \mathcal{F}^{-1}(\mathcal{F}^h(x)). \tag{19}$$

To analyze the domain dependence of different frequency bands, we assume a simple additive decomposition:

$$x^{(d)} = u + v^{(d)}, \tag{20}$$

where $u$ denotes the domain-shared semantic structure (edges, textures, shapes), and $v^{(d)}$ represents the domain-specific style (illumination, color tone, or imaging pipeline). In the frequency domain, this becomes

$$\mathcal{F}(x^{(d)}) = \mathcal{F}(u) + \mathcal{F}(v^{(d)}). \tag{21}$$

*Step 1: High-frequency discrepancy is upper-bounded by the high-frequency tail of style.* For the high-pass band $\Omega_h(r)$ selected by $(I - M)$, we have

$$\|\mathcal{F}^h(x^{(s)}) - \mathcal{F}^h(x^{(t)})\|_2 = \|(I - M) \odot (\mathcal{F}(v^{(s)}) - \mathcal{F}(v^{(t)}))\|_2 \leq \|(I - M) \odot \mathcal{F}(v^{(s)})\|_2 + \|(I - M) \odot \mathcal{F}(v^{(t)})\|_2. \tag{22}$$

Since each domain style $v^{(d)}$ is $C^m$-smooth ($m \geq 1$) with bounded Sobolev norm $\|v^{(d)}\|_{H^m} \leq B$, then the Fourier energy of its high-frequency tail decays as

$$\int_{\|\omega\| > \rho(r)} |\mathcal{F}(v^{(d)})(\omega)|^2 \, d\omega \leq C_m \, \rho(r)^{-2(m-1)} B^2, \tag{23}$$

which implies

$$\|(I - M) \odot \mathcal{F}(v^{(d)})\|_2 \leq C_m^{1/2} \rho(r)^{-(m-1)} B. \tag{24}$$

Substituting into Eq. equation 22, we obtain

$$\|\mathcal{F}^h(x^{(s)}) - \mathcal{F}^h(x^{(t)})\|_2 \leq 2C_m^{1/2} \rho(r)^{-(m-1)} B \equiv \varepsilon(r). \tag{25}$$

As the cutoff frequency $\rho(r)$ increases, $\varepsilon(r) \to 0$, which means the inter-domain difference in the high-frequency band becomes negligible, and the high-frequency representation is effectively determined by the shared semantics $u$.

*Step 2: Low-frequency discrepancy is dominated by style.* For the low-pass band $\Omega_l(r)$, we have

$$\|\mathcal{F}^l(x^{(\mathrm{s})}) - \mathcal{F}^l(x^{(\mathrm{t})})\|_2 = \|M \odot (\mathcal{F}(u) - \mathcal{F}(u) + \mathcal{F}(v^{(\mathrm{s})}) - \mathcal{F}(v^{(\mathrm{t})}))\|_2 = \|M \odot (\mathcal{F}(v^{(\mathrm{s})}) - \mathcal{F}(v^{(\mathrm{t})}))\|_2. \tag{26}$$

Since $\mathcal{F}(v^{(d)})$ concentrates energy near the origin, the right-hand side is non-negligible across domains, showing that low-frequency spectra encode style and illumination variations.

*Step 3: Physical imaging models reinforce this separation.* In practice, cross-domain shifts often arise from: (i) multiplicative/additive low-frequency fields

$$x^{(d)}(p) = a^{(d)}(p)x_{\mathrm{phys}}(p) + b^{(d)}(p), \tag{27}$$

where $a^{(d)}$ and $b^{(d)}$ are slowly varying and thus mainly perturb the low-frequency spectrum; and (ii) convolution with smooth kernels $k^{(d)}$, whose transfer functions $K^{(d)}(\omega)$ are low-pass, further attenuating style at high frequency. Both mechanisms reduce the high-frequency contribution of $v^{(d)}$ and thus tighten the bound $\varepsilon(r)$ above.

*Conclusion.* Combining the above derivations and the Fourier decay property yields

$$\|\mathcal{F}^h(x^{(\mathrm{s})}) - \mathcal{F}^h(x^{(\mathrm{t})})\|_2 \le \varepsilon(r) \to 0, \qquad \|\mathcal{F}^l(x^{(\mathrm{s})}) - \mathcal{F}^l(x^{(\mathrm{t})})\|_2 \to \text{remains significant} \tag{28}$$

Therefore, high-frequency components $(I - M) \odot \mathcal{F}(x)$ encode domain-invariant semantic structures, while low-frequency components $M \odot \mathcal{F}(x)$ capture domain-specific styles. This theoretical analysis explains why the high-frequency cues extracted in Eq.(3) are inherently more robust and domain-invariant in practice.

### A.3 PSEUDOCODE

The pseudocode of PRISM, outlining its main components and training flow, is provided in Algorithm 1.

### A.4 DATASETS

To thoroughly evaluate the proposed framework under both domain shift and semantic shift conditions, we conduct experiments on two widely used benchmarks: **DomainNet** (Peng et al., 2019) and **SSB-C** (Wang et al., 2024a). These datasets encompass diverse visual domains and fine-grained recognition challenges, thereby providing a rigorous test of generalization and robustness.

#### A.4.1 DOMAINNET

DomainNet (Peng et al., 2019) is among the largest benchmarks in domain adaptation and generalization, containing approximately 600,000 images across 345 categories. The dataset spans six heterogeneous domains with distinct visual styles: Real (photographic images), Clipart (cartoon-style drawings), Sketch (hand-drawn sketches), Painting (artistic renderings such as oil and watercolor), Infograph (symbolic infographic-like images), and Quickdraw (doodle-style drawings from Google QuickDraw). The large scale and stylistic diversity introduce strong domain discrepancies, making DomainNet a challenging testbed for algorithms aiming to learn domain-invariant yet discriminative representations.

#### A.4.2 SSB-C

The SSB-C benchmark (Wang et al., 2024a) extends the Semantic Shift Benchmark (SSB) to explicitly measure robustness under semantic and distributional perturbations. The original SSB is built from three fine-grained datasets: CUB-200-2011 (200 bird species with subtle inter-class variations), Stanford Cars (196 categories covering a wide range of brands and models), and FGVC-Aircraft (100 aircraft categories defined by structural differences). SSB-C introduces nine corruption types (e.g., Gaussian noise, frost blur, impulse noise) applied at five severity levels, following the common corruption protocol. This produces a dataset that is nearly **45× larger** than the original SSB, offering a comprehensive benchmark for evaluating robustness in fine-grained recognition.

---

**Algorithm 1:** PRISM

---

**Input** : labeled base set $\mathcal{D}^l$; streaming unlabeled sets $\{\mathcal{D}_t^u\}_{t=1}^T$; model $\theta = \{f, g\}$; mask ratio $r$; SAM proximal strength $\varepsilon$; rank loss weight $\lambda_1$
**Output:** updated model $\theta = \{f, g\}$

1  /* --- High-Frequency-Driven Category Separation (HCS) ---        */
2  **Function** HCS_Split($\mathcal{D}_t^u, f, \{e_c^{t-1}\}, r$):
3      **for** $x \in \mathcal{D}_t^u$ **do**
4          Compute Fourier spectrum $\mathcal{F}(x)$ with mask $M$;
5          Extract high-frequency part $x^h$;
6          $S(x) \leftarrow \nu(\max_c \frac{\langle f(x^h), e_c^{t-1} \rangle}{\|f(x^h)\| \|e_c^{t-1}\|})$
7      **end for**
8      Fit 2-comp GMM on $\{S(x)\}$ and get $\pi(x)$;
9      $\mathcal{D}_{t,\text{kno}}^u \leftarrow \{x | \pi(x) \geq 0.5\}, \mathcal{D}_{t,\text{unk}}^u \leftarrow \{x | \pi(x) < 0.5\}$
10     **return** $\mathcal{D}_{t,kno}^u, \mathcal{D}_{t,unk}^u$
11 /* --- Sparse Assignment Matching (SAM) ---        */
12 **Function** SAM_Assign($\mathcal{D}_{t,kno}^u, f, g, \{e_c^{t-1}\}, \varepsilon$):
13     Build cost $C_{ij} = -\log(g(f(x_{i,\text{kno}}))_j)$;
14     Initialize $\gamma_{ij}^{(0)}$;
15     Solve dual $(\psi, \varphi)$ and update $\gamma$ until convergence;
16     $\tilde{y}_i^{\text{kno}} \leftarrow \arg\max_j \gamma_{ij}^*$
17     **return** $\{\tilde{y}_i^{kno}\}, \gamma^*$
18 /* --- Invariant Knowledge Transfer (IKT) ---        */
19 **Function** IKT_RankLoss($\mathcal{D}_{t,unk}^u, f, \{e_c^{t-1}\}$):
20     Estimate low-frequency stats from prev. stage;
21     **for** $x \in \mathcal{D}_{t,unk}^u$ **do**
22         Generate style-perturbed view $\widehat{x}$;
23         $z = f(x), \widehat{z} = f(\widehat{x})$;
24         Compute PL dists $P(\mathcal{P}|\kappa), P(\mathcal{P}|\widehat{\kappa})$;
25         Accumulate $\ell_{KL}$
26     **end for**
27     $\mathcal{L}_{\text{rank}} \leftarrow$ mean divergence
28     **return** $\mathcal{L}_{rank}$
29 /* --- Affinity Propagation + Online Update ---        */
30 **Function** AP_Cluster($\mathcal{D}_{t,unk}^u, f$):
31     Run Affinity Propagation on $\{f(x)\}$;
32     **return** *novel clusters* $\{\hat{y}^{unk}\}, K_{unk}$
33 **Function** Online_Update($\theta, \mathcal{S}_{kno}, \mathcal{S}_{nov}, \mathcal{L}_{rank}, \lambda_1$):
34     $\mathcal{L}_{\text{ce}} \leftarrow$ cross-entropy on pseudo + novel clusters;
35     $\mathcal{L}_{\text{total}} = \mathcal{L}_{\text{ce}} + \lambda_1 \mathcal{L}_{\text{rank}}$;
36     Update model $\theta = \{f, g\}$;
37     **return** $\theta = \{f, g\}$, *updated prototypes* $\{e_c^t\}$
38 /* --- Main Procedure ---        */
39 Initialize $\theta = \{f, g\}$ and get known-class prototypes $\{e_c^0\}$ on $\mathcal{D}^l$
40 **for** $t = 1$ **to** $T$ **do**
41     $\mathcal{D}_{t,\text{kno}}^u, \mathcal{D}_{t,\text{unk}}^u \leftarrow$ HCS_Split($\mathcal{D}_t^u, f, \{e_c^{t-1}\}, r$)
42     $\{\tilde{y}_i^{\text{kno}}\}, \gamma^* \leftarrow$ SAM_Assign($\mathcal{D}_{t,\text{kno}}^u, f, g, \{e_c^{t-1}\}, \varepsilon$)
43     $\mathcal{L}_{\text{rank}} \leftarrow$ IKT_RankLoss($\mathcal{D}_{t,\text{unk}}^u, f, \{e_c^{t-1}\}$)
44     $\{\hat{y}^{\text{unk}}\}, K_{\text{unk}} \leftarrow$ AP_Cluster($\mathcal{D}_{t,\text{unk}}^u, f$)
45     Build pseudo-labeled sets $\mathcal{S}_{\text{kno}}, \mathcal{S}_{\text{unk}}$;
46     $\theta, \{e_c^t\} \leftarrow$ Online_Update($\theta, \mathcal{S}_{\text{kno}}, \mathcal{S}_{\text{unk}}, \mathcal{L}_{\text{rank}}, \lambda_1$)
47 **end for**
48 **return** *updated model* $\theta = \{f, g\}$

---

### A.4.3 EVALUATION PROTOCOL

For each benchmark, a subset of categories is initially designated as labeled known classes to build the first training session. In subsequent sessions, new categories are gradually introduced, simulating the progressive emergence of novel classes. Detailed statistics of category splits are presented in Table 5, while the proportion of known and unknown samples across unseen domains is summarized in Table 6. These staged splits emulate real-world deployment scenarios in which both novel categories and domain shifts arise over time. Methods are evaluated by their ability to simultaneously recognize known classes and discover unknown ones, with particular emphasis on generalization and semantic separability.

Table 7: Clustering results (**mean ± std**) on the DomainNet benchmark. The Real domain is treated as the known domain, while each of the other domains serves in turn as the unknown domain. We present the averaged accuracies on All / Old / New classes across all stages for both domains.

| Methods | Real → Painting | | | | | | Real → Sketch | | | | | | Real → Quickdraw | | | | | | Real → Clipart | | | | | | Real → Infograph | | | | | |
| | Real | | | Painting | | | Real | | | Sketch | | | Real | | | Quickdraw | | | Real | | | Clipart | | | Real | | | Infograph | | |
| | All | Old | New | All | Old | New | All | Old | New | All | Old | New | All | Old | New | All | Old | New | All | Old | New | All | Old | New | All | Old | New | All | Old | New |
| GCD | 51.3±2.2 | 67.2±2.1 | 48.4±1.7 | 27.4±1.4 | 26.7±1.9 | 28.1±1.0 | 52.3±1.0 | 65.7±1.1 | 41.7±1.8 | 9.2±0.8 | 14.5±1.1 | 10.1±0.6 | 38.7±2.0 | 56.2±2.0 | 29.6±1.0 | 5.0±0.5 | 4.7±0.4 | 5.8±0.5 | 46.7±2.0 | 68.7±0.8 | 40.1±1.0 | 14.5±1.9 | 21.2±1.6 | 10.1±1.4 | 39.8±2.3 | 55.3±0.9 | 32.4±1.3 | 8.1±0.3 | 9.8±1.0 | 6.4±0.5 |
| SimGCD | 48.4±1.7 | 63.9±1.3 | 41.3±1.1 | 22.6±0.9 | 22.4±1.3 | 23.5±1.3 | 48.5±1.9 | 60.2±1.2 | 36.5±0.9 | 7.2±0.9 | 11.3±1.5 | 9.2±1.2 | 32.4±1.6 | 50.3±2.1 | 23.5±1.4 | 4.2±0.5 | 4.0±0.3 | 5.1±0.7 | 40.2±1.9 | 58.8±0.9 | 35.5±1.7 | 10.3±1.9 | 18.8±1.4 | 8.2±2.0 | 35.6±1.5 | 49.2±1.4 | 27.8±1.3 | 6.7±0.6 | 7.8±0.2 | 5.2±0.4 |
| SPTNet | 49.8±1.9 | 64.5±2.2 | 42.5±1.2 | 24.1±1.1 | 23.5±1.0 | 24.3±1.5 | 49.9±1.3 | 62.3±1.1 | 37.8±1.3 | 7.9±0.5 | 11.7±1.8 | 9.6±0.7 | 34.6±1.9 | 52.6±1.3 | 24.8±1.3 | 4.9±0.5 | 4.6±0.5 | 5.5±0.9 | 43.1±1.7 | 60.3±1.0 | 35.9±1.3 | 11.6±2.1 | 19.3±1.2 | 8.9±1.9 | 38.9±1.2 | 51.4±1.1 | 29.8±1.0 | 7.2±0.9 | 8.0±0.5 | 5.9±0.3 |
| RLCD | 50.8±1.5 | 66.2±2.2 | 44.1±2.0 | 25.5±1.8 | 24.6±1.2 | 25.8±1.2 | 51.2±1.1 | 64.8±2.1 | 40.1±1.8 | 8.4±0.9 | 12.1±0.5 | 10.0±0.3 | 36.1±2.1 | 54.0±1.3 | 25.7±1.2 | 4.8±0.2 | 4.7±0.4 | 5.3±0.7 | 45.2±1.6 | 62.1±1.4 | 36.9±1.8 | 13.5±1.2 | 20.9±1.4 | 9.8±1.5 | 37.1±1.6 | 55.2±2.0 | 32.5±1.3 | 8.4±0.7 | 8.9±0.8 | 6.8±0.6 |
| G&M | 47.1±1.3 | 62.3±2.2 | 41.2±1.8 | 26.3±0.9 | 25.5±1.0 | 26.2±1.2 | 50.9±2.1 | 63.4±0.8 | 42.3±2.0 | 10.9±0.7 | 15.1±0.6 | 10.5±0.4 | 34.1±2.0 | 50.2±1.5 | 27.3±1.1 | 4.3±0.4 | 4.1±0.7 | 5.2±0.4 | 40.3±2.3 | 61.1±1.3 | 34.2±1.5 | 11.4±1.8 | 19.2±1.1 | 8.8±1.7 | 32.4±2.2 | 50.1±2.1 | 27.6±1.3 | 7.5±0.9 | 9.2±0.5 | 5.5±0.8 |
| PA-CGCD | 55.4±1.6 | 70.3±1.1 | 48.1±1.5 | 30.1±1.4 | 30.8±1.3 | 30.2±1.3 | 55.1±1.9 | 70.7±1.6 | 46.6±1.2 | 12.3±0.7 | 16.1±0.5 | 11.2±0.4 | 43.6±1.7 | 60.4±1.9 | 34.2±1.5 | 5.1±0.4 | 5.0±0.5 | 6.0±0.6 | 52.2±1.8 | 70.3±1.3 | 44.6±1.5 | 17.8±1.8 | 24.5±1.9 | 12.3±1.3 | 45.2±1.4 | 61.3±1.2 | 38.1±1.5 | 9.0±0.3 | 11.8±0.8 | 7.1±0.4 |
| DEAN | 56.0±1.2 | 71.7±1.9 | 47.9±2.0 | 32.8±1.1 | 34.4±1.4 | 31.5±2.1 | 56.7±2.0 | 71.5±2.2 | 47.6±1.5 | 12.9±0.5 | 16.8±0.9 | 11.2±0.7 | 44.0±2.1 | 61.0±1.7 | 35.1±2.1 | 5.3±0.8 | 5.1±0.6 | 6.2±0.2 | 55.1±2.2 | 72.7±1.2 | 47.8±1.2 | 20.3±1.9 | 26.7±1.5 | 15.0±1.2 | 46.7±1.7 | 62.5±1.5 | 40.0±1.4 | 9.5±0.4 | 12.5±0.6 | 7.9±0.9 |
| PromptCD | 56.5±1.1 | 71.2±1.7 | 50.3±1.8 | 31.5±1.5 | 32.1±1.9 | 31.2±1.4 | 57.4±1.3 | 73.6±1.5 | 48.6±1.2 | 13.4±0.8 | 17.7±0.4 | 12.1±1.0 | 45.2±1.3 | 62.3±1.6 | 36.7±1.0 | 5.8±0.3 | 5.1±0.8 | 6.5±0.3 | 54.1±1.5 | 71.2±1.1 | 46.7±1.4 | 19.8±1.5 | 26.1±1.6 | 14.4±1.4 | 47.1±1.5 | 63.1±1.3 | 40.2±1.2 | 9.2±0.3 | 12.2±0.6 | 7.8±0.5 |
| VB-CGCD | 57.3±1.8 | 71.0±1.2 | 52.4±1.1 | 32.4±2.2 | 33.6±2.3 | 32.5±2.1 | 56.9±1.4 | 73.1±1.1 | 48.8±1.9 | 13.9±0.6 | 18.1±0.7 | 12.9±0.6 | 47.1±1.0 | 62.1±2.2 | 38.1±1.5 | 6.0±0.5 | 4.9±0.4 | 6.8±0.7 | 55.4±1.7 | 72.0±1.2 | 47.5±2.3 | 19.6±2.0 | 25.8±2.2 | 14.2±2.2 | 46.3±1.8 | 63.9±2.2 | 41.9±1.1 | 9.4±0.3 | 12.4±0.5 | 8.0±0.4 |
| PRISM | 60.9±1.5 | 74.1±1.4 | 55.1±2.1 | 39.2±1.5 | 39.0±1.4 | 38.2±1.7 | 60.1±1.2 | 73.4±2.0 | 51.0±1.1 | 16.9±0.4 | 20.1±0.8 | 15.9±0.3 | 54.0±1.0 | 74.0±2.1 | 49.2±1.9 | 7.1±0.5 | 6.5±0.4 | 7.4±0.6 | 58.0±1.5 | 72.3±1.2 | 51.2±2.1 | 24.0±1.8 | 30.4±1.4 | 19.1±1.1 | 60.1±1.4 | 73.8±1.4 | 53.1±1.9 | 10.9±0.5 | 14.1±0.5 | 9.8±0.6 |

Table 8: Clustering results (**mean ± std**) on the SSB-C benchmarks. For each dataset, we evaluate on both *Original* and *Corrupted* domains, reporting average accuracies over *All*, *Old*, and *New* categories across different stages.

| Methods | CUB-C | | | | | | Stanford Cars-C | | | | | | FGVC-Aircraft-C | | | | | |
| | Original | | | Corrupted | | | Original | | | Corrupted | | | Original | | | Corrupted | | |
| | All | Old | New | All | Old | New | All | Old | New | All | Old | New | All | Old | New | All | Old | New |
| GCD | 29.4±1.4 | 47.7±1.5 | 23.4±1.5 | 26.8±1.3 | 45.9±1.5 | 20.1±2.2 | 26.4±1.0 | 56.1±1.8 | 21.5±1.7 | 22.3±1.7 | 43.1±1.0 | 11.2±1.6 | 27.7±1.0 | 33.6±1.2 | 24.9±2.1 | 28.8±2.2 | 41.4±1.4 | 28.8±1.8 |
| SimGCD | 26.6±1.5 | 44.5±2.0 | 21.0±2.1 | 23.4±2.0 | 42.4±1.9 | 17.7±1.2 | 23.1±1.6 | 52.5±1.4 | 18.9±1.1 | 19.3±0.8 | 39.7±2.2 | 9.8±1.5 | 25.4±2.1 | 30.1±1.6 | 22.1±2.3 | 25.2±2.0 | 38.1±1.0 | 25.8±1.4 |
| SPTNet | 27.8±2.3 | 45.2±1.5 | 22.0±1.8 | 25.1±1.2 | 44.2±1.2 | 18.1±0.8 | 24.9±1.7 | 55.0±2.1 | 20.3±1.3 | 21.1±1.1 | 41.6±2.0 | 9.9±1.0 | 26.1±1.7 | 31.2±2.3 | 23.3±1.2 | 26.9±1.7 | 39.5±1.6 | 26.7±1.7 |
| RLCD | 29.1±1.3 | 46.8±1.1 | 23.8±1.6 | 26.2±1.4 | 45.3±1.3 | 19.4±1.0 | 26.8±1.9 | 56.9±1.6 | 22.1±1.8 | 22.9±1.4 | 43.2±1.1 | 9.7±1.7 | 27.8±1.5 | 32.3±1.0 | 24.2±1.9 | 27.3±1.4 | 40.7±1.9 | 28.1±1.4 |
| G&M | 16.4±1.5 | 34.1±1.3 | 10.5±0.9 | 13.7±2.1 | 32.1±1.3 | 7.7±1.5 | 15.7±2.0 | 43.8±1.9 | 12.3±1.2 | 11.4±1.7 | 30.5±1.5 | 6.7±2.1 | 20.5±2.1 | 24.8±0.8 | 17.9±1.1 | 21.6±2.1 | 32.7±2.0 | 22.3±1.3 |
| PA-CGCD | 28.3±1.7 | 46.5±1.6 | 22.7±1.8 | 25.4±1.2 | 44.7±1.9 | 18.4±1.6 | 25.2±1.9 | 55.1±2.2 | 20.9±1.0 | 21.2±1.1 | 41.5±2.3 | 10.2±1.2 | 26.4±1.3 | 31.4±1.7 | 23.7±1.6 | 27.8±2.2 | 40.1±2.3 | 27.2±1.2 |
| DEAN | 28.9±1.2 | 47.1±2.1 | 23.0±1.1 | 26.3±1.5 | 46.2±2.3 | 18.2±1.4 | 26.1±1.7 | 58.1±1.9 | 19.4±0.9 | 22.1±1.6 | 41.2±1.2 | 12.9±2.0 | 28.1±1.3 | 32.8±1.9 | 28.9±1.7 | 29.1±2.3 | 40.1±2.2 | 30.3±1.1 |
| PromptCD | 30.1±1.1 | 48.1±1.3 | 24.5±1.2 | 27.4±1.6 | 46.1±1.4 | 20.3±1.5 | 27.4±1.7 | 57.4±2.0 | 22.1±1.1 | 23.1±1.6 | 44.4±1.9 | 11.4±1.3 | 29.9±1.8 | 34.5±1.2 | 26.4±2.3 | 30.3±1.7 | 42.9±2.0 | 29.9±1.3 |
| VB-CGCD | 34.2±1.4 | 51.8±1.3 | 26.3±1.6 | 31.7±1.1 | 49.2±1.3 | 23.4±1.4 | 31.6±1.5 | 59.9±1.8 | 26.1±1.2 | 26.3±1.5 | 47.9±1.6 | 15.1±1.0 | 33.2±1.6 | 37.3±1.2 | 29.7±1.1 | 32.3±1.9 | 44.5±2.0 | 31.6±1.8 |
| PRISM | 49.3±1.2 | 64.9±1.3 | 44.2±1.3 | 44.0±1.2 | 60.9±1.5 | 37.0±1.0 | 36.9±1.5 | 60.0±1.0 | 29.1±1.4 | 33.3±1.4 | 56.5±1.0 | 23.5±0.9 | 40.1±1.1 | 48.9±1.1 | 40.1±1.4 | 36.4±1.3 | 46.1±1.4 | 34.1±1.2 |

### A.5 COMPREHENSIVE CLUSTERING EVALUATION

To assess both robustness and effectiveness, we perform extensive multi-stage clustering studies on the **SSB-C** and **DomainNet** benchmarks. The summarized outcomes in Tables 7 and 8 report the overall performance with corresponding standard deviations (*mean ± std*), showing that our approach consistently outperforms prior baselines in terms of accuracy and stability.

In addition, a fine-grained analysis of stage-by-stage performance is provided in Tables 9 and 10. These results include clustering accuracy (%) on *All*, *Old*, and *New* categories across incremental stages, as well as the overall averages. Table 9 focuses on the DomainNet benchmark under different domain shift scenarios, whereas Table 10 presents evaluations on FGVC-Aircraft-C, Stanford Cars-C, and CUB-C. Such detailed investigations further highlight the strength of our method in reliably identifying novel categories under both distributional changes and sequential learning settings.

### A.6 ATTENTION MAP VISUALIZATION

To further probe how our model performs spatial reasoning, we inspect the attention distributions of the last transformer block, focusing on the relationship between the `[CLS]` token and the individual patch tokens across different heads. For each sample, we calculate the attention weights and highlight in red the top 10% regions receiving the strongest responses, thereby revealing the areas the model regards as most informative. Figure 4 illustrates representative visualizations from both source and target domains, including examples from seen as well as novel categories. Regardless of the data setting, our method consistently emphasizes semantically meaningful parts of the object, rather than being distracted by superficial appearance differences or domain-specific artifacts. This indicates that the learned attention patterns capture object-level semantics in a stable manner. Such consistency in attention allocation underscores the model's ability to filter out irrelevant background details and concentrate on discriminative structures. By anchoring the focus on task-relevant cues, the representations acquired by our framework generalize better across domains and facilitate reliable discovery of novel categories.

Table 9: Stage-wise clustering performance (%) of different methods on the DomainNet benchmark. Results are reported for all categories (All), previously known categories (Old), and newly discovered categories (New) at each incremental stage, along with the overall average.

| Methods | Stage 1 | | | Stage 2 | | | Stage 3 | | | Average | | | Stage 1 | | | Stage 2 | | | Stage 3 | | | Average | | |
|---|---|---|---|---|---|---|---|---|---|---|---|---|---|---|---|---|---|---|---|---|---|---|---|---|
| | All | Old | New | All | Old | New | All | Old | New | All | Old | New | All | Old | New | All | Old | New | All | Old | New | All | Old | New |
| | *Real → Painting* | | | | | | | | | | | | | | | | | | | | | | | |
| | *Real* | | | | | | | | | | | | *Painting* | | | | | | | | | | | |
| GCD | 54.4 | 70.6 | 47.7 | 47.8 | 63.9 | 43.1 | 51.7 | 67.1 | 45.4 | 51.3 | 67.2 | 45.4 | 28.6 | 28.0 | 29.3 | 26.1 | 25.4 | 26.8 | 27.5 | 26.7 | 28.3 | 27.4 | 26.7 | 28.1 |
| SimGCD | 51.7 | 66.3 | 44.1 | 45.6 | 61.5 | 38.4 | 48.0 | 63.9 | 41.4 | 48.4 | 63.9 | 41.3 | 23.8 | 23.6 | 24.7 | 21.4 | 21.3 | 22.3 | 22.6 | 22.3 | 23.5 | 22.6 | 22.4 | 23.5 |
| SPTNet | 52.9 | 67.1 | 45.3 | 46.4 | 61.7 | 40.0 | 50.0 | 64.7 | 42.1 | 49.8 | 64.5 | 42.5 | 25.3 | 24.7 | 25.8 | 22.9 | 22.2 | 22.9 | 24.0 | 23.6 | 24.3 | 24.1 | 23.5 | 24.3 |
| RLCD | 53.3 | 69.4 | 46.6 | 48.5 | 63.2 | 41.4 | 50.7 | 66.0 | 44.3 | 50.8 | 66.2 | 44.1 | 26.7 | 25.8 | 27.0 | 24.2 | 23.3 | 24.6 | 25.6 | 24.7 | 25.8 | 25.5 | 24.6 | 25.8 |
| G&M | 50.0 | 66.1 | 45.0 | 44.5 | 58.6 | 37.4 | 46.9 | 62.2 | 41.2 | 47.1 | 62.3 | 41.2 | 27.7 | 26.6 | 27.3 | 24.9 | 24.4 | 25.1 | 26.4 | 25.5 | 26.2 | 26.3 | 25.5 | 26.2 |
| PA-CGCD | 58.1 | 73.6 | 50.8 | 52.8 | 67.3 | 45.2 | 55.3 | 70.0 | 48.3 | 55.4 | 70.3 | 48.1 | 31.3 | 32.0 | 31.5 | 28.8 | 29.5 | 29.0 | 30.2 | 31.0 | 30.1 | 30.1 | 30.8 | 30.2 |
| DEAN | 58.6 | 75.1 | 51.0 | 52.9 | 68.3 | 45.3 | 56.5 | 71.7 | 47.4 | 56.0 | 71.7 | 47.9 | 33.9 | 35.8 | 32.6 | 31.7 | 33.0 | 30.4 | 32.8 | 34.4 | 31.5 | 32.8 | 34.4 | 31.5 |
| PromptCCD | 59.1 | 74.6 | 53.8 | 53.9 | 67.9 | 47.1 | 56.5 | 71.1 | 50.0 | 56.5 | 71.2 | 50.3 | 32.7 | 33.4 | 32.5 | 30.2 | 30.8 | 29.9 | 31.6 | 32.1 | 31.2 | 31.5 | 32.1 | 31.2 |
| VB-CGCD | 60.3 | 73.1 | 54.9 | 54.4 | 68.8 | 49.5 | 57.2 | 71.1 | 52.8 | 57.3 | 71.0 | 52.4 | 33.6 | 34.9 | 33.8 | 31.2 | 32.4 | 31.2 | 32.4 | 33.5 | 32.6 | 32.4 | 33.6 | 32.5 |
| PRISM | 63.3 | 76.6 | 58.9 | 58.6 | 71.7 | 51.4 | 60.8 | 74.0 | 55.0 | 60.9 | 74.1 | 55.1 | 40.6 | 40.3 | 39.6 | 37.8 | 37.8 | 36.9 | 39.2 | 38.9 | 38.1 | 39.2 | 39.0 | 38.2 |
| | *Real → Sketch* | | | | | | | | | | | | | | | | | | | | | | | |
| | *Real* | | | | | | | | | | | | *Sketch* | | | | | | | | | | | |
| GCD | 55.4 | 69.1 | 44.0 | 48.8 | 62.4 | 39.4 | 52.7 | 65.6 | 41.7 | 52.3 | 65.7 | 41.7 | 10.4 | 15.8 | 11.3 | 7.9 | 13.2 | 8.8 | 9.3 | 14.5 | 10.3 | 9.2 | 14.5 | 10.1 |
| SimGCD | 51.8 | 62.6 | 39.3 | 45.7 | 57.8 | 33.6 | 48.1 | 60.2 | 36.6 | 48.5 | 60.2 | 36.5 | 8.4 | 12.5 | 10.4 | 6.0 | 10.2 | 8.0 | 7.2 | 11.2 | 9.2 | 7.2 | 11.3 | 9.2 |
| SPTNet | 53.0 | 64.9 | 40.6 | 46.5 | 59.5 | 35.3 | 50.1 | 62.5 | 37.4 | 49.9 | 62.3 | 37.8 | 9.1 | 12.9 | 11.1 | 6.7 | 10.4 | 8.2 | 7.8 | 11.8 | 9.6 | 7.9 | 11.7 | 9.6 |
| RLCD | 53.7 | 68.0 | 42.6 | 48.9 | 61.8 | 37.4 | 51.1 | 64.6 | 40.3 | 51.2 | 64.8 | 40.1 | 9.6 | 13.3 | 11.2 | 7.1 | 10.8 | 8.8 | 8.5 | 12.2 | 10.0 | 8.4 | 12.1 | 10.0 |
| G&M | 51.8 | 67.2 | 46.1 | 46.3 | 59.7 | 38.5 | 48.7 | 63.3 | 42.3 | 48.9 | 63.4 | 42.3 | 12.3 | 16.2 | 11.6 | 9.5 | 14.0 | 9.4 | 11.0 | 15.1 | 10.5 | 10.9 | 15.1 | 10.5 |
| PA-CGCD | 57.8 | 74.0 | 49.3 | 52.5 | 67.7 | 43.7 | 55.0 | 70.4 | 46.8 | 55.1 | 70.7 | 46.6 | 13.5 | 17.3 | 12.5 | 11.0 | 14.8 | 10.0 | 12.4 | 16.3 | 11.1 | 12.3 | 16.1 | 11.2 |
| DEAN | 59.3 | 74.9 | 50.7 | 53.6 | 68.1 | 45.0 | 57.2 | 71.5 | 47.1 | 56.7 | 71.5 | 47.6 | 14.0 | 18.2 | 12.3 | 11.8 | 15.4 | 10.1 | 12.9 | 16.8 | 11.2 | 12.9 | 16.8 | 11.2 |
| PromptCCD | 60.0 | 77.0 | 52.1 | 54.8 | 70.3 | 45.4 | 57.4 | 73.5 | 48.3 | 57.4 | 73.6 | 48.6 | 14.6 | 19.0 | 13.4 | 12.1 | 16.4 | 10.8 | 13.5 | 17.7 | 12.1 | 13.4 | 17.7 | 12.1 |
| VB-CGCD | 59.9 | 75.2 | 51.3 | 54.0 | 70.9 | 45.9 | 56.8 | 73.2 | 49.2 | 56.9 | 73.1 | 48.8 | 15.1 | 19.4 | 14.2 | 12.7 | 16.9 | 11.6 | 13.9 | 18.0 | 13.0 | 13.9 | 18.1 | 12.9 |
| PRISM | 62.5 | 75.9 | 54.8 | 57.8 | 71.0 | 47.3 | 60.0 | 73.3 | 50.9 | 60.1 | 73.4 | 51.0 | 18.3 | 21.4 | 17.3 | 15.5 | 18.9 | 14.6 | 16.9 | 20.0 | 15.8 | 16.9 | 20.1 | 15.9 |
| | *Real → Quickdraw* | | | | | | | | | | | | | | | | | | | | | | | |
| | *Real* | | | | | | | | | | | | *Quickdraw* | | | | | | | | | | | |
| GCD | 41.8 | 59.6 | 31.9 | 39.1 | 56.1 | 29.6 | 35.2 | 52.9 | 27.3 | 38.7 | 56.2 | 29.6 | 6.2 | 6.0 | 7.0 | 5.1 | 4.7 | 6.0 | 3.7 | 3.4 | 4.5 | 5.0 | 4.7 | 5.8 |
| SimGCD | 35.7 | 52.7 | 26.3 | 32.0 | 50.3 | 23.6 | 29.6 | 47.9 | 20.6 | 32.4 | 50.3 | 23.5 | 5.4 | 5.2 | 6.3 | 4.2 | 3.9 | 5.1 | 3.0 | 2.9 | 3.9 | 4.2 | 4.0 | 5.1 |
| SPTNet | 37.9 | 55.2 | 27.6 | 35.0 | 52.8 | 24.4 | 31.4 | 49.8 | 22.3 | 34.8 | 52.6 | 24.8 | 6.1 | 5.8 | 7.0 | 4.8 | 4.7 | 5.5 | 3.7 | 3.3 | 4.1 | 4.9 | 4.6 | 5.5 |
| RLCD | 38.6 | 57.2 | 28.2 | 36.0 | 53.8 | 25.9 | 33.8 | 51.0 | 23.0 | 36.1 | 54.0 | 25.7 | 6.0 | 5.9 | 6.5 | 4.9 | 4.8 | 5.3 | 3.5 | 3.4 | 4.1 | 4.8 | 4.7 | 5.3 |
| G&M | 37.0 | 54.0 | 31.1 | 33.9 | 50.1 | 27.3 | 31.5 | 46.5 | 23.5 | 34.1 | 50.2 | 27.3 | 4.9 | 5.2 | 6.3 | 3.6 | 4.1 | 5.2 | 2.1 | 3.0 | 4.1 | 3.5 | 4.1 | 5.2 |
| PA-CGCD | 46.3 | 63.7 | 36.9 | 43.5 | 60.1 | 34.4 | 41.0 | 57.4 | 31.3 | 43.6 | 60.4 | 34.2 | 6.3 | 6.2 | 7.3 | 5.2 | 5.2 | 5.9 | 3.8 | 3.7 | 4.8 | 5.1 | 5.0 | 6.0 |
| DEAN | 46.6 | 64.4 | 38.2 | 44.5 | 61.0 | 34.6 | 40.9 | 57.6 | 32.5 | 44.0 | 61.0 | 35.1 | 6.2 | 6.4 | 7.2 | 5.1 | 5.0 | 6.1 | 4.0 | 3.6 | 5.0 | 5.1 | 5.0 | 6.1 |
| PromptCCD | 47.8 | 65.7 | 40.2 | 45.2 | 62.2 | 36.4 | 42.6 | 59.0 | 33.5 | 45.2 | 62.3 | 36.7 | 7.0 | 6.4 | 7.8 | 5.9 | 5.1 | 6.5 | 4.5 | 3.8 | 5.2 | 5.8 | 5.1 | 6.5 |
| VB-CGCD | 50.1 | 64.2 | 40.6 | 47.0 | 62.2 | 38.5 | 44.2 | 59.9 | 35.2 | 47.1 | 62.1 | 38.1 | 6.8 | 6.2 | 7.4 | 5.6 | 4.8 | 6.2 | 4.4 | 3.7 | 4.8 | 5.6 | 4.9 | 6.1 |
| PRISM | 56.4 | 76.5 | 53.0 | 53.9 | 73.9 | 49.1 | 51.7 | 71.6 | 45.5 | 54.0 | 74.0 | 49.2 | 8.5 | 7.8 | 8.8 | 7.1 | 6.4 | 7.3 | 5.7 | 5.3 | 6.1 | 7.1 | 6.5 | 7.4 |
| | *Real → Clipart* | | | | | | | | | | | | | | | | | | | | | | | |
| | *Real* | | | | | | | | | | | | *Clipart* | | | | | | | | | | | |
| GCD | 49.8 | 69.1 | 42.4 | 43.2 | 62.4 | 37.8 | 47.1 | 65.6 | 40.1 | 46.7 | 65.7 | 40.1 | 15.7 | 22.5 | 11.3 | 13.2 | 19.9 | 8.8 | 14.6 | 21.2 | 10.3 | 14.5 | 21.2 | 10.1 |
| SimGCD | 43.5 | 61.2 | 36.3 | 37.4 | 56.4 | 30.6 | 39.8 | 58.8 | 33.6 | 40.2 | 58.8 | 33.5 | 11.5 | 20.0 | 9.4 | 9.1 | 17.7 | 7.0 | 10.3 | 18.7 | 8.2 | 10.3 | 18.8 | 8.2 |
| SPTNet | 46.2 | 62.9 | 38.7 | 39.7 | 57.5 | 33.4 | 43.3 | 60.5 | 35.5 | 43.1 | 60.3 | 35.9 | 12.8 | 20.5 | 10.4 | 10.4 | 18.0 | 7.5 | 11.5 | 19.4 | 8.9 | 11.6 | 19.3 | 8.9 |
| RLCD | 47.7 | 65.3 | 39.4 | 42.9 | 59.1 | 34.2 | 45.1 | 61.9 | 37.1 | 45.2 | 62.1 | 36.9 | 14.7 | 22.1 | 11.0 | 12.2 | 19.6 | 8.6 | 13.6 | 21.0 | 9.8 | 13.5 | 20.9 | 9.8 |
| G&M | 43.2 | 64.9 | 38.0 | 37.7 | 57.4 | 30.4 | 40.1 | 61.0 | 34.2 | 40.3 | 61.1 | 34.2 | 13.7 | 20.3 | 9.9 | 10.9 | 18.1 | 7.7 | 12.4 | 19.2 | 8.8 | 12.3 | 19.2 | 8.8 |
| PA-CGCD | 54.9 | 73.6 | 47.3 | 49.6 | 67.3 | 41.7 | 52.1 | 70.0 | 44.8 | 52.2 | 70.3 | 44.6 | 19.0 | 25.7 | 13.6 | 16.5 | 23.2 | 11.1 | 17.9 | 24.7 | 12.2 | 17.8 | 24.5 | 12.3 |
| DEAN | 57.7 | 76.1 | 50.6 | 52.0 | 69.3 | 44.9 | 55.6 | 72.7 | 47.0 | 55.1 | 72.7 | 47.5 | 21.4 | 28.1 | 16.1 | 19.2 | 25.3 | 13.9 | 20.3 | 26.7 | 15.0 | 20.3 | 26.7 | 15.0 |
| PromptCCD | 56.7 | 74.6 | 50.2 | 51.5 | 67.9 | 43.5 | 54.1 | 71.1 | 46.4 | 54.1 | 71.2 | 46.7 | 21.0 | 27.4 | 15.7 | 18.5 | 24.8 | 13.1 | 19.9 | 26.1 | 14.4 | 19.8 | 26.1 | 14.4 |
| VB-CGCD | 58.4 | 74.1 | 50.0 | 52.5 | 69.8 | 44.6 | 55.3 | 72.1 | 47.9 | 55.4 | 72.0 | 47.5 | 20.8 | 27.1 | 15.5 | 18.4 | 24.6 | 12.9 | 19.6 | 25.7 | 14.3 | 19.6 | 25.8 | 14.2 |
| PRISM | 60.4 | 74.8 | 55.0 | 55.7 | 69.9 | 47.5 | 57.9 | 72.2 | 51.1 | 58.0 | 72.3 | 51.2 | 25.4 | 31.7 | 20.5 | 22.6 | 29.2 | 17.8 | 24.0 | 30.3 | 19.0 | 24.0 | 30.4 | 19.1 |
| | *Real → Infograph* | | | | | | | | | | | | | | | | | | | | | | | |
| | *Real* | | | | | | | | | | | | *Infograph* | | | | | | | | | | | |
| GCD | 42.9 | 58.7 | 34.7 | 40.2 | 55.2 | 32.4 | 36.3 | 52.0 | 30.1 | 39.8 | 55.3 | 32.4 | 9.3 | 11.1 | 7.6 | 8.2 | 9.8 | 6.6 | 6.8 | 8.5 | 5.1 | 8.1 | 9.8 | 6.4 |
| SimGCD | 36.9 | 51.6 | 30.6 | 33.2 | 49.2 | 27.9 | 30.8 | 46.8 | 24.9 | 33.6 | 49.2 | 27.8 | 7.9 | 9.0 | 6.4 | 6.7 | 7.7 | 5.2 | 5.5 | 6.7 | 4.0 | 6.7 | 7.8 | 5.2 |
| SPTNet | 39.0 | 54.0 | 32.6 | 36.1 | 51.6 | 29.4 | 32.5 | 48.6 | 27.3 | 35.9 | 51.4 | 29.8 | 8.4 | 9.2 | 7.4 | 7.1 | 8.1 | 5.9 | 6.0 | 6.7 | 4.5 | 7.2 | 8.0 | 5.9 |
| RLCD | 39.6 | 56.4 | 35.0 | 37.0 | 53.0 | 32.7 | 34.8 | 50.2 | 29.8 | 37.1 | 53.2 | 32.5 | 9.6 | 10.1 | 8.0 | 8.5 | 9.0 | 6.8 | 7.1 | 7.6 | 5.6 | 8.4 | 8.9 | 6.8 |
| G&M | 35.3 | 53.9 | 31.4 | 32.2 | 50.0 | 27.6 | 29.8 | 46.4 | 23.8 | 32.4 | 50.1 | 27.6 | 8.9 | 10.3 | 6.6 | 7.6 | 9.2 | 5.5 | 6.1 | 8.1 | 4.4 | 7.5 | 9.2 | 5.5 |
| PA-CGCD | 47.9 | 64.6 | 40.8 | 45.1 | 61.0 | 38.3 | 42.6 | 58.3 | 35.2 | 45.2 | 61.3 | 38.1 | 10.2 | 13.0 | 8.4 | 9.1 | 12.0 | 7.0 | 7.7 | 10.5 | 5.9 | 9.0 | 11.8 | 7.1 |
| DEAN | 49.3 | 65.7 | 43.9 | 47.2 | 62.3 | 40.3 | 43.6 | 58.9 | 38.2 | 46.7 | 62.3 | 40.8 | 10.6 | 13.9 | 9.0 | 9.5 | 12.5 | 7.9 | 8.4 | 11.1 | 6.8 | 9.5 | 12.5 | 7.9 |
| PromptCCD | 49.7 | 66.5 | 43.7 | 47.1 | 63.0 | 39.9 | 44.5 | 59.8 | 37.0 | 47.1 | 63.1 | 40.2 | 10.4 | 13.5 | 9.1 | 9.3 | 12.2 | 7.8 | 7.9 | 10.9 | 6.5 | 9.2 | 12.2 | 7.8 |
| VB-CGCD | 51.3 | 66.0 | 44.4 | 48.2 | 64.0 | 42.3 | 45.4 | 61.7 | 39.0 | 48.3 | 63.9 | 41.9 | 10.6 | 13.7 | 9.3 | 9.4 | 12.3 | 8.1 | 8.2 | 11.2 | 6.7 | 9.4 | 12.4 | 8.0 |
| PRISM | 62.5 | 76.3 | 56.9 | 60.0 | 73.7 | 53.0 | 57.8 | 71.4 | 49.4 | 60.1 | 73.8 | 53.1 | 12.3 | 15.4 | 11.2 | 10.9 | 14.0 | 9.7 | 9.5 | 12.9 | 8.5 | 10.9 | 14.1 | 9.8 |

## A.7 INTEGRATING CONTEMPORARY DOMAIN ADAPTATION METHODS FOR OW-CCD

To further investigate whether the challenges of OW-CCD can be addressed by existing techniques, we directly applied several contemporary domain adaptation (DA) methods, including Mixstyle (Xu et al., 2020), class-unknown adversarial adaptation (cUADAL) (Jang et al., 2022), unknown-oriented learning (UOL) (Liu et al., 2022), and the adjustment-and-alignment approach (ANNA) (Li et al., 2023b). The results, summarized in Table 11, show that these methods bring only limited improvements and sometimes even lead to negative transfer, as they focus on distribution alignment but lack the ability to robustly discover novel categories. In contrast, our proposed method consistently outperforms these DA baselines across all benchmarks. The results underscore two key insights: (1) addressing OW-CCD requires going beyond simple domain alignment, by explicitly modeling the interplay between known and unknown categories under evolving distributions; and (2) the proposed design provides a more principled solution tailored for OW-CCD. Taken together, these findings validate the necessity of customizing algorithms for OW-CCD, rather than relying on direct adaptations of existing DA methods.

Table 10: Stage-wise clustering accuracy (%) of all methods on FGVC-Aircraft-C, Stanford Cars-C, and CUB-C datasets. We report the accuracy on all classes (All), known classes (Old), and novel classes (New) at each incremental stage, as well as the average.

| Methods | Stage 1 | | | Stage 2 | | | Stage 3 | | | Average | | | Stage 1 | | | Stage 2 | | | Stage 3 | | | Average | | |
|---|---|---|---|---|---|---|---|---|---|---|---|---|---|---|---|---|---|---|---|---|---|---|---|---|
| | All | Old | New | All | Old | New | All | Old | New | All | Old | New | All | Old | New | All | Old | New | All | Old | New | All | Old | New |
| | | | | | | *Original* | | | | | | | | | | | | *Corrupted* | | | | | | |
| | | | | | | | | | | | | *FGVC-Aircraft-C* | | | | | | | | | | | | |
| GCD | 30.8 | 37.0 | 27.2 | 24.2 | 30.3 | 22.6 | 28.1 | 33.5 | 24.9 | 27.7 | 33.6 | 24.9 | 30.0 | 42.7 | 30.0 | 27.5 | 40.1 | 27.5 | 28.9 | 41.4 | 29.0 | 28.8 | 41.4 | 28.8 |
| SimGCD | 28.7 | 32.5 | 24.9 | 22.6 | 27.7 | 19.2 | 25.0 | 30.1 | 22.2 | 25.4 | 30.1 | 22.1 | 26.4 | 39.3 | 27.0 | 24.0 | 37.0 | 24.6 | 25.2 | 38.0 | 25.8 | 25.2 | 38.1 | 25.8 |
| SPTNet | 29.2 | 33.8 | 26.1 | 22.7 | 28.4 | 20.8 | 26.3 | 31.4 | 22.9 | 26.1 | 31.2 | 23.3 | 28.1 | 40.7 | 28.2 | 25.7 | 38.2 | 25.3 | 26.8 | 39.6 | 26.7 | 26.9 | 39.5 | 26.7 |
| RLCD | 30.3 | 35.5 | 26.7 | 25.5 | 29.3 | 21.5 | 27.7 | 32.1 | 24.4 | 27.8 | 32.3 | 24.2 | 29.0 | 41.9 | 29.3 | 26.5 | 39.4 | 26.9 | 27.9 | 40.8 | 28.1 | 27.8 | 40.7 | 28.1 |
| G&M | 23.4 | 28.6 | 21.7 | 17.9 | 21.1 | 14.1 | 20.3 | 24.7 | 17.9 | 20.5 | 24.8 | 17.9 | 23.0 | 33.8 | 23.4 | 20.2 | 31.6 | 21.2 | 21.7 | 32.7 | 22.3 | 21.6 | 32.7 | 22.3 |
| PA-CGCD | 29.1 | 34.7 | 26.4 | 23.8 | 28.4 | 20.8 | 26.3 | 31.1 | 23.9 | 26.4 | 31.4 | 23.7 | 29.0 | 41.3 | 28.5 | 26.5 | 38.8 | 26.0 | 27.9 | 40.3 | 27.1 | 27.8 | 40.1 | 27.2 |
| DEAN | 30.7 | 36.2 | 32.0 | 25.0 | 29.4 | 26.3 | 28.6 | 32.8 | 28.4 | 28.1 | 32.8 | 28.9 | 30.2 | 41.5 | 31.4 | 28.0 | 38.7 | 29.2 | 29.1 | 40.1 | 30.3 | 29.1 | 40.1 | 30.3 |
| PromptCCD | 32.5 | 37.9 | 29.9 | 27.3 | 31.2 | 23.2 | 29.9 | 34.4 | 26.1 | 29.9 | 34.5 | 26.4 | 31.5 | 44.2 | 31.2 | 29.0 | 41.6 | 28.6 | 30.4 | 42.9 | 29.9 | 30.3 | 42.9 | 29.9 |
| VB-CGCD | 36.2 | 39.4 | 32.2 | 30.3 | 35.1 | 26.8 | 33.1 | 37.4 | 30.1 | 33.2 | 37.3 | 29.7 | 33.5 | 45.8 | 32.9 | 31.1 | 43.3 | 30.3 | 32.3 | 44.4 | 31.7 | 32.3 | 44.5 | 31.6 |
| PRISM | 42.5 | 51.4 | 43.9 | 37.8 | 46.5 | 36.4 | 40.0 | 48.8 | 40.0 | 40.1 | 48.9 | 40.1 | 37.8 | 47.4 | 35.5 | 35.0 | 44.9 | 32.8 | 36.4 | 46.0 | 34.0 | 36.4 | 46.1 | 34.1 |
| | | | | | | | | | | | | *Stanford Cars-C* | | | | | | | | | | | | |
| | | | | | | *Original* | | | | | | | | | | | | *Corrupted* | | | | | | |
| GCD | 29.5 | 59.5 | 23.8 | 26.8 | 56.0 | 21.5 | 22.9 | 52.8 | 19.2 | 26.4 | 56.1 | 21.5 | 23.5 | 44.4 | 12.4 | 22.4 | 43.1 | 11.4 | 21.0 | 41.8 | 9.9 | 22.3 | 43.1 | 11.2 |
| SimGCD | 26.4 | 54.9 | 21.7 | 22.7 | 52.5 | 19.0 | 20.3 | 50.1 | 16.0 | 23.1 | 52.5 | 18.9 | 20.5 | 40.9 | 11.0 | 19.3 | 39.6 | 9.8 | 18.1 | 38.6 | 8.6 | 19.3 | 39.7 | 9.8 |
| SPTNet | 28.0 | 57.6 | 23.1 | 25.1 | 55.2 | 19.9 | 21.5 | 52.2 | 17.8 | 24.9 | 55.0 | 20.3 | 22.3 | 42.8 | 11.4 | 21.0 | 41.7 | 9.9 | 19.9 | 40.3 | 8.5 | 21.1 | 41.6 | 9.9 |
| RLCD | 28.7 | 60.1 | 24.6 | 26.1 | 56.7 | 22.3 | 23.9 | 53.9 | 19.4 | 26.2 | 56.9 | 22.1 | 24.1 | 44.4 | 10.9 | 23.0 | 43.3 | 9.7 | 21.6 | 41.9 | 8.5 | 22.9 | 43.2 | 9.7 |
| G&M | 18.6 | 47.6 | 16.1 | 15.5 | 43.7 | 12.3 | 13.1 | 40.1 | 8.5 | 15.7 | 43.8 | 12.3 | 12.8 | 31.6 | 7.8 | 11.5 | 30.5 | 6.7 | 10.0 | 29.4 | 5.6 | 11.4 | 30.5 | 6.7 |
| PA-CGCD | 27.9 | 58.4 | 23.6 | 25.1 | 54.8 | 21.1 | 22.6 | 52.1 | 18.0 | 25.2 | 55.1 | 20.9 | 22.4 | 42.7 | 11.5 | 21.3 | 41.7 | 10.1 | 19.9 | 40.2 | 9.0 | 21.2 | 41.5 | 10.2 |
| DEAN | 28.7 | 61.5 | 22.5 | 26.6 | 58.1 | 18.9 | 23.0 | 54.7 | 16.8 | 26.1 | 58.1 | 19.4 | 23.2 | 42.6 | 14.0 | 22.1 | 41.2 | 12.9 | 21.0 | 39.8 | 11.8 | 22.1 | 41.2 | 12.9 |
| PromptCCD | 30.0 | 60.8 | 25.6 | 27.4 | 57.3 | 21.8 | 24.8 | 54.1 | 18.9 | 27.4 | 57.4 | 22.1 | 24.3 | 45.7 | 12.7 | 23.2 | 44.4 | 11.4 | 21.8 | 43.1 | 10.1 | 23.1 | 44.4 | 11.4 |
| VB-CGCD | 34.6 | 62.0 | 28.6 | 31.5 | 60.0 | 26.5 | 28.7 | 57.7 | 23.2 | 31.6 | 59.9 | 26.1 | 27.5 | 49.2 | 16.4 | 26.3 | 47.8 | 15.2 | 25.1 | 46.7 | 13.8 | 26.3 | 47.9 | 15.1 |
| PRISM | 39.3 | 62.5 | 32.9 | 36.8 | 59.9 | 29.0 | 34.6 | 57.6 | 25.4 | 36.9 | 60.0 | 29.1 | 34.7 | 57.8 | 24.9 | 33.3 | 56.4 | 23.4 | 31.9 | 55.3 | 22.2 | 33.3 | 56.5 | 23.5 |
| | | | | | | | | | | | | *CUB-C* | | | | | | | | | | | | |
| | | | | | | *Original* | | | | | | | | | | | | *Corrupted* | | | | | | |
| GCD | 32.5 | 51.1 | 25.7 | 25.9 | 44.4 | 21.1 | 29.8 | 47.6 | 23.4 | 29.4 | 47.7 | 23.4 | 28.0 | 47.2 | 21.3 | 25.5 | 44.6 | 18.8 | 26.9 | 45.9 | 20.3 | 26.8 | 45.9 | 20.1 |
| SimGCD | 29.9 | 46.9 | 23.8 | 23.8 | 42.1 | 18.1 | 26.2 | 44.5 | 21.1 | 26.6 | 44.5 | 21.0 | 24.6 | 43.6 | 18.9 | 22.2 | 41.3 | 16.5 | 23.4 | 42.3 | 17.7 | 23.4 | 42.4 | 17.7 |
| SPTNet | 30.9 | 47.8 | 24.8 | 24.4 | 42.4 | 19.5 | 28.0 | 45.4 | 21.6 | 27.8 | 45.2 | 22.0 | 26.3 | 45.4 | 19.6 | 23.9 | 42.9 | 16.7 | 25.0 | 44.3 | 18.1 | 25.1 | 44.2 | 18.1 |
| RLCD | 31.6 | 50.0 | 26.3 | 26.8 | 43.8 | 21.1 | 29.0 | 46.6 | 24.0 | 29.1 | 46.8 | 23.8 | 28.4 | 46.5 | 20.6 | 25.9 | 44.0 | 18.2 | 27.3 | 45.4 | 19.4 | 27.2 | 45.3 | 19.4 |
| G&M | 19.3 | 37.9 | 14.3 | 13.8 | 30.4 | 6.7 | 16.2 | 34.0 | 10.5 | 16.4 | 34.1 | 10.5 | 15.1 | 33.2 | 8.8 | 12.3 | 31.0 | 6.6 | 13.8 | 32.1 | 7.7 | 13.7 | 32.1 | 7.7 |
| PA-CGCD | 31.0 | 49.8 | 25.4 | 25.7 | 43.5 | 19.8 | 28.2 | 46.2 | 22.9 | 28.3 | 46.5 | 22.7 | 26.6 | 45.9 | 19.7 | 24.1 | 43.4 | 17.2 | 25.5 | 44.9 | 18.3 | 25.4 | 44.7 | 18.4 |
| DEAN | 31.5 | 50.5 | 26.1 | 25.8 | 43.7 | 20.4 | 29.4 | 47.1 | 22.5 | 28.9 | 47.1 | 23.0 | 27.4 | 47.6 | 19.3 | 25.2 | 44.8 | 17.1 | 26.3 | 46.2 | 18.2 | 26.3 | 46.2 | 18.2 |
| PromptCCD | 32.7 | 51.5 | 28.0 | 27.5 | 44.8 | 21.3 | 30.1 | 48.0 | 24.2 | 30.1 | 48.1 | 24.5 | 28.6 | 47.4 | 21.6 | 26.1 | 44.8 | 19.0 | 27.5 | 46.1 | 20.3 | 27.4 | 46.1 | 20.3 |
| VB-CGCD | 37.2 | 53.9 | 28.8 | 31.3 | 49.6 | 23.4 | 34.1 | 51.9 | 26.7 | 34.2 | 51.8 | 26.3 | 32.9 | 50.5 | 24.7 | 30.5 | 48.0 | 22.1 | 31.7 | 49.1 | 23.5 | 31.7 | 49.2 | 23.4 |
| PRISM | 51.7 | 67.4 | 48.0 | 47.0 | 62.5 | 40.5 | 49.2 | 64.8 | 44.1 | 49.3 | 64.9 | 44.2 | 45.4 | 62.2 | 38.4 | 42.6 | 59.7 | 35.7 | 44.0 | 60.8 | 36.9 | 44.0 | 60.9 | 37.0 |

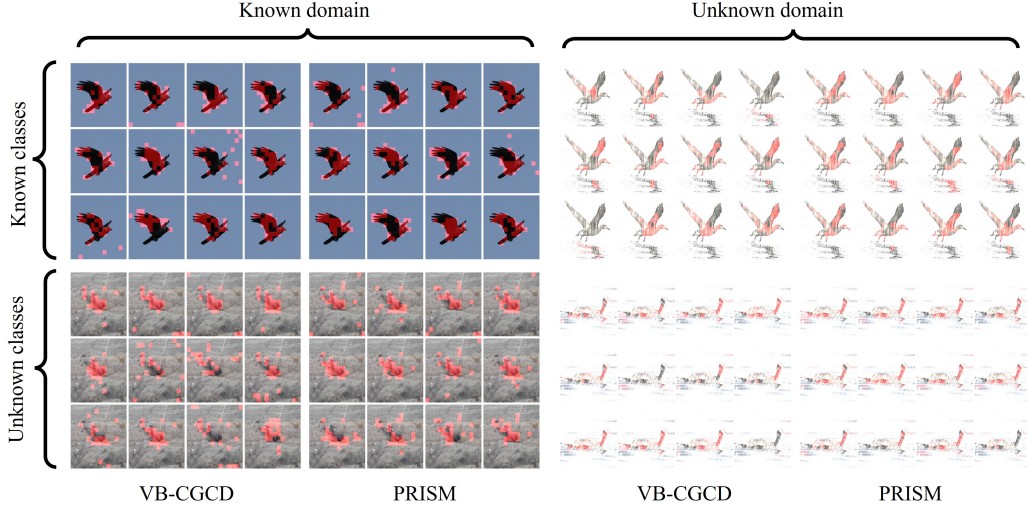

Figure 4: Attention heatmaps from the final ViT layer on the CUB-C benchmark. Red indicates the top 10% patches with the highest attention weights across multiple heads. Compared to background patterns, our model maintains a stronger focus on semantic object areas in both known and unknown domains, highlighting resilience to appearance variations.

## A.8 EMPIRICAL STUDY WITH DINOv2 BACKBONE

To further assess the robustness and effectiveness of our proposed framework, we conduct additional experiments using a stronger pretrained backbone, DINOv2, which has recently demonstrated superior representation learning ability in various vision tasks. As shown in Table 12, upgrading the

Table 11: Clustering performance of other DA methods.

| Method | Real | | | Painting | | |
|---|---|---|---|---|---|---|
| | All | Old | New | All | Old | New |
| UOL | 55.1 | 69.0 | 45.6 | 29.0 | 29.3 | 27.6 |
| Mixstyle | 53.2 | 67.1 | 44.0 | 26.0 | 25.2 | 25.1 |
| cUADAL | 55.7 | 70.1 | 46.4 | 29.9 | 28.8 | 26.3 |
| ANNA | 54.8 | 68.4 | 48.7 | 30.7 | 28.9 | 26.8 |
| PRISM | **60.9** | **74.1** | **55.1** | **39.2** | **39.0** | **38.2** |

Table 12: Clustering performance on SSB-C benchmarks using DINOv2 as the backbone. Each dataset includes both Original and Corrupted settings, and we report the average All / Old / New accuracy across all stages for both domains.

| Methods | CUB-C | | | | | | Stanford Cars-C | | | | | | FGVC-Aircraft-C | | | | | |
|---|---|---|---|---|---|---|---|---|---|---|---|---|---|---|---|---|---|---|
| | Original | | | Corrupted | | | Original | | | Corrupted | | | Original | | | Corrupted | | |
| | All | Old | New | All | Old | New | All | Old | New | All | Old | New | All | Old | New | All | Old | New |
| GCD | 41.2 | 55.6 | 34.9 | 38.5 | 53.9 | 31.5 | 46.3 | 65.4 | 43.4 | 42.3 | 53.7 | 32.3 | 38.1 | 39.8 | 36.1 | 39.4 | 49.3 | 40.4 |
| SimGCD | 39.3 | 52.4 | 33.6 | 35.3 | 51.4 | 30.5 | 42.5 | 63.3 | 41.0 | 39.5 | 50.0 | 30.9 | 36.1 | 37.4 | 34.1 | 36.1 | 44.2 | 34.7 |
| SPTNet | 40.5 | 52.4 | 34.3 | 36.9 | 51.5 | 30.6 | 45.0 | 64.5 | 41.4 | 40.4 | 51.7 | 32.1 | 36.0 | 36.9 | 31.3 | 37.8 | 48.4 | 36.4 |
| RLCD | 42.4 | 54.3 | 35.6 | 38.3 | 53.4 | 32.3 | 47.1 | 65.9 | 43.1 | 42.1 | 53.1 | 34.3 | 37.8 | 38.1 | 33.8 | 39.3 | 50.1 | 38.2 |
| G&M | 28.2 | 42.6 | 22.4 | 25.3 | 39.3 | 20.7 | 36.1 | 52.8 | 34.6 | 30.9 | 41.0 | 29.5 | 31.4 | 30.9 | 28.6 | 31.8 | 40.3 | 31.7 |
| PA-CGCD | 40.9 | 53.8 | 35.2 | 38.0 | 53.2 | 31.1 | 45.2 | 65.6 | 43.1 | 40.1 | 51.6 | 31.3 | 38.3 | 40.2 | 35.0 | 37.0 | 45.7 | 37.3 |
| DEAN | 41.7 | 54.5 | 35.5 | 37.8 | 53.6 | 29.6 | 45.8 | 67.4 | 41.7 | 41.7 | 50.8 | 35.7 | 40.0 | 40.2 | 38.1 | 38.8 | 45.6 | 41.1 |
| PromptCCD | 42.7 | 55.6 | 36.0 | 39.1 | 54.3 | 33.2 | 48.4 | 67.8 | 43.7 | 43.3 | 53.7 | 34.1 | 40.0 | 42.4 | 35.9 | 40.0 | 48.3 | 41.0 |
| VB-CGCD | 43.1 | 56.2 | 38.1 | 40.7 | 55.9 | 35.9 | 50.5 | **68.9** | 45.7 | 45.4 | 56.1 | 35.3 | 42.3 | 43.6 | 37.8 | 41.8 | 49.8 | 42.5 |
| **PRISM** | **63.1** | **73.8** | **54.9** | **56.4** | **71.1** | **49.8** | **56.8** | 68.8 | **50.1** | **53.5** | **67.3** | **43.2** | **50.7** | **56.4** | **50.1** | **46.3** | **51.1** | **44.7** |

Table 13: Clustering performance under the setting where multiple domains are treated as unknown. Specifically, we construct the unknown set by combining the five domains from DomainNet except for the Real domain, and report clustering accuracy separately for each domain.

| Methods | Real | | | Painting | | | Sketch | | | Quickdraw | | | Clipart | | | Infograph | | |
|---|---|---|---|---|---|---|---|---|---|---|---|---|---|---|---|---|---|---|
| | All | Old | New | All | Old | New | All | Old | New | All | Old | New | All | Old | New | All | Old | New |
| GCD | 50.9 | 66.7 | 44.9 | 26.9 | 26.3 | 27.7 | 8.8 | 14.0 | 9.6 | 4.7 | 2.3 | 3.6 | 12.2 | 20.8 | 9.7 | 7.6 | 9.4 | 6.0 |
| SimGCD | 48.0 | 63.5 | 40.8 | 22.1 | 22.1 | 23.2 | 6.7 | 10.9 | 8.9 | 3.7 | 1.6 | 2.7 | 8.0 | 18.3 | 7.8 | 6.4 | 7.4 | 4.8 |
| SPTNet | 49.4 | 64.1 | 42.1 | 23.7 | 23.2 | 24.0 | 7.6 | 11.2 | 9.2 | 4.5 | 2.4 | 3.0 | 10.3 | 18.9 | 8.5 | 6.9 | 7.5 | 5.5 |
| RLCD | 50.5 | 65.1 | 43.2 | 25.0 | 24.9 | 25.1 | 8.1 | 11.9 | 10.1 | 4.3 | 2.2 | 2.9 | 10.6 | 19.2 | 8.9 | 7.2 | 8.0 | 5.6 |
| G&M | 46.7 | 61.9 | 40.8 | 25.9 | 25.1 | 25.8 | 10.5 | 14.7 | 10.2 | 3.9 | 1.8 | 2.8 | 10.1 | 18.7 | 8.4 | 7.1 | 8.7 | 5.1 |
| PA-CGCD | 55.0 | 69.9 | 47.6 | 29.6 | 30.4 | 29.7 | 11.9 | 15.7 | 10.8 | 4.8 | 2.5 | 3.7 | 15.4 | 24.1 | 12.0 | 8.6 | 11.4 | 6.8 |
| DEAN | 55.5 | 71.2 | 47.5 | 32.3 | 33.9 | 31.1 | 12.5 | 16.4 | 10.8 | 5.0 | 2.6 | 3.9 | 18.0 | 26.3 | 14.6 | 9.1 | **12.9** | 7.4 |
| PromptCCD | 56.2 | 70.7 | 49.8 | 31.1 | 31.8 | 30.8 | 13.0 | 17.3 | 11.7 | 5.3 | 2.7 | 4.2 | 17.6 | 25.8 | 14.0 | 8.8 | 11.8 | 7.4 |
| VB-CGCD | 57.1 | 71.4 | 50.3 | 31.9 | 32.4 | 31.7 | 13.9 | 18.0 | 12.4 | 5.7 | 3.2 | **5.1** | 18.2 | 26.7 | 15.1 | 9.3 | 12.5 | 8.2 |
| **PRISM** | **60.4** | **73.0** | **54.8** | **38.8** | **39.6** | **37.8** | **15.8** | **19.5** | **14.9** | **6.9** | **5.0** | 4.9 | **21.9** | **29.8** | **19.3** | **10.9** | 12.6 | **9.3** |

Table 14: Clustering results on the DomainNet benchmark with extended-stage online discovery. We consider Real as the known domain, while each remaining domain is treated as unknown. Scores are averaged across all stages (including the 4-stage extension) and reported in terms of All / Old / New accuracy for each domain pair.

| Methods | Real → Painting | | | | | | Real → Sketch | | | | | | Real → Quickdraw | | | | | | Real → Clipart | | | | | | Real → Infograph | | | | | |
|---|---|---|---|---|---|---|---|---|---|---|---|---|---|---|---|---|---|---|---|---|---|---|---|---|---|---|---|---|---|---|
| | Real | | | Painting | | | Real | | | Sketch | | | Real | | | Quickdraw | | | Real | | | Clipart | | | Real | | | Infograph | | |
| | All | Old | New | All | Old | New | All | Old | New | All | Old | New | All | Old | New | All | Old | New | All | Old | New | All | Old | New | All | Old | New | All | Old | New |
| GCD | 50.9 | 66.8 | 44.9 | 27.8 | 27.2 | 28.5 | 52.8 | 66.3 | 42.2 | 8.7 | 13.9 | 9.7 | 38.1 | 55.9 | 29.2 | 5.4 | 5.3 | 6.3 | 47.0 | 66.2 | 40.6 | 13.9 | 20.7 | 9.5 | 39.4 | 55.0 | 31.9 | 8.6 | 10.2 | 6.8 |
| SimGCD | 47.9 | 63.5 | 40.8 | 22.9 | 23.0 | 24.0 | 49.0 | 60.5 | 36.9 | 6.7 | 10.9 | 8.7 | 32.1 | 49.8 | 23.2 | 4.5 | 4.5 | 5.6 | 40.6 | 59.3 | 34.1 | 9.7 | 18.4 | 7.8 | 33.1 | 48.8 | 27.3 | 7.1 | 8.2 | 5.7 |
| SPTNet | 49.3 | 64.1 | 42.0 | 24.5 | 23.9 | 24.9 | 50.4 | 62.8 | 38.3 | 7.6 | 11.1 | 9.3 | 34.3 | 52.3 | 24.4 | 5.3 | 5.0 | 6.0 | 43.4 | 60.6 | 36.2 | 11.1 | 19.0 | 8.4 | 35.4 | 50.9 | 29.3 | 7.5 | 8.5 | 6.5 |
| RLCD | 49.9 | 64.8 | 42.7 | 25.0 | 24.3 | 25.1 | 51.0 | 63.3 | 38.9 | 8.4 | 11.9 | 10.0 | 35.0 | 53.1 | 24.9 | 5.4 | 5.1 | 6.2 | 43.9 | 61.2 | 36.7 | 12.3 | 19.2 | 8.6 | 35.7 | 51.2 | 29.9 | 7.8 | 8.9 | 6.7 |
| G&M | 46.7 | 62.0 | 40.8 | 26.7 | 26.1 | 26.7 | 52.3 | 63.8 | 42.9 | 10.5 | 14.6 | 10.2 | 33.6 | 49.8 | 26.9 | 4.8 | 4.7 | 5.7 | 40.8 | 61.5 | 34.8 | 11.8 | 18.7 | 8.5 | 31.9 | 49.7 | 27.1 | 7.9 | 9.8 | 5.9 |
| PA-CGCD | 55.1 | 70.0 | 47.7 | 30.4 | 31.3 | 30.6 | 55.7 | 71.1 | 47.1 | 11.7 | 15.6 | 10.8 | 43.0 | 59.9 | 33.7 | 5.6 | 5.3 | 6.5 | 52.6 | **74.8** | 42.1 | 17.2 | 24.1 | 12.0 | 44.6 | 61.0 | 37.5 | 9.4 | 12.3 | 7.7 |
| DEAN | 55.6 | 71.2 | 47.6 | 33.2 | 34.9 | 31.9 | 57.1 | 72.0 | 48.2 | 12.4 | 16.4 | 10.6 | 43.0 | 60.1 | 33.7 | 5.6 | 5.4 | 6.6 | 55.7 | 73.3 | 48.0 | 19.9 | 26.3 | 14.7 | 46.2 | 62.0 | 40.3 | 10.1 | **14.5** | 7.4 |
| PromptCCD | 56.1 | 70.8 | 49.9 | 31.9 | 32.6 | 31.7 | 58.0 | 73.9 | 49.2 | 13.1 | 17.4 | 11.7 | 44.7 | 61.9 | 36.3 | 6.2 | 5.5 | 7.0 | 54.7 | 71.5 | 47.0 | 19.4 | 25.5 | 14.0 | 46.7 | 62.6 | 39.7 | 9.7 | 12.6 | 8.2 |
| VB-CGCD | 57.0 | 71.2 | 50.3 | 32.1 | 33.4 | 32.4 | 58.7 | **74.3** | 49.9 | 13.8 | 18.0 | 12.1 | 45.1 | 62.7 | 36.9 | 6.9 | 5.8 | 7.3 | 55.1 | 72.0 | 47.7 | 19.9 | 26.1 | 14.6 | 47.3 | 63.2 | 40.1 | 10.0 | 12.9 | 8.8 |
| **PRISM** | **60.8** | **73.9** | **54.5** | **40.1** | **38.5** | **39.5** | **60.7** | 73.3 | **52.1** | **16.3** | **19.7** | **15.3** | **53.3** | **73.1** | **48.4** | **7.3** | **6.4** | **7.6** | **57.7** | 72.8 | **51.9** | **23.4** | **29.0** | **18.8** | **59.1** | **74.5** | **50.8** | **11.8** | 13.2 | **10.0** |

backbone to DINOv2 consistently boosts the performance of all compared algorithms across different benchmarks. This confirms that stronger feature extractors can provide more transferable and discriminative representations for the OW-CCD tasks.

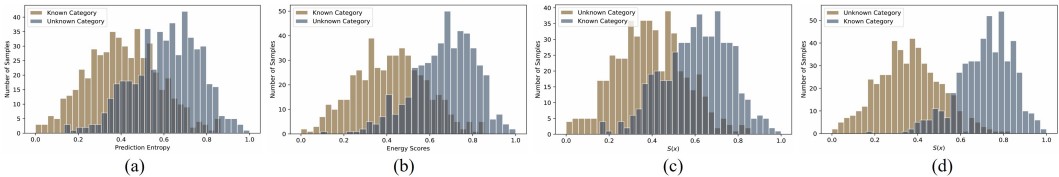

Figure 5: Comparison of different separation strategies.

More importantly, under this enhanced backbone setting, our proposed method still achieves the best overall performance and maintains a clear margin over state-of-the-art baselines. This demonstrates that the improvements brought by our framework are orthogonal to backbone advances, and our method continues to deliver substantial gains even when combined with powerful feature extractors. These results highlight the scalability and practical value of our framework when deployed with next-generation backbones such as DINOv2.

### A.9 EVALUATION UNDER MULTIPLE UNKNOWN DOMAINS

To further assess the robustness and practicality of our framework in more complex real-world scenarios, we conduct an additional experiment under a mixed-domain setting. On the DomainNet benchmark, we treat the `Real` domain as the known domain and merge all the remaining domains (`Clipart`, `Painting`, `Sketch`, and `Infograph`) into a single unknown domain. Compared with single-domain shifts, this mixed-domain setting introduces much greater diversity in both visual styles and semantic structures, making continual category discovery significantly more challenging.

As shown in Table 13, our method consistently outperforms state-of-the-art baselines across all metrics. In particular, we observe notable improvements on novel category discovery, indicating that the proposed approach remains effective even when the unlabeled data come from multiple heterogeneous domains. These results suggest that our framework generalizes well to realistic scenarios where unlabeled streams are inherently multi-sourced and non-stationary.

### A.10 EVALUATION UNDER MORE-STAGE SETTING

In our main evaluation, we design the online discovery task under a three-phase setting to verify the effectiveness of the proposed framework. To further examine its robustness and scalability under more demanding conditions, we additionally explore a four-phase scenario, where novel categories emerge in a slower and more fragmented manner.

As shown in Table 14, our approach consistently surpasses competitive baselines across all evaluation metrics and category partitions (*All*, *Old*, *New*). These results highlight the framework's ability to cope with increasingly incremental category arrivals, confirming its adaptability to dynamic and extended discovery processes.

### A.11 PARAMETER SENSITIVITY ANALYSIS

We further investigate how our method behaves under variations of the loss balancing coefficient $\lambda_1$, with results shown in Figure 6 (a)–(b) and (g)–(h). Experiments are conducted on both DomainNet and CUB-C. For DomainNet, the *Real* domain is treated as the known domain and the *Painting* domain as the unknown domain. For CUB-C, we follow the same protocol by designating the *Original* domain as the known domain and the *Corrupted* domain as the unknown domain. By adjusting $\lambda_1$, we measure clustering accuracy on all, old, and novel categories within both domains. A clear performance drop is observed when this coefficient is set to zero, highlighting the indispensability of the IKT module. This component aligns listwise ranking distributions between unknown samples and known prototypes before and after spectral perturbation, thereby mitigating spurious style effects and retaining transferable semantic knowledge. More importantly, across a wide range of $\lambda_1$ values, the accuracy remains consistently high, indicating that our approach is largely insensitive to this parameter and thus robust against hyperparameter tuning.

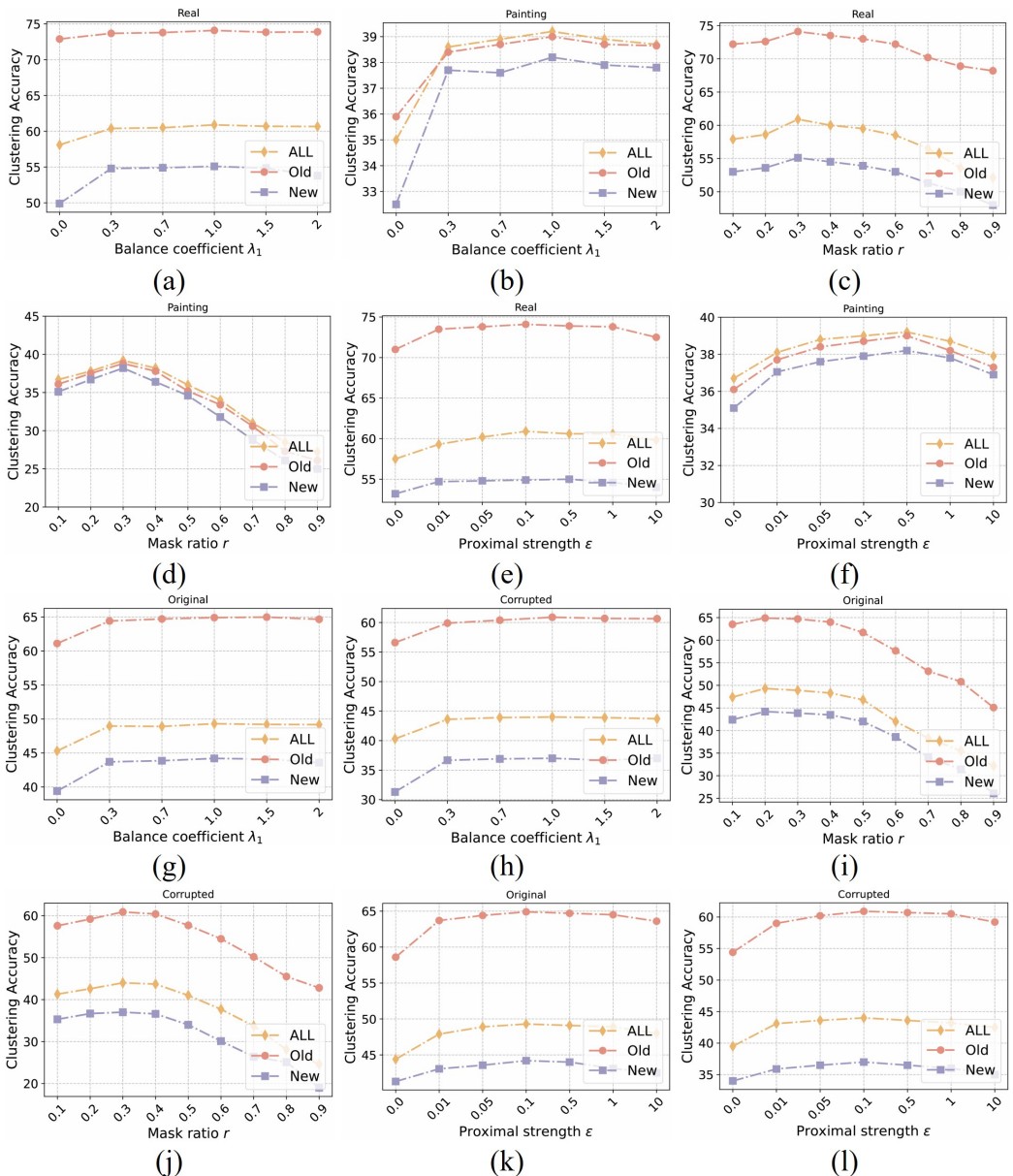

Figure 6: Sensitivity analysis of key hyperparameters.

The mask ratio $r$ regulates the binary mask $M$ used to split an image into low- and high-frequency parts, controlling their relative contribution. In our design, high-frequency signals guide the separation of known and novel categories, while low-frequency content is perturbed within the IKT module to encourage transferability. As presented in Figure 6 (c)–(d) and (i)–(j), performance peaks at $r = 0.3$, which we adopt as the default setting in all experiments unless stated otherwise. Increasing $r$ allows the model to exploit richer semantic cues from high-frequency components and enhances robustness of the IKT module by perturbing a broader spectrum of low-frequency information. Nevertheless, when $r$ becomes excessively large, the model suffers from limited high-frequency cues for separation and over-distortion of semantics within the IKT module, which together harm discriminative ability.

We further conduct a parameter sensitivity analysis on the proximal strength coefficient $\varepsilon$ in the SAM module. Specifically, we perform experiments on the *Real → Painting* domain adaptation scenario, varying $\varepsilon$ within the range $\{0.0, 0.01, 0.05, 0.1, 0.5, 1, 10\}$. As shown in Figure 6 (e)–(f) and

(k)–(l), the performance of our model remains stable under moderate changes of $\varepsilon$. However, when $\varepsilon$ becomes excessively large, the proximal regularization term dominates over the sparse assignment matching, which may lead to performance degradation. Therefore, we empirically set $\varepsilon = 0.5$ in all experiments.

## A.12 Computational Complexity Analysis

We profile the computational overhead of each component on an RTX 4090 with batch size 128 and input resolution $224 \times 224$. HCS performs one forward DFT/IDFT pair per image (via `torch.fft`), applies a binary frequency-plane mask, runs a cosine-similarity scan against previous-stage prototypes, and fits a 1D GMM on scalar scores; the cost is dominated by FFTs, with masking/cosine/GMM negligible. For the known-like subset, SAM solves a proximal OT subproblem with a few lightweight dual updates over $(\psi, \varphi)$ and a closed-form refresh of $\gamma$; computing the cost matrix and sparse projection is minor compared to FFT work. For the unknown-like subset, IKT reuses low-frequency statistics from the previous stage to sample perturbed styles, reconstructs low-frequency spectra and fuses them with the original high-frequency part before an inverse DFT, then compares two feature views (original vs. style-transferred) to all prototypes to build Plackett–Luce listwise distributions and a KL divergence; the extra compute is mainly the single IFFT per unknown sample, with ranking terms lightweight. Overall, the framework adds approximately **5.2 GFLOPs** and increases per-iteration time by $\sim 0.65$ s, and thanks to GPU-accelerated FFTs, efficient proximal OT updates, and amortized clustering, the overhead remains manageable and scales well to large open-world streams.

Table 15: Stage-wise clustering accuracy (%) of all methods on the CUB-C dataset under the dynamic domain-incremental setting. At each stage, new domains are progressively introduced (Stage 1: Gaussian, Shot, Impulse Noise; Stage 2: Zoom Blur, Snow, Frost; Stage 3: Fog, Speckle, Spatter). We report the accuracy on all classes (All), known classes (Old), and novel classes (New) at each stage, as well as the average across all stages.

| Methods | Stage 1 | | | Stage 2 | | | Stage 3 | | | Average | | | Stage 1 | | | Stage 2 | | | Stage 3 | | | Average | | |
|---|---|---|---|---|---|---|---|---|---|---|---|---|---|---|---|---|---|---|---|---|---|---|---|---|
| | All | Old | New | All | Old | New | All | Old | New | All | Old | New | All | Old | New | All | Old | New | All | Old | New | All | Old | New |
| | | | | | | | | Original | | | | | | | | | | | CUB-C Corrupted | | | | | | |
| GCD | 31.1 | 49.4 | 24.1 | 28.6 | 46.2 | 22.1 | 24.9 | 43.2 | 20.1 | 28.2 | 46.3 | 22.1 | 23.8 | 43.0 | 17.2 | 26.6 | 46.0 | 20.0 | 25.5 | 44.5 | 19.0 | 25.3 | 44.5 | 18.7 |
| SimGCD | 28.2 | 45.1 | 22.3 | 24.8 | 43.0 | 20.0 | 22.6 | 40.9 | 17.2 | 25.2 | 43.0 | 19.8 | 20.7 | 39.8 | 15.1 | 23.3 | 42.4 | 17.7 | 22.0 | 40.9 | 16.4 | 22.0 | 41.0 | 16.4 |
| SPTNet | 29.5 | 46.0 | 23.3 | 26.8 | 43.9 | 20.3 | 23.5 | 41.1 | 18.5 | 26.6 | 43.7 | 20.7 | 22.4 | 41.3 | 15.1 | 25.1 | 44.1 | 18.5 | 23.6 | 43.0 | 16.8 | 23.7 | 42.8 | 16.8 |
| RLCD | 30.0 | 48.4 | 24.6 | 27.7 | 45.2 | 22.7 | 25.8 | 42.6 | 19.9 | 27.8 | 45.4 | 22.4 | 23.3 | 42.6 | 16.6 | 26.1 | 45.3 | 19.3 | 25.0 | 44.1 | 18.1 | 24.8 | 44.0 | 18.0 |
| G&M | 17.8 | 36.4 | 12.8 | 14.9 | 32.7 | 9.2 | 12.9 | 29.3 | 5.6 | 15.2 | 32.8 | 9.2 | 10.6 | 29.6 | 5.1 | 13.9 | 31.8 | 7.4 | 12.4 | 30.7 | 6.4 | 12.3 | 30.7 | 6.3 |
| PA-CGCD | 29.4 | 48.1 | 23.9 | 26.9 | 44.7 | 21.6 | 24.6 | 42.2 | 18.7 | 27.0 | 45.0 | 21.4 | 22.5 | 41.8 | 15.7 | 25.3 | 44.6 | 18.5 | 24.2 | 43.5 | 16.8 | 24.0 | 43.3 | 17.0 |
| DEAN | 29.9 | 48.9 | 24.5 | 28.2 | 45.7 | 21.2 | 24.8 | 42.5 | 19.4 | 27.6 | 45.7 | 21.7 | 23.7 | 43.2 | 15.8 | 26.1 | 46.4 | 18.1 | 24.8 | 44.7 | 16.8 | 24.9 | 44.8 | 16.9 |
| PromptCCD | 31.1 | 49.8 | 26.4 | 28.7 | 46.5 | 22.8 | 26.3 | 43.5 | 20.1 | 28.7 | 46.6 | 23.1 | 24.6 | 43.2 | 17.4 | 27.3 | 46.3 | 20.4 | 26.2 | 44.6 | 18.8 | 26.0 | 44.7 | 18.9 |
| VB-CGCD | 34.7 | 50.2 | 27.2 | 31.9 | 48.5 | 25.4 | 29.3 | 46.5 | 22.4 | 32.0 | 48.4 | 25.0 | 26.9 | 45.4 | 19.5 | 29.6 | 48.4 | 22.4 | 28.4 | 46.6 | 21.1 | 28.3 | 46.8 | 21.0 |
| **PRISM** | **51.1** | **66.7** | **47.6** | **48.9** | **64.4** | **43.9** | **47.0** | **62.4** | **40.5** | **49.0** | **64.5** | **44.0** | **41.9** | **59.1** | **35.1** | **45.1** | **61.9** | **38.2** | **43.6** | **60.2** | **36.5** | **43.5** | **60.4** | **36.6** |

## A.13 Experiments on Dynamic Domain-Incremental Setting

To further evaluate the performance of our proposed algorithm in dynamic domains, we conducted an additional domain-incremental experiment. Specifically, we trained on the CUB-C dataset over three stages, where new domains were progressively introduced at each stage. In Stage 1, we introduced three types of perturbations: Gaussian Noise, Shot Noise, and Impulse Noise. In Stage 2, we further incorporated Zoom Blur, Snow, and Frost. In Stage 3, additional perturbations including Fog, Speckle, and Spatter were introduced. In this way, we simulated a realistic dynamic and non-stationary data stream scenario, where each stage may involve domains unseen in the previous stage. As shown in Table 15, we observed that under this more challenging dynamic-domain setting, our model still achieved significant performance improvements and substantially outperformed other competing methods, further demonstrating the robustness and effectiveness of the proposed approach.

## A.14 Visualization of HCS Separation Across Fine-Grained Datasets

We further visualize the separation behavior of HCS on several fine-grained datasets, as illustrated in Fig. 7. Across most datasets, the separation scores of known and unknown samples show a certain degree of discrepancy, indicating that HCS can reliably distinguish the two groups using

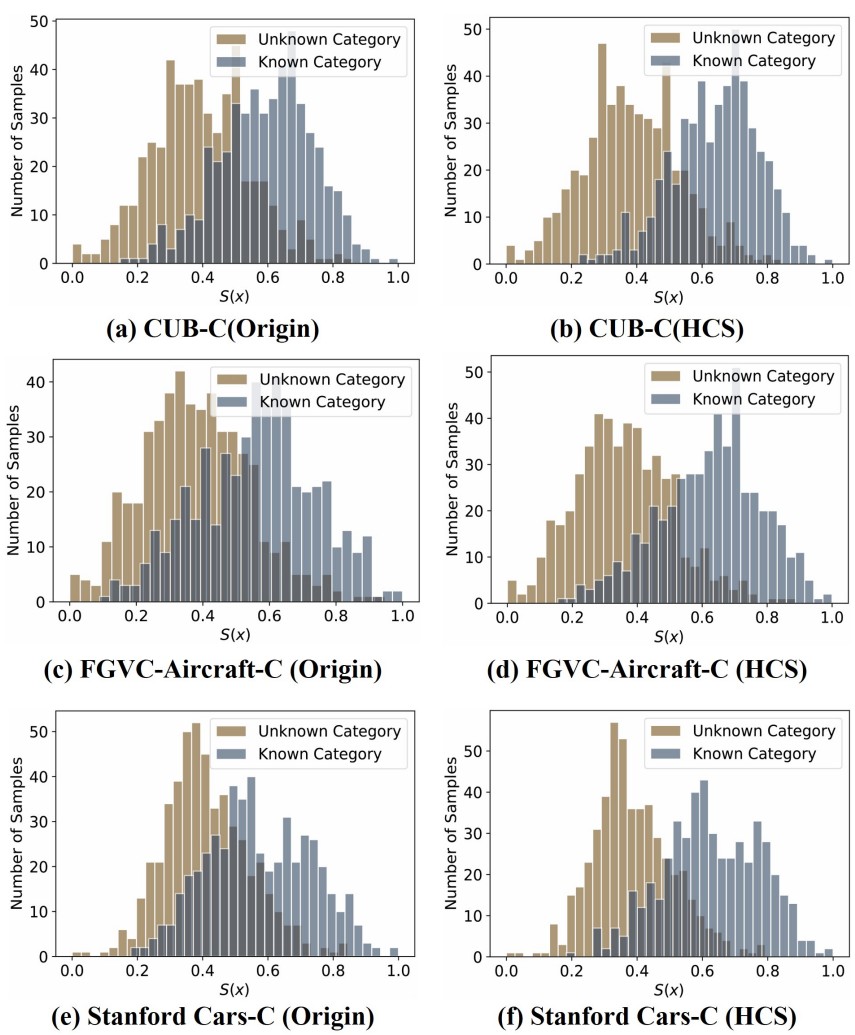

Figure 7: Comparison of separation performance across multiple fine-grained datasets.

high-frequency information. Even on challenging fine-grained benchmarks—such as Stanford Cars-C—where categories share strong visual resemblance, the score distributions still exhibit an approximate bimodal pattern, demonstrating that meaningful separation can be achieved.

These observations highlight two important facts. First, high-frequency cues consistently provide better separation than raw images, as the removal of low-frequency domain biases makes the remaining semantic differences more distinguishable. Second, although the separability on fine-grained datasets is naturally reduced due to subtle inter-class variations, high-frequency decomposition still improves the separation of known and unknown samples.

Overall, the results in Fig. 7 confirm that high-frequency cues offer more reliable and robust separation than original images across diverse datasets, including challenging fine-grained scenarios.

## A.15 BAD CASE ANALYSIS

Although the proposed HCS module provides robust separation between known and unknown samples across diverse datasets and corruption types, certain challenging scenarios may still lead to reduced performance. A representative example arises under structural blurring—such as zoom blur—which directly suppresses the semantic high-frequency cues that HCS relies on.

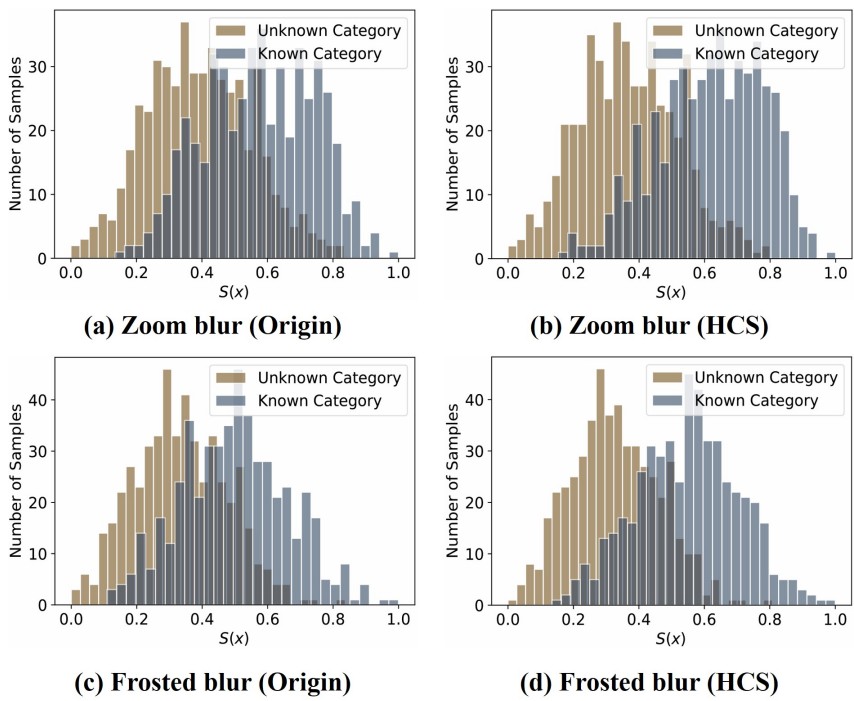

(a) **Zoom blur (Origin)**  (b) **Zoom blur (HCS)**

(c) **Frosted blur (Origin)**  (d) **Frosted blur (HCS)**

Figure 8: Illustration of a challenging case, where structural blurring suppresses semantic high-frequency cues and weakens separation, yet HCS still outperforms raw-image representations.

Structural blur smears object boundaries, attenuates fine textures, and destroys edge sharpness, thereby weakening the discriminative high-frequency structures essential for our separation mechanism. As illustrated in Appendix Fig. 8, the extracted high-frequency representations become less informative in these cases, causing the separation scores of known and unknown samples to move closer.

Nevertheless, even under such challenging corruptions, HCS consistently outperforms using raw images. This is because the high-frequency decomposition still removes domain-specific low-frequency biases, and the preserved semantic structures—although partially degraded—remain more discriminative than full-spectrum representations. The resulting separation curve demonstrates that the degradation introduced by structural blur affects both HCS and raw images, but the impact is notably smaller for HCS.

Overall, these bad cases highlight an inherent limitation: when corruptions significantly destroy semantic high-frequency content, the separability achievable by HCS naturally decreases. This observation suggests a promising future direction—integrating structure-preserving or deblurring techniques to further enhance the robustness of high-frequency-based separation under severe image degradations.

## A.16 COMPARISON WITH EXISTING DOMAIN-SHIFT GCD METHODS

Although prior domain-shift GCD approaches such as HiLo (Wang et al., 2024a) and CDAD-Net (Rongali et al., 2024) cannot be directly adapted to the same *domain-shift + CCD* setting considered in our work, our framework can still be applied to the standard cross-domain GCD scenario for fair comparison. Following the experimental protocol of HiLo, we evaluate our method alongside HiLo and CDAD-Net under the cross-domain GCD setting (note that this is *not* the proposed OW-CCD scenario). As shown in the results, our approach outperforms both HiLo and CDAD-Net in most cases, demonstrating that the proposed method is flexible and effective even when applied to conventional cross-domain GCD tasks.

Table 16: Clustering results of various methods on the DomainNet benchmark. Each experiment uses *Real* as the source domain, with one of *Painting*, *Sketch*, *Quickdraw*, *Clipart*, or *Infograph* serving as the target. Clustering accuracy is reported for both domains.

| Methods | Real + Painting | | | | | | Real + Sketch | | | | | | Real + Quickdraw | | | | | | Real + Clipart | | | | | | Real + Infograph | | | | | |
| | Real | | | Painting | | | Real | | | Sketch | | | Real | | | Quickdraw | | | Real | | | Clipart | | | Real | | | Infograph | | |
| | All | Old | New | All | Old | New | All | Old | New | All | Old | New | All | Old | New | All | Old | New | All | Old | New | All | Old | New | All | Old | New | All | Old | New |
| CDAD-Net | 63.6 | 77.9 | 56.3 | 38.4 | 38.4 | 37.5 | 61.9 | 76.3 | 52.1 | 17.3 | 20.9 | 15.9 | 48.5 | 66.5 | 36.7 | 6.4 | 5.6 | 7.3 | 61.3 | 77.0 | 53.1 | 25.2 | 31.9 | 19.0 | 56.5 | 68.0 | 47.1 | 11.8 | 15.6 | 9.4 |
| HiLo | 64.4 | 77.6 | 57.5 | 42.1 | 42.9 | 41.3 | 63.3 | 77.9 | 55.9 | 19.4 | 22.4 | 17.1 | 58.6 | 76.4 | 52.5 | 7.4 | 6.9 | 8.0 | 63.8 | **77.6** | 56.6 | 27.7 | 34.6 | 21.7 | 64.2 | **78.1** | 57.0 | 13.7 | 16.4 | 11.9 |
| PRISM | **68.7** | 77.8 | **63.3** | **47.2** | **46.8** | **45.8** | **68.7** | 78.2 | **63.0** | **23.8** | **24.9** | **22.8** | **61.8** | 77.5 | **57.1** | **9.0** | **7.3** | **9.4** | **67.2** | 77.4 | **62.0** | **30.4** | **36.4** | **27.6** | **69.1** | 77.6 | **61.2** | **16.7** | **18.7** | **14.3** |

Table 17: Clustering results of various methods on the SSB-C benchmark. For each dataset (CUB, Scars, and FGVC), the clean set serves as the source domain, while its corrupted counterpart is treated as the target domain. Clustering accuracy is reported for both domains.

| Methods | CUB-C | | | | | | Scars-C | | | | | | FGVC-C | | | | | |
| | Original | | | Corrupted | | | Original | | | Corrupted | | | Original | | | Corrupted | | |
| | All | Old | New | All | Old | New | All | Old | New | All | Old | New | All | Old | New | All | Old | New |
| CDAD-Net | 40.4 | 38.9 | 39.3 | 37.7 | 39.1 | 34.2 | 32.1 | 42.9 | 32.2 | 28.8 | 35.6 | 21.4 | 33.8 | 35.5 | 31.2 | 27.8 | 29.6 | 25.6 |
| HiLo | 56.8 | 54.0 | 60.3 | 52.0 | 53.6 | 50.5 | 39.5 | 44.8 | 37.0 | 35.6 | 42.9 | 28.4 | 44.2 | 50.6 | 47.4 | 31.2 | 29.0 | 33.4 |
| PRISM | **60.1** | **58.7** | **63.1** | **56.2** | **55.1** | **54.9** | **44.0** | **47.4** | **40.6** | **40.1** | **43.5** | **34.5** | **47.9** | **55.1** | **51.8** | **35.7** | **31.8** | **39.1** |

## A.17 SOCIETAL IMPACT AND FUTURE DIRECTIONS

We study Open-World Continual Category Discovery (OW-CCD), which reflects the reality of dynamic data streams where category distributions are non-stationary and new classes emerge over time. This research has broad societal implications, as it equips AI systems to adaptively recognize evolving concepts in real-world scenarios such as medical diagnostics, ecological monitoring, and social media moderation, thereby enhancing their reliability and fairness in open environments. However, limitations remain: (1) the open world introduces vast distributional shifts and complex dynamics, where current models still struggle to maintain stable performance; and (2) most existing work relies on single-modality data, whereas extending to multi-modal OW-CCD is crucial to fully exploit diverse real-world signals (e.g., combining image, text, and sensor data) for robust knowledge discovery. These challenges highlight promising future directions, motivating research into more resilient algorithms and multi-modal learning frameworks for open-world continual discovery.

## A.18 ETHICS STATEMENT

This research does not involve human participants, animal subjects, or the use of sensitive personal data, nor does it present any potentially harmful applications. All experiments are conducted on publicly available benchmark datasets that are properly licensed for academic use. The authors are committed to adhering to ethical research standards and to promoting fairness, transparency, and responsible development of AI technologies.

## A.19 USE OF LLMS

During the preparation of this manuscript, we made limited use of publicly available large language models (LLMs) solely to assist with English writing. All technical content, including the formulation of ideas, design of methodologies, implementation of experiments, and interpretation of results, was entirely conceived and written by the authors without LLM involvement. The role of LLMs was strictly confined to stylistic and linguistic improvements, comparable to grammar- or spell-checking software. No novel research insights, data, or analyses were generated by LLMs, and all scientific claims and results presented in this work remain the sole responsibility of the authors.

## A.20 REPRODUCIBILITY

To ensure reproducibility, we provide a comprehensive description of the model design in Section 3 and detailed experimental settings in Section 4.1. Furthermore, we include the full pseudocode of the proposed PRISM framework in Appendix A.3, clearly outlining its main components and training flow. These details collectively enable faithful reimplementation and verification of our results.

