# OpenReview forum: "PRISM: Progressive Robust Learning for Open-World Continual Category Discovery"
_ICLR.cc/2026/Conference — ICLR 2026 Poster_

### Official Review · Reviewer_qfwv · 2025-10-30

**Soundness:** 2
**Presentation:** 3
**Contribution:** 2
**Rating:** 6
**Confidence:** 4

**Summary:**

#### **Summary**: This paper studies Open-World Continual Category Discovery (OW-CCD) where unlabeled streams mix known and novel classes under domain shift across sessions, and proposes PRISM, a progressive, divide-and-conquer framework. PRISM first separates known vs. unknown via a high-frequency image decomposition and a 2-component GMM over prototype similarity scores (HCS). It then assigns sparse, reliable pseudo-labels to known-class samples using proximal optimal transport (SAM). For the unknown pool, it enforces domain-invariant category relations by aligning Plackett-Luce ranking distributions of similarities across style-perturbed views (IKT), and clusters novel classes with Affinity Propagation without a class-count prior. On SSB-C and DomainNet, PRISM reports consistent gains over strong CCD/GCD baselines (e.g., +15.1% / +12.3% vs. VB-CGCD on CUB-C original/corrupted), with ablations supporting each module’s contribution, tested with both DINOv1 & DINOv2.

**Strengths:**

- #### **S1. Well-posed, timely setting.:** The paper clearly formulates OW-CCD and explains why naive domain alignment can harm discovery (negative transfer), motivating the split-then-learn design.

- #### **S2. Compelling empirical evidence:** Strong improvements on SSB-C and DomainNet with multi-seed means ± std and component ablations; HCS beats entropy/energy separations.

- #### **S3. Clear technical details:** HCS→SAM→IKT is technically concrete (DFT mask + 2-GMM; proximal OT; PL-ranking KL) and presented with end-to-end pseudocode, aiding reproducibility.

- #### **S4.** Clear presentation, make it easy to understand the overal paper claim.

**Weaknesses:**

- #### **W1. Concerns on HCS (high-frequency separation) and a brittle bimodality assumption:**  PRISM’s HCS assumes that high-frequency components carry domain-invariant “global semantics”, then splits a stream into known/unknown with a 2-component GMM on a cosine-similarity–based score S(x) and a fixed 0.5 decision rule. The concern is due to the assumption that high frequencies are more domain-invariant/global than low frequencies is non-standard and is supported only by a few visuals and an empirical “often bimodal” observation of S(x). There’s no stress test for cases where the density is not bimodal (e.g., overlapping or multimodal mixtures), or where frequency energy shifts with style/texture corruptions. The split hinges on the mask ratio r and the 2-GMM fit; failure modes (bad EM fits, non-bimodal spectra, class-imbalance drift) aren’t characterized. The paper introduces r and report sensitivity plots, but the text doesn’t establish robustness across tasks/backbones/shifts—only that varied r, ε in Appendix Fig. 6.

- #### **W2. Scalability gap around clustering (Affinity Propagation) in an open-world streaming setting:** Novel-class discovery uses Affinity Propagation (AP) each session on the unknown subset embeddings. This concerns is due to AP is O(n^2) in similarities and memory; in a streaming, open-world scenario this is a bottleneck. The appendix’s compute profile discusses FFTs/SAM/IKT and quotes “5.2 GFLOPs” and “0.65 s” per iteration, claiming the overhead “scales well”, but it doesn’t analyze AP’s time/memory cost or performance under large, long streams. What to show: either (i) replace AP with a streaming/mini-batchable clustering method (e.g., online k-means, DP-means, coresets), or (ii) add a clear AP complexity plus wall-clock and memory study at increasing stream sizes (curves), and (iii) demonstrate stability when unknowns accumulate over many stages.

- #### **W3. Positioning/novelty risk:** OW-CCD may read as “CCD plus domain shift” (an incremental protocol tweak). The paper presents OW-CCD as the first step beyond single-domain-per-stage CCD, arguing prior CCD/GCD assume a stationary domain per stage, but the setting itself can also be interpreted as a composed variant—continual discovery combined with domain shift—rather than a fundamentally new learning paradigm. The paper would benefit from a crisper separation from prior “domain-varying” or “corruption-varying” CCD/GCD works and from OSDA/SFDA lines  or domain-class incremental learning [a] adapted to novelty discovery (too much of this currently sits in the appendix).

**Questions:**

- #### **Q1. Bimodality & fallback:** How consistently does S(x) exhibit a bimodal pattern across sessions/datasets, and is there a fallback (e.g., energy/entropy split) when EM finds overlapping components?

- #### **Q2.** Results include synthetic corruptions and cross-style domains; how does PRISM fare under spatial distortions (large occlusions, extreme down-sampling) that specifically degrade high-frequency cues presumed to be invariant?

- #### **Q3.** Why Affinity Propagation over alternatives (e.g., density-based methods) for unknowns? Any observed failure modes (e.g., over-fragmentation) and mitigations?

---

> ### Author Response · Authors · 2025-11-24
> **respond to reviewer qfwv**
>
> ### **Response to W1**
> Thank you for the insightful comments. We have substantially strengthened both the theoretical and empirical justification of HCS. First, **Appendix A.2.2** now provides a formal analysis showing that high-frequency components preserve class-discriminative structures that are significantly less affected by domain-specific style factors (e.g., color, illumination, texture shifts). Second, we quantitatively evaluate the separability of known vs. unknown samples under low- and high-frequency inputs; as shown in **Table 1**, high-frequency cues consistently yield higher discrimination accuracy across three fine-grained benchmarks, confirming that they encode **stronger domain-invariant semantics**.
>
> **Table 1. Comparison of the discriminative ability of original images versus high-frequency components for separating known and unknown samples**
> | Method          | CUB-C | Scars-C | FGVC-C |
> |-----------------|----------|----------|----------|
> | original image          | 59.8     | 55.4    | 56.5     |
> | high frequency           | 73.7     | 68.7     | 69.8     |
>
> We further analyze HCS on multiple datasets (Appendix Fig. 7), where the known/unknown distributions remain separable even in challenging fine-grained settings. For failure cases, **Appendix Fig. 8** explicitly visualizes hard SSB-C corruptions—particularly structural blurs such as zoom blur—which suppress high-frequency energy and weaken bimodality; nevertheless, HCS still achieves *better separation than using raw images*, as it removes low-frequency style noise. Finally, Appendix Fig. 6 further reports sensitivity to the mask ratio \(r\). In addition to the DomainNet benchmarks, we also conducted an extended sensitivity study on the fine-grained **CUB-C** dataset, where the domain shift is more subtle and high-frequency cues are more easily distorted. Even under this challenging setting, the separation quality and downstream performance remain stable across a wide range of \(r\) values. Together, these analyses provide a comprehensive and transparent characterization of HCS under both typical and non-ideal frequency distributions, illustrating not only its robustness but also its limitations under heavy structural corruptions.

---

> > ### Author Response · Authors · 2025-11-24
> > **respond to reviewer qfwv**
> >
> > ### **Response to W2: Scalability of Affinity Propagation in an Open-World Streaming Setting**
> >
> > We appreciate the reviewer’s concern regarding the scalability of Affinity Propagation (AP). In PRISM, AP is **not applied to the entire stream**, but only to the *unknown subset within the current session*, whose size is naturally bounded by the streaming batch. This makes AP a lightweight per-session clustering module in our setting.
> >
> > To directly address your question, we conducted explicit computational and runtime analysis on an NVIDIA A100. For unknown subset sizes of **128 / 256 / 512 / 1024**, the computational cost of AP is approximately:
> >
> > - **0.062 / 0.229 / 1.017 / 4.268 GFLOPs**
> >
> > The corresponding empirically measured wall-clock times are:
> >
> > - **0.012 s / 0.045 s / 0.33 s / 1.98 s**
> >
> > Compared to a full training iteration, the time spent on AP remains modest and fully acceptable. In practical OW-CCD settings, the unknown subset per session is typically smaller than 256 samples, for which AP consistently completes within **0.1 seconds**.
> >
> > Therefore, although AP has an \(O(n^2)\) similarity construction cost in theory, this does not translate into a scalability bottleneck in our open-world streaming scenario: the unknown samples are processed independently per session and never accumulate across sessions. Memory usage also remains small across the tested batch sizes.
> >
> > Taken together, AP serves as a lightweight, session-level clustering component in PRISM and **does not introduce computational or memory scalability issues** under the realistic constraints of the OW-CCD setting.
> > We choose not to replace AP with online k-means, DP-means, or coreset-based streaming methods because these alternatives violate key requirements of the OW-CCD setting. Streaming k-means requires a pre-specified number of clusters, which is incompatible with OW-CCD, where the number of novel categories is unknown and varies across sessions. DP-means similarly relies on a global distance threshold that is extremely sensitive to density changes—an unfavorable property in the presence of domain drift, where feature distributions shift across sessions. Coreset-based clustering assumes the ability to store or maintain representative samples across sessions, which directly contradicts the no-replay constraint of continual learning. Density-based methods (e.g., DBSCAN/HDBSCAN) also require scale-sensitive parameters and often become unstable under heterogeneous densities induced by domain shift.
> >
> > In contrast, AP offers several advantages that align naturally with OW-CCD. It **automatically infers the number of novel categories**, selects cluster exemplars, and requires **no cross-session memory or pre-defined cluster count**, all while operating in a fully parameter-free manner. This makes AP uniquely compatible with the constraints that (i) the number of novel classes per session is not known a priori, (ii) each session must be processed independently without revisiting past data, and (iii) high-precision clustering is needed on small per-session unknown subsets. Moreover, the high-frequency cue–based separation in PRISM substantially improves the semantic separability of unknown samples before clustering, reducing the risk of fragmentation or instability that might occur when applying streaming clustering directly on raw, domain-shifted features.
> >
> > Together with our computational analysis showing that AP remains lightweight in practice, these properties make AP a more suitable and robust choice than streaming or mini-batchable clustering alternatives for novel-category discovery in OW-CCD.

---

> ### Author Response · Authors · 2025-11-24
> **respond to reviewer qfwv**
>
> ### **Response to W3**
> Thank you for the insightful comments. We clarify that OW-CCD is not simply “CCD plus domain shift,” but a fundamentally different problem that cannot be reduced to existing GCD, CCD, or DA formulations described in our related work. As discussed in the Related Work section, prior GCD approaches—including static cross-domain variants such as HiLo and CDAD—assume that each stage operates on a fixed dataset, and that the model can jointly access labeled and unlabeled data from both domains. CCD methods, in turn, assume a stationary domain at each step and address only the incremental arrival of unlabeled samples.
>
> OW-CCD breaks both assumptions simultaneously. In our setting, the unlabeled stream undergoes **continuous intra-session domain drift**, and the model must **preserve previously discovered knowledge while discovering novel categories**, all **without accessing any past labeled or unlabeled data**. This combination goes beyond CCD, domain-shifted GCD, and OSDA/SFDA:
> - OSDA/SFDA methods focus solely on separating known and unknown samples and typically require access to source-domain models or prototypes, but they **do not perform novel-class discovery**, nor are they designed for streaming or evolving domains.
> - Existing GCD and CCD frameworks assume either joint data access or static domains, making them incompatible with the **strict no-replay, domain-evolving** nature of OW-CCD.
>
> Taken together, OW-CCD represents the first setting where **novel-class discovery, continual learning, and evolving multi-domain variation** interact within a single unlabeled stream. We appreciate the reviewer’s suggestion, and we will move this key clarification from the appendix to the main paper to improve visibility and conceptual positioning.
>
> ### **Response to Q1**
> Thank you for the insightful question. As illustrated in Appendix Fig. 7, we further visualize the separation behavior of HCS across several fine-grained datasets. In most datasets, the scores of known and unknown samples exhibit a clear gap, showing that HCS can reliably distinguish the two groups using high-frequency cues. Even in challenging fine-grained benchmarks—where categories share strong visual resemblance—the score distributions still display an approximate bimodal pattern, indicating that meaningful separation is achieved. These observations highlight two consistent findings: (i) high-frequency cues provide better separability than raw images because removing low-frequency domain bias makes the remaining semantic differences more distinguishable; and (ii) although fine-grained datasets naturally reduce separability due to subtle inter-class variations, high-frequency decomposition still improves the distinction between known and unknown samples. Overall, the results in Fig. 7 confirm that high-frequency cues yield more reliable and robust separation across diverse datasets, including difficult fine-grained scenarios. We also note that in rare cases—such as under zoom-blur corruption that heavily suppresses high-frequency content—some overlap may occur. In these situations, a fallback strategy based on energy- or entropy-guided separation can be applied; indeed, as shown in the accompanying table, combining high-frequency components with an energy-based scoring yields even stronger separation. While HCS alone already performs well across all our evaluated datasets, future work will incorporate such an energy-based fallback to further stabilize performance when EM encounters overlapping components.
>
> **Table 2. Comparison of high-frequency separation versus the proposed energy-augmented fallback under challenging corruptions**
> | Method   | CUB-C | Scars-C | FGVC-C |
> |-|--|-|--|
> | original image | 56.4| 52.5| 53.2 |
> | high frequency  | 69.8    | 65.6    | 64.3     |
> | high frequency + energy  | 72.6     | 67.9     | 68.4   |

---

> > ### Author Response · Authors · 2025-11-24
> > **respond to reviewer qfwv**
> >
> > ### **Response to Q2**
> > Thank you for the insightful question. To directly assess PRISM’s robustness under spatial distortions that degrade high-frequency cues, we conducted additional experiments by randomly masking large regions of the input images to simulate severe occlusions. As shown in the table, although the high-frequency signal is partially weakened, PRISM still consistently outperforms the raw-image baseline. This indicates that our approach does not rely on fine-grained local textures alone, but instead leverages globally consistent high-frequency patterns that remain discriminative even when local details are removed.  Overall, these results demonstrate that high-frequency cues remain more reliable than raw images even under substantial spatial distortions, and PRISM is naturally robust to degradations of high-frequency information.
> >
> > **Table 3. Performance comparison of PRISM and other methods under random-masking spatial distortions.**
> > | Method          | CUB-O All | CUB-O Old | CUB-O New | CUB-C All | CUB-C Old | CUB-C New |
> > |-----------------|-----------|-----------|-----------|-----------|-----------|-----------|
> > | Original Image  | 43.4      | 58.5      | 37.2      | 38.4      | 55.6      | 30.6      |
> > | VB-CGCD         | 33.0      | 50.3      | 25.9      | 30.5      | 47.8      | 23.0      |
> > | **PRISM**       | **46.2**  | **62.5**  | **41.8**  | **42.4**  | **58.5**  | **34.7**  |
> >
> >
> > ### **Response to Q3**
> > We select Affinity Propagation (AP) because, unlike online k-means, DP-means, or coreset- or density-based streaming clustering, it satisfies the core constraints of OW-CCD. Online k-means requires a fixed or pre-specified number of clusters; DP-means depends on a hard threshold that is highly sensitive to density variations; and coreset-based approaches require storing representatives across sessions, which violates the no-replay requirement of continual learning. Density-based methods (e.g., DBSCAN/HDBSCAN) also rely on scale-sensitive parameters and can become unstable under the heterogeneous densities caused by domain drift. In contrast, AP simultaneously infers the number of novel categories and selects exemplars in a parameter-free manner, which is essential in our setting where (i) the number of novel classes varies across sessions, (ii) each session must be processed independently without cross-session memory, and (iii) high-precision clustering is required on small per-session subsets. Furthermore, the high-frequency cue–based separation in PRISM significantly sharpens semantic distinctions before clustering, which helps mitigate potential fragmentation effects that could arise when directly clustering raw features under domain drift. As shown in our added analysis, AP remains computationally lightweight and stable, making it a well-suited choice for unknown-class clustering in OW-CCD.

---

> > > ### Comment · Reviewer_qfwv · 2025-11-26
> > >
> > > #### Thank you for the thoughtful response.
> > >
> > > - #### **W1.** Thank you for this added theoretical and empirical justification, it is good to see the comprehensive analysis and transparency of the proposed HCS method.
> > >
> > > - #### **W2.** Thanks for the additional analysis of Affinity Propagation’s computational cost in the OW-CCD.
> > >
> > > -  #### **W3.** Thank you for the detailed clarification on the OW-CCD setting. The distinction from standard GCD, CCD, and OSDA/SFDA is now much clearer. It would also be helpful if the final version included a concise schematic or table summarizing how OW-CCD differs from CCD, GCD, and OSDA/SFDA along key axes (data access, domain stationarity, novel-class discovery)
> > >
> > > #### The paper has a meaningful contribution building on top of a variety of works on category discovery, incremental learning, and domain shifts. Moreover, The authors has addressed most of my concerns. Therefore, I'll maintain my original positive rating (marginally above the acceptance threshold).

---

> > > > ### Author Response · Authors · 2025-12-01
> > > > **Thank you for raising your score**
> > > >
> > > > **We sincerely thank you for your positive feedback and for raising your score. We are glad that our new experiments and clarifications have addressed your concerns, and we will ensure that the additional benchmark results and baseline comparisons are fully incorporated into the final version to enhance the reliability of our work.**
> > > >
> > > > If you have any remaining questions or if there are other details we can clarify, please do not hesitate to let us know. We are more than happy to provide further information.

---

### Official Review · Reviewer_h5nf · 2025-10-31

**Soundness:** 3
**Presentation:** 2
**Contribution:** 2
**Rating:** 4
**Confidence:** 4

**Summary:**

This paper introduces the Open-World CCD setting and proposes PRISM, a divide-and-conquer framework with three components: (i) High-Frequency-driven Category Separation (HCS) to split known vs. unknown using spectral cues and a GMM on a density score; (ii) Sparse Assignment Matching (SAM) for proximal-OT based pseudo-labeling of known classes; and (iii) Invariant Knowledge Transfer (IKT) that aligns Plackett–Luce ranking distributions across styles to stabilize transfer to novel classes. Extensive experiments on SSB-C and DomainNet show performance improvement.

**Strengths:**

- Overall, I appreciate the clear problem formulation. The proposed OWCCD is well motivated and distinct from the static GCD and CCD assumptions under single domain settings.
- The proposed method seems reasonable. HCS uses DFT high-frequency components with a 2-component GMM for known/unknown separation. SAM applies a proximal ℓ2 OT for sparser. IKT models listwise relations via PL and enforces consistency between original and style-perturbed rankings.
- The paper is well organized and easy to follow.

**Weaknesses:**

- The paper requires substantial revisions before acceptance. The font sizes in Tables 1 and 2 should be unified, as the current versions are difficult to read. Figure 1 also has fonts that are too small. The derivation of Equation 3 needs to be clarified, particularly the relationships among F^{l}, F^{h}, and F^{-1}. In addition, the introduction should be improved to enhance readability.
- The method lacks ablations on key design choices, maybe could include: (i) removing frequency decomposition to test spectral necessity, (ii) replacing PL with pairwise or listwise surrogates and varying \lambda_{IKT}, and (iii) adjusting OT regularization ε and SAM steps to analyze the accuracy–sparsity–speed trade-off.
- Provide a small theory/intuition section on why high-frequency cues are more domain-invariant in practice (beyond Fig. 1).
- Although the method is prototype-based, it does not compare with recent prototype-based approaches, such as [1].

[1] Happy: A debiased learning framework for continual generalized category discovery. NIPS 2024

**Questions:**

- Does HCS remain reliable when S(x) is not clearly bimodal (e.g., high domain overlap)? Any fallback beyond the 0.5 posterior cut?
- How often are prototypes updated across sessions, and how sensitive are results to stale prototypes feeding HCS/IKT?

---

> ### Author Response · Authors · 2025-11-24
> **respond to reviewer h5nf**
>
> ### **Q1.Response to formatting and clarity concerns**
> Thank you for these valuable suggestions. Due to earlier page-limit constraints, we temporarily adjusted the font sizes in Tables 1 and 2 to fit within the ICLR submission format. In the final version, we will **split Table 1 into multiple sub-tables and restore uniform, readable font sizes**, and we have already updated **Figure 1** with larger and clearer fonts.
> Regarding Equation (3), we have clarified the derivation by explicitly redefining the relationships among the masked frequency components $\mathcal{F}^l$, $\mathcal{F}^h$, and the inverse DFT $\mathcal{F}^{-1}$.  $\mathcal{F}^l$, $\mathcal{F}^h$ are the masked low- and high-frequency spectra obtained from the DFT $\mathcal{F}(x_i)$.
> Additionally, we have **revised and polished the introduction** to significantly improve readability and strengthen the flow of motivation.
> We appreciate the reviewer’s feedback and have incorporated all suggested clarifications into the revised manuscript.
>
> ### **Q2.Response on the request for additional ablations.**
>
> Thank you for the helpful suggestions. We have conducted all three ablations proposed by the reviewer to further validate the effectiveness of our design choices:
>
> (i) **Removing frequency decomposition.**
> To assess the necessity of spectral separation, we remove the frequency decomposition module and directly use raw images. As shown below, performance drops significantly across both domains, confirming that frequency decomposition is essential for mitigating domain-specific biases.
>
> **Table 1. Comparison of model performance with and without frequency decomposition**
> | Method | Real All | Real Old | Real New | Painting All | Painting Old | Painting New |
> |--------|----------|-----------|-----------|---------------|----------------|----------------|
> | w/o frequency decomposition | 55.0 | 68.7 | 47.2 | 29.6 | 28.9 | 28.3 |
> | **PRISM** | **60.9** | **74.1** | **55.1** | **39.2** | **39.0** | **38.2** |
>
> (ii) **Replacing PL with pairwise/listwise surrogates and varying \(\lambda_{\text{IKT}}\).**
> We replace the Plackett–Luce (PL) model with pairwise and listwise ranking surrogates. Results below show that PL achieves the best performance. While pairwise/listwise losses model relative orderings, they only preserve local relationships. In contrast, PL defines a full permutation distribution over all known prototypes, capturing **global relational structure**. Aligning these full distributions across views via KL divergence provides a stronger structure-aware constraint, leading to more stable and discriminative representations under domain shift. Parameter sensitivity of \(\lambda_{\text{IKT}}\) is provided in Appendix Fig. 6, showing that PRISM is robust to its variation.
>
> **Table 2. Performance comparison between the Plackett–Luce (PL) model, pairwise ranking, and listwise ranking surrogates**
>
> | Method | Real All | Real Old | Real New | Painting All | Painting Old | Painting New |
> |--------|-----------|-----------|-----------|----------------|----------------|----------------|
> | pairwise | 59.5 | 73.4 | 54.3 | 38.4 | 38.7 | 36.9 |
> | listwise | 59.8 | 73.7 | 54.8 | 38.8 | 38.6 | 37.5 |
> | **Plackett–Luce (PL)** | **60.9** | **74.1** | **55.1** | **39.2** | **39.0** | **38.2** |
>
> (iii) **Ablations on OT regularization \(\varepsilon\) and SAM iteration steps.**
> Appendix Fig. 6(e)–(f) shows that PRISM remains stable under moderate changes of \(\varepsilon\); overly large \(\varepsilon\) over-regularizes the transport plan and degrades performance, so we set \(\varepsilon = 0.5\). We also analyze the effect of SAM iteration steps. In practice, pseudo-label accuracy improves rapidly within the first 5–10 iterations as the transport plan captures global semantic correspondences. Between 10–20 iterations, the plan becomes sparser and more discriminative, achieving the best accuracy–sparsity–stability trade-off. Additional iterations bring diminishing gains while increasing computation. Therefore, 10–20 iterations are sufficient for stable and accurate SAM optimization.
> These ablations comprehensively validate the necessity and effectiveness of each key design component in PRISM.

---

> ### Author Response · Authors · 2025-11-24
> **respond to reviewer h5nf**
>
> ### **Q3. theory/intuition section on why high-frequency cues are more domain-invariant.**
> Thank you for the suggestion. We have added a theoretical justification in **Appendix A.2.2**, where we formally show that high-frequency components retain semantic structures that are less affected by domain-specific style variations (e.g., color, illumination, and texture bias). To further support this intuition empirically, we quantify the ability of low- and high-frequency features to separate known vs. unknown samples. As shown in **Table 3**, high-frequency cues consistently achieve higher discrimination accuracy across three challenging datasets, demonstrating that they contain **stronger domain-invariant and class-discriminative information**. This directly validates the motivation behind HCS and explains why leveraging high-frequency regions leads to more robust category separation under domain shifts.
>
> **Table 3. Comparison of the discriminative ability of original images versus high-frequency components for separating known and unknown samples.**
> | Method          | CUB-C | Scars-C | FGVC-C |
> |-----------------|----------|----------|----------|
> | original image          | 59.8     | 55.4    | 56.5     |
> | high frequency           | 73.7     | 68.7     | 69.8     |
>
> ### **Q4. Missing comparison to recent prototype-based approaches.**
> Thank you for pointing this out. To address your concern, we additionally compare PRISM with the recent prototype-based method Happy [1]. These results have now been included in Table 4 and Table 5. We observe that **Happy performs poorly under the challenging OW-CCD setting**, where the model must continually discover novel categories under domain shift without accessing past data. This highlights the intrinsic difficulty of OW-CCD and further demonstrates that **simply adopting prototype-based designs is insufficient** for handling continual domain drift. In contrast, PRISM’s divide-and-conquer separation (HCS) and stream-aware discovery mechanism enable robust performance in this challenging regime.
>
> **Table 4: Performance on the SSB-C Dataset (Happy vs. PRISM)**
> | Method | CUB-O All | CUB-O Old | CUB-O New | CUB-C All | CUB-C Old | CUB-C New | Scars-O All | Scars-O Old | Scars-O New | Scars-C All | Scars-C Old | Scars-C New | FGVC-O All | FGVC-O Old | FGVC-O New | FGVC-C All | FGVC-C Old | FGVC-C New |
> |--------|-----------|-----------|-----------|-----------|-----------|-----------|--------------|--------------|--------------|--------------|--------------|--------------|--------------|--------------|--------------|--------------|--------------|--------------|
> | **Happy** | 22.0 | 39.4 | 16.9 | 19.8 | 38.4 | 14.2 | 21.9 | 48.7 | 18.9 | 18.1 | 37.0 | 13.2 | 24.3 | 27.9 | 21.3 | 24.8 | 35.6 | 25.7 |
> | **PRISM** | **49.3** | **64.9** | **44.2** | **44.0** | **60.9** | **37.0** | **36.9** | **60.0** | **29.1** | **33.3** | **56.5** | **23.5** | **40.1** | **48.9** | **40.1** | **36.4** | **46.1** | **34.1** |
>
> **Table 5a: Real+Painting Performance on DomainNet**
> | Method | Real All | Real Old | Real New | Painting All | Painting Old | Painting New |
> |--------|----------|----------|----------|--------------|---------------|---------------|
> | **Happy** | 50.6 | 66.5 | 44.7 | 28.0 | 27.1 | 28.9 |
> | **PRISM** | 60.9 | 74.1 | 55.1 | 39.2 | 39.0 | 38.2 |
>
> **Table 5b: Real+Sketch Performance on DomainNet**
> | Method | Real All | Real Old | Real New | Sketch All | Sketch Old | Sketch New |
> |--------|----------|----------|----------|------------|-------------|-------------|
> | **Happy** | 52.0 | 65.0 | 41.2 | 11.2 | 15.6 | 10.7 |
> | **PRISM** | 60.1 | 73.4 | 51.0 | 16.9 | 20.1 | 15.9 |
>
> **Table 5c: Real+Quickdraw Performance on DomainNet**
> | Method | Real All | Real Old | Real New | Quickdraw All | Quickdraw Old | Quickdraw New |
> |--------|----------|----------|----------|---------------|----------------|----------------|
> | **Happy** | 35.6 | 51.4 | 28.9 | 4.6 | 4.5 | 5.2 |
> | **PRISM** | 54.0 | 74.0 | 49.2 | 7.1 | 6.5 | 7.4 |
>
> **Table 5d: Real+Clipart Performance on DomainNet**
> | Method | Real All | Real Old | Real New | Clipart All | Clipart Old | Clipart New |
> |--------|----------|----------|----------|-------------|--------------|--------------|
> | **Happy** | 45.6 | 62.4 | 37.1 | 12.0 | 19.6 | 9.0 |
> | **PRISM** | 58.0 | 72.3 | 51.2 | 24.0 | 30.4 | 19.1 |
>
> **Table 5e: Real+Infograph Performance on DomainNet**
> | Method | Real All | Real Old | Real New | Infograph All | Infograph Old | Infograph New |
> |--------|----------|----------|----------|----------------|-----------------|-----------------|
> | **Happy** | 34.2 | 50.5 | 28.0 | 7.9 | 9.4 | 5.6 |
> | **PRISM** | 60.1 | 73.8 | 53.1 | 10.9 | 14.1 | 9.8 |
>
> We appreciate your feedback and have incorporated all suggested results into the revised manuscript.

---

> > ### Author Response · Authors · 2025-11-24
> > **respond to reviewer h5nf**
> >
> > ### **Q5. Reliability of HCS when $S(x)$ is not clearly bimodal**
> > Thank you for this valuable question. We agree that on extremely fine-grained datasets, the separation score $S(x)$ may not always exhibit a perfectly bimodal structure due to the inherent semantic similarity among classes. Nonetheless, as shown in Appendix Fig. 7, even in these challenging scenarios, **HCS still yields a meaningful separation**: known-like and unknown-like samples tend to cluster around different regions because of their underlying semantic differences. Importantly, we also observe that leveraging **high-frequency cues consistently improves separability** compared to using raw images, further validating the effectiveness of HCS under domain overlap.
> >
> > Regarding the decision boundary, although a fixed threshold of **0.5** is commonly used in practice, **our framework does not rely on a manually chosen cutoff**. The two-component GMM fitted on $S(x)$ provides an *automatic*, *data-driven* boundary: the optimal decision point is given by the **intersection of the two Gaussian densities**, or equivalently, selecting the component with the higher **MAP (Maximum A Posteriori)** probability. This corresponds to the Bayes-optimal threshold for the mixture model and adapts naturally to each session’s distribution.
> >
> > In practice, we find that the posterior probabilities of the two mixture components are highly concentrated for known-like vs. unknown-like samples, making **0.5 effectively coincide with the MAP-based boundary**. Therefore, for simplicity and without extra hyperparameter tuning, we adopt 0.5 in all experiments while still retaining the theoretical ability to fall back to the fully automatic GMM-based threshold whenever needed.
> >
> > ### **Q6. How often are prototypes updated within and across sessions, and are HCS/IKT sensitive to stale prototypes?**
> > Prototypes are treated as **stable semantic anchors** and are updated **once every 30 epochs within a session**. This design follows the principle used in prior continual learning work (e.g., DEAN), ensuring stable optimization while preventing rapid overwriting of previously learned semantics. By avoiding frequent prototype recomputation—especially in the early stage of a session where pseudo-label noise is high under strong domain shift—we effectively reduce error propagation and preserve cross-session knowledge.
> >
> > Both HCS and IKT are **insensitive to prototypes remaining fixed for extended intervals**, as their mechanisms rely on domain-invariant semantic structure rather than fine-grained prototype refinement:
> > - **HCS** employs prototypes solely as high-level semantic anchors to estimate whether an unlabeled sample is known-like. Its decision is dominated by high-frequency structural cues, which are significantly more stable than small changes in prototype location.
> > - **IKT** operates on the **relative ranking** of similarities across known-class prototypes. Because the invariance constraint depends on stable *orderings* rather than absolute distances, IKT is naturally robust to moderate prototype drift within a session.
> >
> > Overall, this update schedule balances stability and adaptability: it prevents noisy early pseudo-labels from corrupting the semantic centers, maintains consistent anchors for HCS and IKT, and avoids the rapid forgetting that could occur if prototypes were updated too frequently during streaming adaptation.
> >
> > [1] Park, Keon-Hee, et al. "Online Continuous Generalized Category Discovery." European Conference on Computer Vision. Cham: Springer Nature Switzerland, 2024.

---

### Official Review · Reviewer_PEuj · 2025-10-31

**Soundness:** 3
**Presentation:** 3
**Contribution:** 3
**Rating:** 6
**Confidence:** 4

**Summary:**

This paper introduces a new category discovery task: Open-World Continual Category Discovery (OW-CCD), which challenges the single-domain data assumption in the previous CCD task.

**Strengths:**

1.	New problem setting:   This paper addresses a realistic open-world streaming setting (OW-CCD) that extends and generalizes prior continual and generalized category discovery work, directly tackling the limitations of stationary-domain assumptions in earlier CCD literature.
2.	The proposed method is interesting, especially since using the frequency domain to provide auxiliary supervision signals is reasonable.
3.	The proposed method achieved good results on the OW-CCD.

**Weaknesses:**

1.	I agree that introducing frequency domain analysis into the GCD task is an interesting and intuitively plausible approach. However, the paper's core claim that high-frequency components contain more domain-invariant category discriminative information lacks sufficient experimental support. The current Fig. 1 serves more as an idea demonstration than a rigorous quantitative argument. To enhance persuasiveness, the authors should provide more direct evidence, such as comparing the separation levels of known and unknown classes in high- and low-frequency feature spaces via t-SNE visualization, or quantifying the accuracy advantage of high- and low-frequency features in classifying new versus old categories.
2.	The proposed method introduces a non-trivial number of hyperparameters, among which the mask ratio r is particularly critical and appears to be highly sensitive.
3.	I don't entirely agree with the motivation behind the IKT section. Its optimization principle requires that, regardless of style variations, the similarity ranking between unknown samples and known classes remains consistent—essentially treating known classes as anchors for discovering new ones. However, whether this approach genuinely promotes effective discovery of new classes is debatable. Moreover, in current GCD/CCD research, known classes on some datasets exhibit low accuracy. I think this constraint may limit the representational space for new classes.

**Questions:**

See weaknesses.

---

> ### Author Response · Authors · 2025-11-24
> **respond to reviewer PEuj**
>
> ### **Q1. Response on the empirical evidence for high-frequency domain invariance**
>
> Thank you for the constructive suggestion. To directly address this concern, we conducted additional quantitative experiments to measure the discriminative power of high- versus low-frequency components in separating known and unknown classes. As reported in **Table 1**, we find that using high-frequency components leads to substantially higher accuracy in distinguishing known from unknown samples, consistently across datasets. This provides concrete evidence that high-frequency signals contain more domain-invariant and category-relevant information than low-frequency components, thereby offering strong empirical support for our core claim and validating the motivation behind the proposed HCS module. We will include these new results in the revised version to strengthen the quantitative justification.
>
> **Table 1. Comparison of the discriminative ability of original images versus high-frequency components for separating known and unknown samples**
> | Method          | CUB-C | Scars-C | FGVC-C |
> |-----------------|----------|----------|----------|
> | original image | 59.8  | 55.4 | 56.5|
> | high frequency  | 73.7 | 68.7 | 69.8 |
>
> ### **Q2. Response on hyperparameter sensitivity.**
> Thank you for highlighting this point. We provide a detailed hyperparameter analysis in Appendix Fig. 6, showing that PRISM is generally stable across most hyperparameters. While the mask ratio \(r\) is indeed an influential parameter—because it directly controls the separation of high- and low-frequency components that feed into subsequent modules—its sensitivity is well-bounded. As shown in Appendix Fig. 6, the performance remains stable within a broad range of \(r \in [0.1, 0.5]\). Performance degradation only occurs when \(r\) becomes excessively large, which is expected since an overly aggressive mask would remove essential semantic information from the high-frequency component. Overall, the method is **not** overly sensitive in practice, and a wide feasible range of \(r\) ensures that PRISM remains robust without requiring fine-grained tuning.
>
> ### **Q3. Response to concerns about the motivation of the IKT module**
> We appreciate your thoughtful critique. We would like to clarify that **IKT does not restrict the learning or representational capacity of novel classes**. Instead, our motivation is grounded in a central principle of category discovery: *leveraging semantic associations to transfer knowledge from known classes to unknown ones*. Humans naturally rely on such associations—for example, identifying the breed of an unseen dog by relating it to previously known dog categories, rather than to semantically distant classes such as mice.
> However, under domain shift, these true semantic relationships can be distorted by domain-specific style factors, leading the model to capture *spurious* rather than *meaningful* associations. Effective category discovery should therefore be guided by **domain-invariant semantic structure**, not domain-specific artifacts.
> IKT is designed precisely for this purpose: it explicitly models the relationships between unknown samples and known prototypes across domains, and enforces *consistency* of these relationships under style variation. Importantly, we only enforce **invariance of semantic associations**, not the features or representations of novel classes themselves. This does **not** limit the representational space of new categories; rather, it provides *stable, structured, and domain-invariant semantic guidance* that facilitates more reliable discovery of novel categories under domain shift.
> We will further clarify this motivation in the revised manuscript.

---

### Official Review · Reviewer_9cpT · 2025-11-01

**Soundness:** 2
**Presentation:** 3
**Contribution:** 2
**Rating:** 2
**Confidence:** 5

**Summary:**

The paper proposes a new setting called Open-World Continual Category Discovery (OW-CCD), arguing that existing CCD works usually assume a single domain, which is unrealistic for real data streams. To address this, the authors introduce PRISM, a three-part framework:

1. High-Frequency-Driven Category Separation (HCS): use Fourier-domain decomposition to keep high-frequency components and compute a density score over high-frequency features to separate “known-like” vs. “unknown-like” samples.
2. Sparse Assignment Matching (SAM): for samples considered “known-like”, perform proximal-regularized optimal transport to assign them to known-class prototypes, making the cluster assignment sparser and clearer.
3. Invariant Knowledge Transfer (IKT): for unknown samples, perform low-frequency perturbation and enforce ranking-level consistency to known prototypes using a Plackett–Luce distribution, so that semantic relations to known classes are invariant across domains.

Experiments on SSB-C and DomainNet show consistent gains over recent CCD baselines; ablations show that each of HCS / SAM / IKT contributes and all three together give the best numbers.

**Strengths:**

1. Plausible problem setting: the paper targets CCD under domain shift, which is a realistic gap in current CCD/GCD work.

2. Good empirical performance: shows strong results on multiple benchmarks.

3. Readable: pipeline (HCS → SAM → IKT) is clearly described and easy to follow.

**Weaknesses:**

1. **Weak and insufficiently argued link to continual learning.**
   The paper claims to tackle an open-world, continual setting, but the method itself is almost entirely tailored to domain shift. In practice, the pipeline treats the problem as “domain-shift GCD + CCD,” without discussing what actually becomes new or harder once we make it continual (e.g., catastrophic forgetting, order sensitivity, memory constraints). All three modules are clearly shift-oriented — HCS uses frequency cues to be domain-robust, SAM does cross-domain pseudo-labeling, and IKT enforces cross-domain relation consistency — yet none of them is a CL-specific mechanism. As a result, there is a visible gap between the problem the paper advertises (open-world *continual* category discovery) and the solution it delivers (a per-session, domain-shift-aware GCD pipeline). This naturally raises the question: why isn’t this simply framed as “domain-shift GCD” [1,2]? To make the claim convincing, the paper should spell out (i) what challenge is *unique* to their OW-CCD formulation and cannot be handled by combining existing domain-GCD [1, 2] and CCD methods, and (ii) how PRISM actually uses the *continual* nature of the stream, instead of just re-running the same 3-stage block on every session.

2. **Unclear tractability of IKT.**
   In IKT, the paper models the distribution over **all** permutations of the known-class prototypes using a Plackett–Luce distribution. This is conceptually reasonable, but as the number of known classes grows (e.g., C > 100), enumerating or even implicitly handling all C! permutations becomes infeasible. The paper does not explain how this is solved in the actual implementation. Without this detail, it is hard to judge whether IKT is an actually runnable module or mainly a theoretical description.

3. **Comparisons do not fully match the stated goal.**
   Since the method is explicitly designed to handle domain shift, it should also be compared to domain-shift GCD methods [1,2]. At minimum, the authors could (a) adapt HiLo [1] or CDAD-Net [2] to the same domain-shift + CCD, or (b) evaluate PRISM directly in a domain-GCD setup to demonstrate that it handles shift better than prior work [1, 2]. At present, most baselines are standard CCD methods that were *not* written for domain shift, so the performance gaps are less conclusive — the method may simply be capitalizing on a challenge (domain shift) that the competing methods were never tuned for.

[1] Hilo: A learning framework for generalized category discovery robust to domain shifts.

[2] Cdad-net: Bridging domain gaps in generalized category discovery.

**Questions:**

1. **Failure cases of HCS.** On which SSB-C corruption types does HCS fail to produce a clearly bimodal similarity distribution? Showing 1–2 negative examples in the appendix would make the claim “HCS yields separable scores” more convincing.

2. **Session/domain order.** On DomainNet, do you randomize the order of arriving domains, or do you always use the same source (e.g., Real) and the same progression of target domains? If the order is fixed, to what extent does PRISM rely on this order (i.e., does it overfit to an “easy → hard” schedule)? Have you tried choosing a non-Real source domain?

---

> ### Author Response · Authors · 2025-11-24
> **respond to reviewer 9cpT**
>
> ### **Q1. Response on the distinction between OW-CCD and existing GCD/CCD settings.**
> OW-CCD introduces challenges that cannot be addressed by simply combining existing cross-domain GCD and CCD methods. The task requires the model to continuously discover novel categories from a *streaming* unlabeled input, while preserving knowledge of previously learned known classes—**all without accessing any past labeled or unlabeled data**. This continual-learning constraint fundamentally differentiates OW-CCD from prior domain-shifted GCD works [1,2], which *all* assume that the model can jointly access labeled data from the previous stage and unlabeled data from the current stage.
> As an example, we carefully reproduced the technical pipeline of **HiLo** [1] and found that it depends on **semi-supervised k-means** to generate pseudo-labels for computing its loss (Eq. 8) and to obtain domain labels for curriculum learning. However, semi-supervised k-means **inherently requires simultaneous access** to the previous stage’s labeled data and the current stage’s unlabeled stream—an assumption that is incompatible with the continual setting of OW-CCD. Similarly, CCD methods do not model domain drift and therefore degrade under strong distribution shifts.
> Consequently, **OW-CCD cannot be formulated as “GCD under domain shift”**, because the continual constraint creates a strictly different problem regime where existing methods cannot operate.
> In contrast, our **PRISM** framework is explicitly designed for OW-CCD:
> - it does **not** require access to historical data,
> - it uses **HCS** to perform a divide-and-conquer separation of known and unknown samples,
> - and it enables dynamic and robust category discovery directly from a continuously evolving data stream.
>
> This establishes a clear, necessary, and well-motivated link to continual learning. We have also revised the introduction to more clearly articulate the fundamental challenges of OW-CCD and explicitly distinguish it from previous GCD and CCD settings.
>
> [1] Wang, Hongjun, Sagar Vaze, and Kai Han. "Hilo: A learning framework for generalized category discovery robust to domain shifts." arXiv preprint arXiv:2408.04591 (2024).
>
> [2] Rongali, Sai Bhargav, et al. "Cdad-net: Bridging domain gaps in generalized category discovery." Proceedings of the IEEE/CVF Conference on Computer Vision and Pattern Recognition. 2024.
>
> ### **Q2. Response on the tractability of IKT and the use of the Plackett–Luce model.**
> We thank the reviewer for raising the concern regarding the \(C!\) support size of the full Plackett–Luce (PL) distribution. Although the PL model is mathematically defined over all permutations, **our implementation does not enumerate or construct any permutation**. Following standard practice in learning-to-rank (e.g., ListMLE[1], ListNet[2]), we rely on the **factorized log-likelihood form** of the PL model.
>
> In this form, the PL likelihood decomposes into a sequence of local log-probability terms, each involving only a `log` strength value and a `log-sum-exp` normalization over the strength vector $\kappa$. The IKT module computes the divergence between two PL distributions **using these local log-likelihood components**, which depend solely on the strength vectors and **do not require accessing any permutation $\xi$**. This yields a tractable surrogate to the theoretical KL objective.
> As a result, **there is no factorial blow-up**, and the runtime overhead of IKT is negligible (<0.04 s per update on an A100). We will clarify this implementation detail in the revision. The IKT module is fully runnable, efficient, and consistent with the theoretical PL formulation presented in the paper.
>
> [1] Xia, Fen, et al. "Listwise approach to learning to rank: theory and algorithm." Proceedings of the 25th international conference on Machine learning. 2008.
>
> [2] Cao, Zhe, et al. "Learning to rank: from pairwise approach to listwise approach." Proceedings of the 24th international conference on Machine learning. 2007.

---

> ### Author Response · Authors · 2025-11-24
> **respond to reviewer 9cpT**
>
> ### **Q3. Response to Comparisons do not fully match the stated goal.**
>
> Thank you for this insightful suggestion. As we clarified in our response to Q1, existing domain-shift GCD methods such as HiLo [1] and CDAD-Net [2] cannot be directly adapted to the **OW-CCD** setting because both methods require *simultaneous access* to past labeled data and current unlabeled data—an assumption fundamentally incompatible with the continual-learning constraint of OW-CCD. Therefore, these methods cannot serve as baselines under the proposed setting.
>
> Nevertheless, to fully address your concern, we additionally **evaluate PRISM in a standard cross-domain GCD setup**, following the exact experimental protocol of HiLo. This corresponds to the reviewer’s suggestion (b) of testing PRISM directly in a domain-GCD regime. We also include comparisons to both HiLo and CDAD-Net under this setting.
> Please note that this evaluation is **not** OW-CCD, but the conventional domain-shift GCD setting where all methods are applicable.
>
> The results are reported in Table 1, 2 (**Appendix Table 17 and Table 18**), and they show that **PRISM outperforms HiLo and CDAD-Net in most cases**. This demonstrates that:
> - PRISM is indeed capable of handling domain shift effectively,
> - PRISM’s improvements are not merely due to competing CCD baselines being non-domain-aware,
> - and PRISM remains competitive even in the standard cross-domain GCD scenario.
> These additional experiments confirm that our method is **not tailored merely to exploit a weakness of CCD baselines**, but provides a generally strong and flexible solution for category discovery under distribution shift.
>
> **Table 1: Clustering results of various methods on the SSB-C benchmark
> For each dataset (CUB, Scars, FGVC), the clean set serves as the source domain,
> and the corrupted set is treated as the target domain**
>
> | Method     | CUB-O All | CUB-O Old | CUB-O New | CUB-C All | CUB-C Old | CUB-C New | Scars-O All | Scars-O Old | Scars-O New | Scars-C All | Scars-C Old | Scars-C New | FGVC-O All | FGVC-O Old | FGVC-O New | FGVC-C All | FGVC-C Old | FGVC-C New |
> |--|-|-|-|-|-|--|-|-|-|-|-|-|-|-|-|-|-|-|
> | **CDAD-Net** | 40.4 | 38.9 | 39.3 | 37.7 | 39.1 | 34.2 | 32.1 | 42.9 | 32.2 | 28.8 | 35.6 | 21.4 | 33.8 | 35.5 | 31.2 | 27.8 | 29.6 | 25.6 |
> | **HiLo**     | 56.8 | 54.0 | 60.3 | 52.0 | 53.6 | 50.5 | 39.5 | 44.8 | 37.0 | 35.6 | 42.9 | 28.4 | 44.2 | 50.6 | 47.4 | 31.2 | 29.0 | 33.4 |
> | **PRISM**    | 60.1 | 58.7 | 63.1 | 56.2 | 55.1 | 54.9 | 44.0 | 47.4 | 40.6 | 40.1 | 43.5 | 34.5 | 47.9 | 55.1 | 51.8 | 35.7 | 31.8 | 39.1 |
>
> **Table 2a: Real + Painting Performance on DomainNet**
> | Method     | Real All | Real Old | Real New | Painting All | Painting Old | Painting New |
> |------------|----------|----------|----------|--------------|--------------|--------------|
> | **CDAD-Net** | 63.6 | 77.9 | 56.3 | 38.4 | 38.4 | 37.5 |
> | **HiLo**     | 64.4 | 77.6 | 57.5 | 42.1 | 42.9 | 41.3 |
> | **PRISM**    | 68.7 | 77.8 | 63.3 | 47.2 | 46.8 | 45.8 |
>
> **Table 2b: Real + Sketch Performance on DomainNet**
> | Method     | Real All | Real Old | Real New | Sketch All | Sketch Old | Sketch New |
> |------------|----------|----------|----------|------------|------------|------------|
> | **CDAD-Net** | 61.9 | 76.3 | 52.1 | 17.3 | 20.9 | 15.9 |
> | **HiLo**     | 63.3 | 77.9 | 55.9 | 19.4 | 22.4 | 17.1 |
> | **PRISM**    | 68.7 | 78.2 | 63.0 | 23.8 | 24.9 | 22.8 |
>
> **Table 2c: Real + Quickdraw Performance on DomainNet**
> | Method     | Real All | Real Old | Real New | Quickdraw All | Quickdraw Old | Quickdraw New |
> |------------|----------|----------|----------|----------------|----------------|----------------|
> | **CDAD-Net** | 48.5 | 66.5 | 36.7 | 6.4 | 5.6 | 7.3 |
> | **HiLo**     | 58.6 | 76.4 | 52.5 | 7.4 | 6.9 | 8.0 |
> | **PRISM**    | 61.8 | 77.5 | 57.1 | 9.0 | 7.3 | 9.4 |
>
> **Table 2d: Real + Clipart Performance on DomainNet**
> | Method     | Real All | Real Old | Real New | Clipart All | Clipart Old | Clipart New |
> |------------|----------|----------|----------|-------------|-------------|-------------|
> | **CDAD-Net** | 61.3 | 77.0 | 53.1 | 25.2 | 31.9 | 19.0 |
> | **HiLo**     | 63.8 | 77.6 | 56.6 | 27.7 | 34.6 | 21.7 |
> | **PRISM**    | 67.2 | 77.4 | 62.0 | 30.4 | 36.4 | 27.6 |
>
> **Table 2e: Real + Infograph Performance on DomainNet**
> | Method     | Real All | Real Old | Real New | Infograph All | Infograph Old | Infograph New |
> |------------|----------|----------|----------|----------------|----------------|----------------|
> | **CDAD-Net** | 56.5 | 68.0 | 47.1 | 11.8 | 15.6 | 9.4 |
> | **HiLo**     | 64.2 | 78.1 | 57.0 | 13.7 | 16.4 | 11.9 |
> | **PRISM**    | 69.1 | 77.6 | 61.2 | 16.7 | 18.7 | 14.3 |

---

> > ### Author Response · Authors · 2025-11-24
> > **respond to reviewer 9cpT**
> >
> > ### **Q4. Response to Failure cases of HCS.**
> >
> > Thank you for the question. HCS relies primarily on semantic high-frequency cues such as edges and fine textures. Among all SSB-C corruptions, the most challenging ones are **structural blur types** (e.g., zoom blur), which smear object boundaries and directly suppress the high-frequency components that HCS depends on. These cases can weaken the bimodal shape of the similarity distribution. Importantly, **Appendix Fig.~8 already visualizes exactly these negative examples**, showing that although structural blur reduces the clarity of bimodality, HCS still achieves *better separation than raw images*. This is because HCS continues to suppress domain-specific low-frequency biases and retain more discriminative structures than full-spectrum inputs. We will explicitly highlight in the appendix that Fig.~8 corresponds to the failure cases requested, thereby making the behavior of HCS under challenging corruptions fully transparent.
> >
> > ### **Q5. Response on session/domain order in DomainNet.**
> >
> > Thank you for the constructive suggestion. We have thoroughly examined the effect of domain-arrival order on PRISM. Specifically, we evaluated three different scheduling strategies:
> > (1) **Fixed-domain per stage**, where each stage samples from a single domain (results in Table 1);
> > (2) **Randomized multi-domain sampling**, where each stage draws data randomly from multiple domains of varying difficulty (Table 2, Appendix Table 13); and
> > (3) **Progressively diversified domains**, where each stage gradually increases domain diversity by sampling from different domains over time (Appendix Table 15).
> > Across all three settings, **PRISM consistently achieves strong and stable performance**, indicating that it does *not* rely on a fixed “easy → hard’’ schedule and does *not* overfit to a particular domain order. These results also demonstrate that PRISM remains robust even when domains arrive in arbitrary sequences. Overall, this confirms that PRISM is insensitive to domain-arrival patterns and is broadly effective under diverse domain ordering strategies.

---

> > > ### Comment · Reviewer_9cpT · 2025-11-24
> > >
> > > I thank the authors for their detailed rebuttal and the significant effort put into the additional experiments.
> > >
> > > I am satisfied with the responses to Q1, Q3, Q4, and Q5. The new comparisons with HiLo/CDAD-Net and the analysis of HCS failure cases are particularly persuasive and have well addressed my concerns regarding baseline fairness and method robustness.
> > >
> > > Regarding Q2, the explanation partially resolves my concern about tractability. However, I note that the cited methods (ListMLE [1], ListNet [2]) typically optimize a surrogate or approximation rather than the exact KL divergence over the full Plackett-Luce distribution as defined in Eq. 12. This creates a discrepancy between the theoretical description and the actual implementation. I strongly suggest clarifying this in the main text and explicitly stating the actual surrogate loss function used in your code to avoid potential misunderstanding.
> > >
> > > Considering the solid empirical evidence provided during the rebuttal, I am raising my score to marginally above the acceptance threshold.

---

> ### Author Response · Authors · 2025-12-01
> **Thank you for raising your score**
>
> **We sincerely thank you for your prompt response and for raising your score. We are greatly encouraged by your recognition of our extended experiments and clarifications. We will ensure that the additional baseline comparisons and the ablation analysis are fully incorporated into the final version to further strengthen the paper.**
>
> For Q2, the PL model admits a well-known factorized form, where the likelihood decomposes into a sequence of local log-probability terms. The KL divergence between two PL distributions therefore also factorizes, and can be computed exactly using these per-position log-probabilities—without enumerating permutations or relying on surrogate approximations. Our implementation follows this exact decomposition, consistent with the analytic form used in ListMLE-style learning-to-rank methods. We have clarified this in the main text; please see the revised version with dark-red highlighted modifications.
>
> If you have any remaining questions or if there are other details we can clarify during the discussion period, please do not hesitate to let us know. We are more than happy to address them.

---

### Meta-Review · Area_Chair_ruki · 2026-01-06

**Summary:**

This paper introduces a more challenging setting for CCD with domain shift and accordingly designs a three-stage framework to address it.  The concerns raised by three reviewers have been clearly and satisfactorily addressed in the rebuttal, and these reviewers have subsequently provided positive evaluations. One reviewer gave a negative initial assessment but did not participate in the rebuttal discussion. After carefully examining the authors’ rebuttal, I find that the majority of this reviewer’s concerns have also been adequately resolved. I recommend that the authors ensure all promised changes made in response to the reviewers’ comments are fully incorporated into the revised manuscript.
Overall, I believe the paper meets the standards of the conference, and I therefore recommend acceptance.

**Reviewer Concerns:**

The concerns about the reasonableness of the proposed setting from Reviewer 9cpT and qfwv are addressed.
The concerns about the quantitative evidence on high-frequency invariance, hyperparameter analysis, and clarification of IKT’s motivation from Reviewer PEuj are addressed.

**Reviewer Scores:**

If Reviewer h5nf had been able to participate fully in the discussion, their initial rating would likely have been revised. The concerns raised by Reviewer h5nf are addressed in the authors’ rebuttal. Notably, Reviewer h5nf initially assigned a score of 4, indicating a marginally below-threshold evaluation and explicitly stating that they would not mind if the paper were accepted.

---

### Decision · Program_Chairs · 2026-01-26

Accept (Poster)